# Rooted Absorbed Prefix Trajectory Balance with Submodular Replay for GFlowNet Training

Xi Wang [1]   Wenbo Lu [1]   Shengjie Wang [1]

## Abstract

Generative Flow Networks (GFlowNets) enable fine-tuning large language models to approximate reward-proportional posteriors, but they remain prone to mode collapse, manifesting as prefix collapse and length bias. We attribute this to two factors: (i) weak credit assignment to early prefixes, and (ii) biased replay that induces a shifted, non-representative training flow distribution. We propose Rooted absorbed prefix Trajectory Balance (**RapTB**), an objective that anchors subtrajectory supervision at the root and propagates terminal rewards to intermediate prefixes via absorbed suffix-based backups, providing dense prefix-level learning signals. To mitigate replay-induced distribution shift, we further introduce **SubM**, a submodular replay refresh strategy that promotes both high reward and diversity. Empirically, on tasks such as molecule generation with LLM using SMILES strings, RapTB combined with SubM consistently improves optimization performance and molecular diversity while preserving high validity. The code is released on https://github.com/ComDec/ChemGFN.

## 1. Introduction

Generative Flow Networks (GFlowNets) learn a stochastic policy on a directed acyclic graph (DAG) that constructs objects sequentially, so that completed trajectories are sampled with probability proportional to the rewards (Bengio et al., 2021; 2023; Hu et al., 2023). In contrast to reward-maximizing reinforcement learning, the objective of GFlowNets is distributional: spread probability mass across many high-reward modes in proportion to reward, rather than concentrating on a single optimum (Kaelbling

et al., 1996). The GFlowNet objective is distributional: spread probability mass across many high-reward modes in proportion to reward, rather than concentrating on a single optimum. This objective admits an equivalent entropy-regularized RL formulation (Tiapkin et al., 2024; Deleu et al., 2024); our contributions operate within this off-policy regime, improving credit assignment and replay coverage for TB-family balance objectives on terminable prefix trees. This method extends naturally to large language models (LLMs) in a terminable prefix-tree formulation (Hu et al., 2024). Every prefix state has an explicit termination edge to a terminal node. The termination edge can be implemented as an EOS action.

In practice, LLM-GFlowNets suffer from mode collapse. We identify two specific failures: (i) prefix collapse, where early-token entropy drops sharply and distinct terminals share near-identical prefixes; and (ii) length bias, where the model favors sequences that are systematically too short or too long. We trace these issues to two factors: (i) weak credit assignment, as terminal-only rewards provide high-variance and ambiguous feedback for intermediate steps (Madan et al., 2023); and (ii) replay bias, where training is confined to a tiny fraction of the search space, and repeated reinforcement of this narrow subset causes the distribution to collapse (Shen et al., 2023).

We address these failure modes with two complementary mechanisms: one that strengthens prefix-level credit assignment and the other broadens the support of replay. **RapTB** retains terminal Trajectory Balance (TB) (Malkin et al., 2022a) as the primary constraint and adds a lightweight rooted-prefix objective that provides supervision at intermediate prefixes. Concretely, RapTB densifies training signals by propagating the terminal reward to each prefix via suffix-based backups. In parallel, **SubM** refreshes the replay buffer by selecting a subset of trajectories that maximizes a submodular objective over candidates. The objective jointly encourages high reward, trajectory diversity, and length coverage, expanding the support of the training distribution. Across molecular and arithmetic generation tasks, we find that TB objective (Malkin et al., 2022a) can quickly over-concentrate on shared prefixes, while Subtrajectory Balance (SubTB) (Madan et al., 2023; Hu et al., 2024) can drift in its

[1]Courant Institute School of Mathematics, Computing, and Data Science, New York University. Correspondence to: Shengjie Wang <sw5973@nyu.edu>.

*Proceedings of the 43rd International Conference on Machine Learning*, Seoul, South Korea. PMLR 306, 2026. Copyright 2026 by the author(s).

termination probabilities. RapTB mitigates both prefix collapse and length bias, and SubM further improves coverage and distribution matching.

**Contributions.**

- We empirically characterize mode collapse in LLM-GFlowNets as a reproducible combination of prefix collapse and length bias. We provide evidence that it is driven by high-variance terminal credit assignment and replay-induced training distribution shifts.

- We propose RapTB, which augments TB with a rooted prefix objective that propagates terminal reward to intermediate prefixes via suffix-based backups. It provides dense training signals and reduces variance.

- We introduce Submodular Replay (SubM), a replay refresh rule that balances reward, diversity, and length coverage in one submodular objective. It improves replay coverage and stabilizes training. We validate RapTB+SubM across five tasks, model scales up to 32B, and comparisons against RL and alternative GFlowNet baselines (Appendices B–C).

**Positioning.** We focus on TB-family objectives. While TB suffers from high variance under terminal rewards (Madan et al., 2023), existing subtrajectory methods for LLMs induce *termination drift* via conflicting overlapping constraints. RapTB resolves this by restricting dense supervision to rooted prefixes using variance-reduced absorbed targets. This formulation eliminates destabilizing boundary conditions and explicitly detaches auxiliary termination gradients to prevent drift, all while preserving the global TB anchor. Orthogonally, SubM stabilizes replay via coverage-aware subset refresh to mitigate the reward-tilted collapse observed in prior work.

## 2. Related Work

**GFlowNets and applications.** GFlowNets learn generative policies whose terminal distribution is proportional to reward (Bengio et al., 2021; 2023). Trajectory Balance stabilizes training via global path consistency (Malkin et al., 2022a), and subsequent work connects GFlowNets to variational inference and policy gradients (Malkin et al., 2022b; Niu et al., 2024). The framework has been applied broadly in scientific discovery (Jain et al., 2022; 2023; Nguyen et al., 2023; Hernandez-Garcia et al., 2023). To reduce variance, Subtrajectory Balance introduces dense subtrajectory constraints (Madan et al., 2023), but directly adapting it to terminable prefix trees for LLMs (Hu et al., 2024) can introduce termination drift. RapTB targets this structural mismatch by restricting dense supervision to rooted prefixes while using suffix-absorbed targets for lower-variance prefix credit.

**Experience replay and exploration in GFlowNets.** Experience replay improves sample efficiency in GFlowNets, but reward-prioritized replay can induce rich-get-richer collapse and reduce coverage (Shen et al., 2023; Vemgal et al., 2023; Hu et al., 2024). Prior work mitigates this with heavier exploration mechanisms, including local search (Kim et al., 2023), evolutionary population maintenance (Ikram et al., 2024), MCTS-style rollouts (Morozov et al., 2024), and retrospective synthesis (He et al., 2024). In contrast, we formulate replay refresh as lightweight submodular maximization with a greedy near-optimality guarantee (Kirchhoff & Bilmes, 2014; Kothawade et al., 2022; Killamsetty et al., 2021), selecting a subset that jointly balances reward, diversity, and length coverage. Concurrently, TBA (Bartoldson et al., 2026) scales TB-based LLM post-training via asynchronous exploration and importance-weighted replay; its systems-level contributions are orthogonal to our objective-level improvements, and RapTB can in principle replace the TB loss within TBA's framework.

## 3. Method

### 3.1. GFlowNets on Terminable Prefix Trees

We consider a GFlowNet defined on a directed acyclic graph (DAG) with a unique source $s_0$ and terminal set $\mathcal{X}$. A trajectory $\xi = (s_0 \rightarrow \cdots \rightarrow s_\tau \rightarrow x)$ ends at a terminal node $x$, and the target terminal distribution satisfies $p^\star(x) \propto R(x)$ (Bengio et al., 2021; 2023).

Following Hu et al. (2024), we instantiate a GFlowNet on the prefix tree induced by an autoregressive language model augmented with a stop symbol $\top$ (i.e., each prefix has a termination action), such as an End-of-Sentence (EOS) token. Given a prompt, the LLM generates a sequence of tokens. As the prompt is held constant for the generation process, we omit it in the following for simplicity. Therefore, the GFlowNet state is represented by a generated prefix $s_{0:i}$.

From state $s_{0:i}$, the forward policy either emits a token $s_{i+1} \in \mathcal{V}$ (the vocabulary) and transitions to $s_{0:i+1}$, or terminates by emitting $\top$. We parameterize the forward policy by the language model distribution $q_\theta(\cdot \mid s_{0:i})$ over $\mathcal{V} \cup \{\top\}$, and identify

$$P_F^\theta(s_{0:i+1} \mid s_{0:i}) = q_\theta(s_{i+1} \mid s_{0:i}),$$
$$P_F^\theta(s_{0:i}^\top \mid s_{0:i}) = q_\theta(\top \mid s_{0:i}).$$

where $s_{0:i}^\top$ denotes the terminated sequence of $s_{0:i}$ by appending the terminal token $\top$, i.e., $(s_0, s_1, \ldots, s_i, \top)$.

Because each non-root prefix has a unique parent, the backward kernel is deterministic, and the state space forms a tree. Therefore, the probability of sampling a terminated

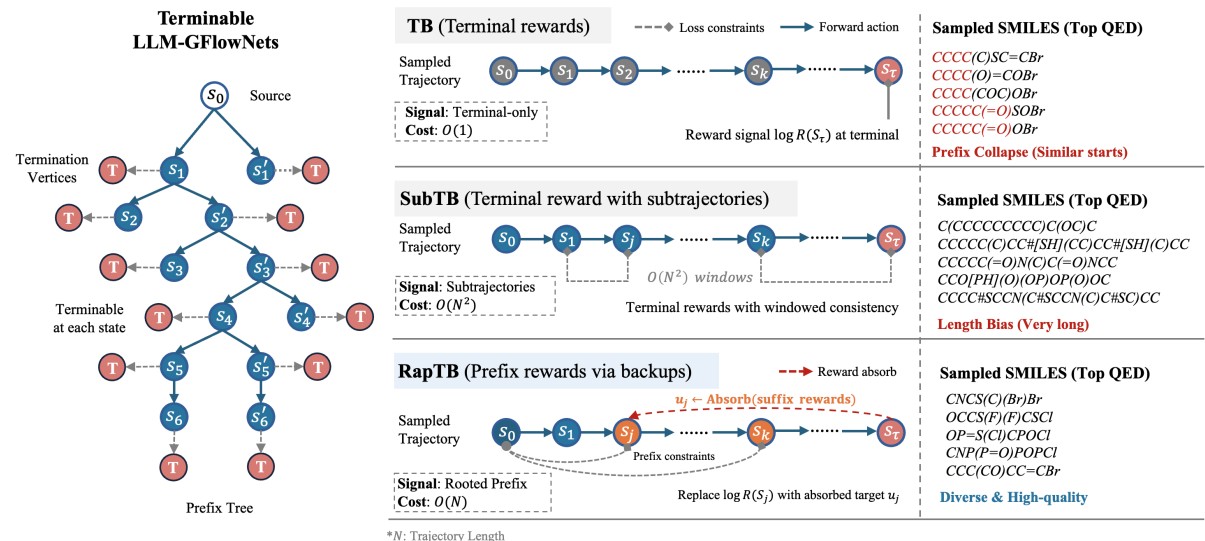

*Figure 1.* **Training objectives for LLM-GFlowNets.** TB uses only terminal reward $\log R(s_\tau)$ ($O(1)$). SubTB adds $O(N^2)$ windowed consistency constraints. RapTB replaces prefix stop-rewards with suffix-absorbed targets $u_j$ and applies $O(N)$ rooted prefix constraints. ($N$: trajectory length). The right column shows examples of generated molecules trained with different losses. QED (Quantitative Estimate of Drug-likeness) is a comprehensive metric for measuring a molecule's drug-likeness and is used as the task reward during training.

sequence $\xi = s_{0:\tau}^\top$ factorizes as

$$q_\theta^\top(\xi) = \Big( \prod_{i=0}^{\tau-1} q_\theta(s_{i+1} \mid s_{0:i}) \Big) q_\theta(\top \mid s_{0:\tau}).$$

**Mixed reward.** To stabilize exploration under sparse or high-variance task rewards, for any prefix $s_{0:i}$, we define the stop-reward as a mixture of (i) a frozen reference language model prior and (ii) an external task-specific score, following Hu et al. (2024). Let $P_{\text{ref}}$ denote the probability assigned by a fixed, pre-trained reference LM to a terminated sequence, which serves as a prior that regularizes generation toward fluent and well-formed outputs. For a prefix $s_{0:i}$, we define the mixed stop-reward as

$$\log R(s_{0:i}^\top) = \kappa \log P_{\text{ref}}(s_{0:i}^\top) + \lambda S(s_{0:i}^\top),$$

where $S(s_{1:i}^\top)$ is the task-only reward component and $\kappa, \lambda \geq 0$ control the mixture ratio. The task-only component $S(\cdot)$ depends on the application; for example, in small-molecule generation it may correspond to a property score (e.g., binding affinity or drug-likeness), while in symbolic expression generation it may reflect functional correctness or coverage. The exact reward definitions and ablation studies are provided in Appendix F.1.

### 3.2. Trajectory Balance: Global Path Consistency

Trajectory Balance (Malkin et al., 2022a) enforces global consistency between forward trajectories and terminal rewards by introducing a learnable normalizer $Z_\theta > 0$. For a

terminated trajectory $\xi = s_{0:\tau}^\top$, the TB log-residual is

$$\Delta^{\text{TB}}(\xi) = \log Z_\theta + \sum_{i=0}^{\tau-1} \log P_F^\theta(s_{i+1} \mid s_{0:i}) \\ + \log P_F^\theta(\top \mid s_{0:\tau}) - \log R(s_{0:\tau}^\top), \quad (1)$$

and TB minimizes

$$\mathcal{L}_{\text{TB}} = \mathbb{E}_{\xi \sim P_F^\theta}\big[\Delta^{\text{TB}}(\xi)^2\big].$$

TB provides a clean global anchor for reward-proportional sampling, but terminal-only rewards yield high-variance credit assignment on long horizons. In practice, a few high-reward trajectories dominate updates; since many trajectories share early prefixes, these prefixes are repeatedly reinforced while alternatives are under-trained, leading to self-reinforcing prefix collapse and reduced diversity (Shen et al., 2023; Madan et al., 2023).

### 3.3. Subtrajectory Balance: Dense but Over-Constrained Supervision

Subtrajectory Balance (Madan et al., 2023) reduces variance by enforcing TB-style consistency on subtrajectories. Following Hu et al. (2023), for any subtrajectory indexed by $i, j, 0 \leq i < j \leq \tau$, define $\Delta_{i \to j}^{\text{SubTB}}(\xi)$ as

$$\sum_{k=i}^{j-1} \log P_F^\theta(s_{k+1} \mid s_{0:k}) + \log R(s_{0:i}^\top) - \log R(s_{0:j}^\top) \\ + \log P_F^\theta(\top \mid s_{0:j}) - \log P_F^\theta(\top \mid s_{0:i}), \quad (2)$$

The overall objective is the combination of all subtrajectory objectives, optionally weighted by $w_l$ based on the length of the subtrajectory:

$$\mathcal{L}_{\text{SubTB}}(\xi) \triangleq \frac{\sum_{\ell=1}^{\tau} w_\ell \sum_{i=0}^{\tau-\ell} \left(\Delta_{i \to i+\ell}^{\text{SubTB}}(\xi)\right)^2}{\sum_{\ell=1}^{\tau} w_\ell \sum_{i=0}^{\tau-\ell} 1}, \quad (3)$$

While SubTB provides dense supervision by enforcing consistency on many subtrajectories, in terminable prefix trees it introduces a large number of overlapping constraints that share the same termination head. Each subtrajectory implicitly treats its endpoint as a pseudo-terminal, imposing a distinct boundary condition involving the termination probability $q_\theta(\top \mid s_{0:i})$. These heterogeneous boundary conditions are difficult to satisfy simultaneously, and gradients from many windows accumulate on the shared termination logits. As a result, the model can reduce SubTB residuals by adjusting termination probabilities rather than improving token-level transitions, leading to biased termination behavior such as systematic length drift. This over-constraining effect motivates our more conservative approach to densifying prefix-level supervision.

### 3.4. RapTB: Rooted Absorbed Prefix Trajectory Balance

RapTB addresses the trade-off between TB's high variance and SubTB's over-constrained objective. It restricts dense supervision to rooted-prefix residuals, reducing destabilizing boundary conditions, and utilizes partial credit as additional training target by "absorbing" credits from suffixes.

**Rooted Prefix Residuals.** Instead of constraining all subtrajectories as in SubTB, we only focus on subtrajectories that are "rooted," originating from $s_0$ (Appendix F.2). The rooted residual at step $k$ is defined as the difference between the TB residual of the current prefix and the root:

$$\bar{\Delta}_k(\xi) \triangleq \Delta_k^{\text{TB}}(\xi) - \Delta_0^{\text{TB}}(\xi). \quad (4)$$

By eliminating the global constant $\log Z_\theta$, this formulation creates a local consistency signal anchored to $s_0$. Unlike SubTB, which creates conflicting boundary conditions via overlapping windows, our approach provides incremental, step-by-step supervision.

**Absorbed Suffix Rewards.** To stabilize training against stochastic variance, we introduce the absorbed suffix reward. This approach constructs a lower-variance target by backing up the rewards from the observed suffix $s_{k:\tau}$, employing an aggregation mechanism (See details in Appendix F.3). This distills hindsight information into a smoothed signal, guiding the policy more reliably than terminal feedback (See analysis in Appendix F.5). For a trajectory $s_{0:\tau}$, let $u_j$

denote the task-only reward at position $j$, i.e., $\lambda S(s_{0:j}^\top)$, and we define the absorbed target at $k$ as:

$$u_k^{\max} \triangleq \max_{j \in [k,\tau]} u_j, \quad (5)$$

$$u_k^{soft} \triangleq \frac{1}{\beta} \log \sum_{j=k}^{\tau} \exp\left(\beta u_j - \beta \rho (j-k)\right), \quad (6)$$

$$u_k^{tgt} \triangleq \alpha\, u_k^{\max} + (1-\alpha)\, u_k^{soft}, \qquad \alpha \in [0,1]. \quad (7)$$

The $u_k^{\max}$ provides a lower bound on prefix credit: the credit (or size of the flow in GFlowNet) in the prefix should be no smaller than the one in its continuation. The $u_k^{soft}$ term smoothly aggregates multiple suffix rewards in log space, where $\beta > 0$ controls the smoothness. The distance penalty $\rho(j-k)$ downweights distant evidence, and $\rho \geq 0$ controls the penalty strength.

The absorbed target $u_k^{tgt}$ is the task-only partial credit assigned to the prefix $s_{0:k}$. We can then treat it as an "estimated reward" for $s_{0:k}$ and train the model to match it. Operationally, this is equivalent to recomputing the rooted TB residual using a surrogate stop-reward in which the task-only component $u_k$ is replaced by $u_k^{tgt}$ (details in Appendix F):

$$\mathcal{L}_{\text{aux}}(\xi) \triangleq \frac{\sum_{k=1}^{\tau} w_k \left(\bar{\Delta}_k(\xi) + u_k - u_k^{tgt}\right)^2}{\sum_{k=1}^{\tau} w_k}, \quad (8)$$

where $w_k$ is the length weight.

**Final Objective.** The RapTB objective integrates global TB consistency with this dense guidance.

$$\mathcal{L}_{\text{RapTB}} = \mathbb{E}_{\xi \sim P_F^\theta}\Big[\underbrace{\Delta^{\text{TB}}(\xi)^2}_{\text{Anchor}} + \underbrace{\eta\, \mathcal{L}_{\text{aux}}(\xi)}_{\text{Partial Credit}}\Big], \quad (9)$$

where $\eta$ balances global consistency with auxiliary term. The TB term remains the only exact balance condition whose optimum matches the reward-proportional target. The auxiliary term is a variance-reducing regularizer that can improves optimization.

**Fixed-point property.** Without absorption ($u_k^{tgt}=u_k$), the rooted residual $\bar{\Delta}_k$ vanishes whenever all prefix TB residuals are zero, so RapTB shares TB's global optimum exactly. With absorption, $u_k^{tgt} \geq u_k$ (the suffix set includes $j=k$), so the absorbed residual is generally nonzero at the TB optimum. Let $\theta^*$ satisfy $\mathcal{L}_{\text{TB}}(\theta^*)=0$ with finite auxiliary cost $C^*=\eta\mathcal{L}_{\text{aux}}(\theta^*)$. For any global minimizer $\hat{\theta}$ of $\mathcal{L}_{\text{RapTB}}$: $\mathcal{L}_{\text{TB}}(\hat{\theta}) \leq \mathcal{L}_{\text{RapTB}}(\hat{\theta}) \leq C^*$, so the TB deviation is bounded by $\eta\mathcal{L}_{\text{aux}}(\theta^*)$ and vanishes as $\eta \to 0$. The auxiliary term thus acts as a variance-reducing regularizer that does not destroy the global TB anchor (Appendix F.5).

**Compare three losses.** TB uses a global objective with sparse supervision, while SubTB increases supervision density but over-constrains hinder optimization in prefix trees. RapTB preserves TB as the sole exact balance constraint and adds (i) rooted prefix supervision that avoids heterogeneous window boundaries and (ii) absorbed suffix rewards that reduce variance and improve credit assignment.

### 3.5. Submodular Replay: Diversity- and Length-balanced Experience Selection

RapTB addresses within-trajectory credit assignment. In parallel, to explicitly enforce diversity in addition to the exploration of GFlowNet, we maintain a fixed-size replay buffer of size $B$ and update it by selecting a representative, diverse, and length-balanced subset from the union of the current buffer and a newly collected batch (details in Appendix G.4.1).

**Submodular selection.** At each buffer update step, we form the ground set $\mathcal{G}$ as the union of the current buffer and a new generated batch, then select $S \subseteq \mathcal{G}$ with $|S| = B$ by maximizing a monotone submodular function subject to a cardinality constraint (Killamsetty et al., 2021; Kothawade et al., 2022; Kirchhoff & Bilmes, 2014).

Let $\mathrm{sim}(v, x) \in [0, 1]$ be a task-appropriate similarity. In experiments, we use Morgan fingerprints with Tanimoto similarity for SMILES, and $n$-gram shingle Jaccard similarity for text generation tasks. We define the facility-location coverage operator, which reflects how well the buffer represents a sample $v$.

$$\mathrm{msim}(v, S) \triangleq \max_{x \in S} \mathrm{sim}(v, x), \qquad \mathrm{msim}(v, \emptyset) \triangleq 0. \tag{10}$$

The overall submodular objective $f(S)$ is

$$\underbrace{\sum_{x \in S} \mathrm{static}(x)}_{\text{quality / feasibility}} + \lambda_{\mathrm{div}} \underbrace{\sum_{v \in \mathcal{G}} \mathrm{msim}(v, S)}_{\text{facility-location coverage}} + \lambda_{\mathrm{len}} \underbrace{f_{\mathrm{len}}(S)}_{\text{length coverage}}. \tag{11}$$

When weights are nonnegative, each term is monotone submodular and so is their sum. $\mathrm{static}(x)$ denotes a fixed per-sample quality/feasibility term. Validity gating restricts the candidate pool used in the facility-location term, without changing the form of the objective. For length coverage, we discretize samples into bins and use concave-over-counts histogram coverage:

$$f_{\mathrm{len}}(S) \triangleq \sum_{b=1}^{B_{\mathrm{bin}}} \alpha_b\, g\big(c_b(S)\big), \quad g(c) \triangleq \log(1 + c), \tag{12}$$

where $c_b(S)$ is the count in bin $b$ and $\alpha_b \geq 0$ can bias coverage toward desired lengths; implementation details are in Appendix G.4.1.

**Greedy update and efficiency.** For every gradient step, we update the fixed-size buffer by optimizing $\max_{S \subseteq \mathcal{G}, |S| \leq B} f(S)$ using greedy algorithm. With cached similarities and histogram counts, one update costs $O(B|\mathcal{G}|)$. In our settings $B$ and $|\mathcal{G}|$ are small, so the overhead is negligible ($\sim 10\mathrm{ms}$ extra time cost per update).

## 4. Experiments

### 4.1. Tasks

**Scaffold-conditioned SMILES optimization.** We study conditional molecular generation (Li et al., 2024a;b; Xu et al., 2025; Huang et al., 2025; Fei et al., 2026; Huang et al.) where the conditioning input is a fixed molecular scaffold and the model generates a completion by adding fragments. Each terminal sequence is a SMILES string $x \in \mathcal{X}$ that must be chemically valid. We optimize a property objective based on the Estimate of Drug-likeness (Bickerton et al., 2012; Xu et al., 2024). Training aims to learn a GFlowNet sampler whose induced terminal distribution assigns higher probability to high-reward scaffold-consistent molecules while maintaining diversity among valid completions.

**Expr24 arithmetic expression generation.** We also evaluate on a discrete, fully verifiable sparse-reward task: generating an arithmetic expression whose value equals 24. A terminal $x \in \mathcal{X}$ is a variable-length token sequence consisting of digits and operators $\{+, -, \times, \div\}$, evaluated with standard operator precedence. The task score is sparse and exact:

$$R(x) = \mathbb{I}[\mathrm{eval}(x) = 24].$$

This task isolates exploration, credit assignment, termination/length bias, and collapse behavior without domain-specific feasibility issues (e.g., chemical validity).

**CommonGen: Concept-to-Sentence Generation.** To ascertain whether the termination drift identified in the **synthetic** Expr24 task generalizes to **realistic** scenarios, we employ a diagnostic subset of CommonGen (Lin et al., 2020). Unlike the synthetic setting, this task introduces strong pretrained linguistic priors. We treat the reference model as a anchor, allowing us to strictly isolate objective-induced termination drift—deviations from natural stopping priors driven solely by reward maximization.

### 4.2. Compared objectives and replay strategies

We compare three objectives adapted to terminable LLM-GFlowNets: TB , SubTB, and RapTB. Unless otherwise specified, methods employ the standard reward-prioritized replay (RP) from Hu et al. (2024); for ablation purposes, we also include Reward-prioritized replay training (PRT) (Shen et al., 2023). We also introduce our proposed submodular

replay (SubM), which acts to explicitly promote diversity among stored high-reward trajectories.

**Implementation summary.** We fine-tune Llama-3.2-1B with LoRA (rank 16) using AdamW (lr $10^{-4}$). All methods share the same model architecture, tokenizer, decoding constraints, and optimizer configuration. Replay-buffer sizes, SubM refresh settings, and decoding constraints are task-specific and reported in Appendix G.

## 4.3. Metrics

We evaluate (i) feasibility and quality (Acc/Score/BLEU (Papineni et al., 2002)), (ii) diversity (Entropy for text; FPDiv for SMILES), (iii) length/termination calibration, and (iv) prefix-collapse diagnostics (Surv/PefEnt/Top1 versus depth). For Expr24, we additionally report coverage (Unique$_\checkmark$, NormCov) and distributional fidelity (KL/JS) against the full set of exact solutions. Unless stated otherwise, we report means over multiple seeds with 95% confidence intervals; all metric definitions and computation details are provided in Appendix E.

## 4.4. Results on Scaffold-conditioned SMILES Optimization

### 4.4.1. OVERALL PERFORMANCE

Table 1 reports all compared methods on scaffold-conditioned SMILES generation. RL baselines (PPO, GRPO) achieve high validity but collapse to near-zero diversity: PPO concentrates on a single mode, and GRPO achieves Entropy $\leq 0.98$. This confirms the fundamental gap between reward maximization, which concentrates mass on the single best mode, and reward-proportional sampling, which spreads mass across all high-reward modes (Hu et al., 2024).

Among GFlowNet objectives, SubTB suffers from severe validity degradation (Acc 0.328), consistent with the termination drift analyzed in Section 4.5. TB achieves near-perfect validity but concentrates on short sequences (Len 3.065), yielding weaker reward quality and diversity. RapTB+SubM achieves the best quality–diversity trade-off while maintaining high validity. As detailed in Appendix A.1, applying SubM to TB also yields substantial improvements by broadening replay coverage, underscoring SubM's role as a generic component for GFlowNets. We additionally compare against AvgPrefixTB, an alternative prefix-level GFlowNet objective, in Appendix B.2.

### 4.4.2. PER-LENGTH ANALYSIS

Aggregate metrics can be confounded by length shifts (e.g., diversity is more likely to be larger for longer sequences). Figure 2 reports the valid-only length histogram and

*Table 1.* **SMILES generation performance.** Unless specified, metrics are computed on valid samples. `Len` represents average token length. Results are averaged over seeds; see Appendix A.1 for 95% CIs and per-length breakdowns.

| Method | Acc ↑ | Score ↑ | Entropy ↑ | FPDiv ↑ | Len |
|---|---|---|---|---|---|
| PPO | 1.000 | 0.604 | $\approx 0$ | – | – |
| GRPO | 0.997 | 0.661 | 0.98 | – | 10.0 |
| TB | **0.998** | 0.717 | 2.503 | 0.807 | 3.065 |
| SubTB | 0.328 | 0.755 | 2.127 | 0.836 | 8.354 |
| RapTB | 0.996 | 0.740 | 2.448 | 0.860 | 6.142 |
| RapTB + SubM | 0.988 | **0.844** | **2.726** | **0.898** | 7.435 |

length-conditioned score/diversity. RapTB+SubM remains strong across most lengths, whereas TB concentrates on short lengths and degrades in the long-length regime.

### 4.4.3. PREFIX COLLAPSE ANALYSIS

Diverse terminals can still share highly concentrated early prefixes and only branch late, a failure mode we refer to as *prefix collapse*. We therefore compute position-wise prefix statistics on all valid samples. Figure 3 reports prefix diagnostics by prefix length $k$. TB displays rapid survival decay (failure to sustain generation) and sharply increasing top-1 mass at longer prefixes (dominance of a single prefix), indicating severe concentration on a few shared partial trajectories. RapTB sustains higher prefix entropy and lower top-1 mass deeper into the trajectory, consistent with earlier and broader branching among correct samples.

### 4.4.4. LONGER-HORIZON STRESS TEST

Increasing $L_{\max}$ to 15 exposes severe length collapse in TB, which concentrates mass on short trajectories and fails to reach the long-horizon regime (Table 2). While SubTB improves coverage at the cost of validity, RapTB effectively mitigates this length bias, unlocking access to extended trajectories without compromising accuracy. Crucially, RapTB+SubM achieves the most robust performance: it maximizes long-horizon coverage (Frac(11+)), yields the best quality-diversity trade-off (Score & MacroFPDiv), and exhibits the lowest prefix concentration (Top1), demonstrating resistance to mode collapse.

## 4.5. Variable-length Expr24 under sparse rewards

**Enumerable solutions enable controlled diagnostics.** `Expr24` disentangles two factors affecting performance: external coverage (the capacity to discover modes) and internal credit assignment (the fidelity of probability mass allocation). Because the full correct set $\mathcal{Y}^*$ is enumerable, we can (i) control coverage via an oracle replay buffer sampling from it, and (ii) directly compare the learned terminal distribution $\pi$ to a reference $p^*$ using bidirectional KL and

*Table 2.* **Longer-horizon stress test on SMILES ($L_{\max} = 15$).** We report length fractions of valid samples (Length Dist.) and diversity metrics. Prefix diagnostics are averaged over $k$: Surv (Prefix survival), PefEnt (prefix entropy), and Top1 (Top-1 mass). MacroFP macro-averages FPDiv across length bins 0–5, 6–10, and 11+. RapTB+SubM achieves the best balance of long-horizon coverage and diversity. CIs omitted for space (Appendix A.1).

| Method | Performance | | Length Dist. | | | Prefix Diagnostics | | | Diversity | |
| --- | --- | --- | --- | --- | --- | --- | --- | --- | --- | --- |
| | Acc | Score | 0–5↓ | 6–10 | 11+↑ | Surv↑ | PefEnt↑ | Top1↓ | MacroFP↑ | FPDiv↑ |
| TB | **0.999** | 0.716 | 0.858 | 0.129 | 0.013 | 0.207 | 2.99 | 0.303 | 0.653 | 0.813 |
| SubTB | 0.636 | 0.742 | 0.286 | 0.442 | 0.271 | 0.561 | 4.82 | 0.085 | 0.716 | 0.770 |
| RapTB | 0.988 | 0.768 | 0.113 | 0.318 | 0.568 | 0.681 | **5.59** | 0.084 | 0.793 | 0.810 |
| RapTB+SubM | 0.972 | **0.849** | **0.094** | 0.205 | **0.701** | **0.751** | 5.32 | **0.071** | **0.805** | **0.868** |

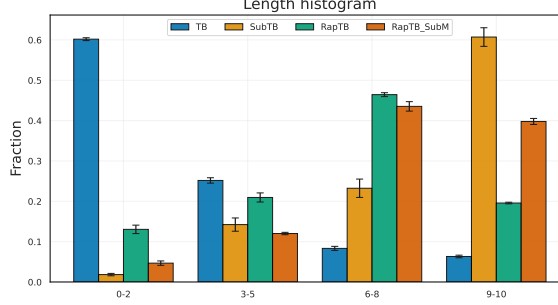

*(a)* Valid-only length histogram.

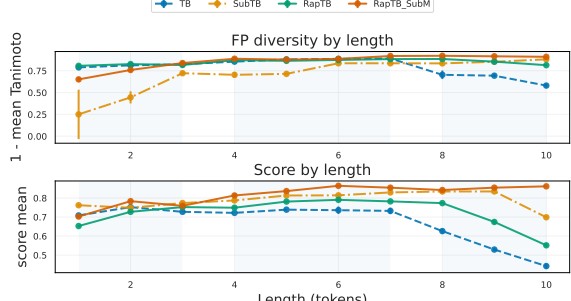

*(b)* Valid-only score and FPDiv versus length.

*Figure 2.* **Length-stratified analysis on SMILES ($L_{\max} = 10$).** (a) Distribution of valid generation lengths. (b) Mean Score and FPDiv conditioned on length.

token-wise JS, where $\text{KL}(p^* \to \pi)$ highlights mode dropping and $\text{KL}(\pi \to p^*)$ reflects over-concentration.

**Expr24 Results.** Table 3 shows that RapTB achieves a superior trade-off between correctness and coverage. Under standard RP, TB suffers from severe mode collapse (Unique$_{\checkmark} \approx 5$), whereas RapTB significantly improves diversity without compromising accuracy. This advantage is amplified by SubM: RapTB+SubM doubles the normalized coverage of the strongest baseline (0.209 vs. 0.100) while maintaining near-perfect accuracy ($>0.99$). Finally, the Oracle setting verifies the objective's effectiveness: RapTB outperforms TB in both accuracy (0.945 vs. 0.919) and distribution matching (lower KL/JS), indicating that RapTB

learns a more precise distribution.

**Complementarity of RapTB and SubM.** Table 3 reveals complementary roles. In coverage-limited regimes, SubM alone can outperform RapTB alone: TB+SubM achieves NormCov 0.100 while RapTB+RP reaches only 0.039. Once coverage is sufficient, RapTB's credit-assignment benefit becomes dominant: with SubM, RapTB doubles Norm-Cov to 0.209. Under Oracle replay, RapTB still improves Acc (0.945 vs. 0.919) and JS (0.013 vs. 0.016), confirming its benefit in distributional fidelity.

**Additional tasks and scaling.** We further evaluate on AMP biological sequence generation (Jain et al., 2022) (amino-acid vocabulary, non-differentiable reward) and scale to 3B, 8B (Llama-3.2), and 32B (Qwen3) on SMILES. On AMP, RapTB+SubM achieves the best performance–diversity–novelty trade-off within 3K steps, while SubTB collapses to maximum length (49.3), inflating its diversity (Appendix B.3). Across model scales, SubTB's termination drift persists at every size (Acc: 0.311/0.391/0.795 at 3B/8B/32B), confirming the failure is structural and architecture-independent. RapTB+SubM consistently achieves the best quality–diversity trade-off at all scales (Appendix C).

**Diagnosis of SubTB's abnormal behaviors.** In the variable-length Expr24 setting, we observe pronounced termination drift: the log-probability of sampled termination, $\log p_{\text{term}}(\tau)$, degrades to extremely negative values (Table 4). This severely impacts the hit rate when the stopping condition is a decision variable. We attribute this phenomenon to the enforcement of numerous arbitrary-start windows, which can be partially satisfied by a global shift in $\log q_\theta(\top \mid s_{0:\tau})$ (see analysis in Appendix F.6). To investigate whether termination drift is the dominant failure mode, we employ ROOTSUBTBLOGZ, which restricts SubTB windows to be rooted and reintroduces a learnable global normalizer $Z_\theta$. As shown in Table 4, this modification mitigates termination drift and restores accuracy to nearly 100%.

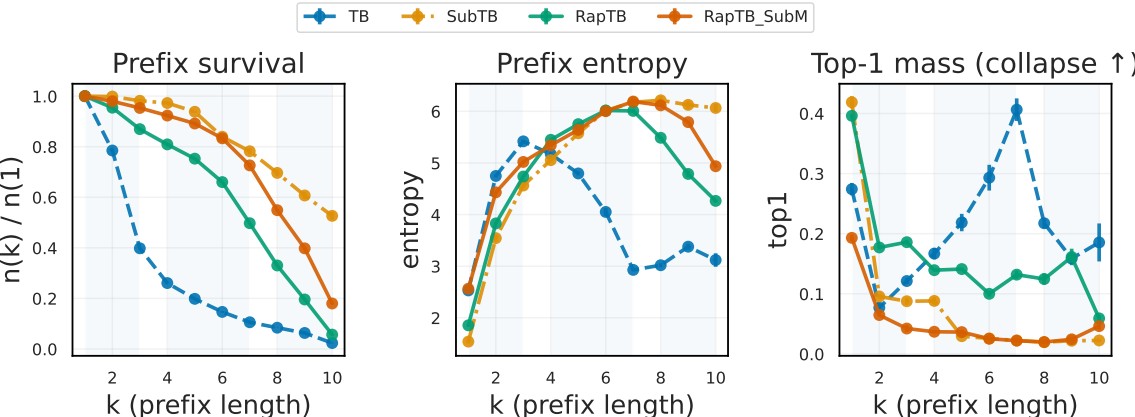

*Figure 3.* **Prefix-collapse diagnostics on SMILES ($L_{\max} = 10$).** Metrics vs. prefix length $k$ computed on correct samples: prefix survival (fraction of samples reaching length $k$), prefix entropy (diversity of prefix), and top-1 mass (frequency of the most common prefix).

*Table 3.* **Expr24 results under different replay schemes.** Results are averaged over different random seeds; 95% confidence intervals are reported in Appendix A.2.

| Replay | Objective | Diversity | | Quality | Distributional Fidelity | | |
|---|---|---|---|---|---|---|---|
| | | Unique✓↑ | NormCov↑ | Acc↑ | KL($\pi \to p^*$)↓ | KL($p^* \to \pi$)↓ | JS$_{\text{tok}}$↓ |
| N/A | PPO | 1 | 0.000 | 0.003 | – | – | – |
| | GRPO | 1 | 0.000 | 0.002 | – | – | – |
| PRT | TB | 103.7 | 0.016 | 0.999 | 1.105 | 7.803 | 0.292 |
| | SubTB | 292.0 | 0.046 | 0.311 | 0.424 | 0.672 | 0.107 |
| | RapTB | 129.3 | 0.020 | 0.992 | 0.908 | 5.538 | 0.230 |
| RP | TB | 5.3 | 0.001 | 1.000 | 1.297 | 11.403 | 0.339 |
| | SubTB | 324.7 | 0.051 | 0.229 | 0.455 | 0.865 | 0.109 |
| | RapTB | 246.7 | 0.039 | 0.991 | 0.561 | 4.480 | 0.147 |
| SubM | TB | 642.0 | 0.100 | 0.996 | 0.182 | 0.441 | 0.049 |
| | SubTB | 331.3 | 0.052 | 0.061 | 0.149 | 0.286 | 0.040 |
| | RapTB | 1337.3 | 0.209 | 0.994 | 0.169 | 0.623 | 0.048 |
| Oracle | TB | 5198.0 | 0.812 | 0.919 | 0.062 | 0.066 | 0.016 |
| | SubTB | 35.7 | 0.006 | 0.006 | 0.266 | 1.491 | 0.071 |
| | RapTB | 5220.7 | 0.816 | 0.945 | 0.052 | 0.056 | 0.013 |

## 4.6. Results on CommonGen

Table 5 confirms that optimization pressure can indeed override linguistic priors. Despite the model's natural tendency to stop, SubTB deviates catastrophically from the anchor, saturating length (20.00) by suppressing stopping logits ($\Delta \log p_{\text{term}} \approx -28.32$). Conversely, RapTB maintains calibration ($\approx -0.94$). Notably, RapTB+SubM achieves superior BLEU (33.23) at natural lengths (11.83), demonstrating robust performance.

## 4.7. Ablations.

Table 6 ablates the key design choices of RapTB and SubM on SMILES. Removing reward absorption degrades both score and diversity, suggesting that suffix evidence provides

useful prefix credit. Using only max or only soft backups increases score but reduces diversity, while the mixed backup improves the balance. Detaching termination gradients in the auxiliary branch is also important. Without it, the model collapses to very short sequences (Len 3.40) and the score drops. SubM components are complementary. Reward-only improves score, diversity-only improves FPDiv, and length-only increases long-horizon coverage. Combining them yields the strongest overall trade-off.

Additional sensitivity analysis over $(\beta, \rho, \eta, k_{\min})$ across both tasks is provided in Appendix D; across all 18 configurations tested, no catastrophic failure, length collapse, or termination drift is observed.

*Table 4.* **Termination/length calibration diagnostic on Expr24.** More negative values indicate overly suppressed termination. $\log p_{term}(\tau)$ is the termination log-probability at the sampled stop step, computed from the model's raw $q_\theta(\top \mid s_{0:\tau})$.

| Method | Acc↑ | NormCov↑ | $\log p_{term}(\tau)$ | $\log Z$ |
|---|---|---|---|---|
| **RP Replay** | | | | |
| TB | 0.999 | 0.001 | -0.000 | 0.063 |
| SubTB | 0.229 | 0.051 | -79.638 | – |
| RapTB | 0.991 | 0.039 | -0.065 | 0.062 |
| RootSubTBLogZ | 0.999 | 0.023 | -0.068 | 0.062 |
| **Oracle Replay** | | | | |
| TB | 0.922 | 0.813 | -0.436 | 0.037 |
| SubTB | 0.006 | 0.006 | -86.415 | – |
| RapTB | 0.945 | 0.816 | -0.644 | 0.038 |
| RootSubTBLogZ | 0.885 | 0.727 | -1.432 | 0.036 |

*Table 5.* **CommonGen performance.** $\Delta \log p_{term}$ measures the deviation of the learned policy's termination logits from the pre-trained reference anchor.

| Method | Entropy↑ | BLEU-4↑ | Len | $\Delta \log p_{term}$ |
|---|---|---|---|---|
| TB | 2.966 | 5.95 | 13.86 | -1.34 |
| SubTB | 3.719 | 24.39 | 20.00 | -28.32 |
| RapTB | 3.933 | 11.75 | 15.63 | -0.94 |
| RapTB+SubM | **4.102** | **33.23** | 11.83 | 4.89 |

## 5. Discussion and Practical Guidance

**Mitigating Replay-Induced Collapse.** Standard reward-prioritized replay often induces "rich-get-richer" dynamics (Shen et al., 2023), where the training distribution collapses onto a narrow set of repeated high-reward modes. SubM explicitly counters this by enforcing structural diversity within the buffer. This prevents near-duplicate dominance and ensures the policy learns from a broad, representative landscape rather than degenerate subsets.

**Credit Assignment and Consistency.** A key limitation of SubTB is that enforcing constraints on arbitrary windows creates conflicting boundary conditions, effectively hardening optimization and destabilizing termination. RapTB resolves this by grounding dense supervision to rooted prefixes, ensuring all partial trajectory updates remain consistent with the global partition function $Z$. A future direction is adaptive subtrajectory selection, where the model learns to identify and prioritize essential substructures online.

## 6. Conclusion

We studied mode collapse in terminable LLM-GFlowNets and identified two coupled and reproducible failure modes: prefix collapse and length bias. To address unstable credit assignment without inducing termination drift, we proposed RapTB, which augments terminal Trajectory Balance with

*Table 6.* **Ablation study on SMILES generation.**

| Variant | Score ↑ | FPDiv ↑ | Entropy ↑ | Len |
|---|---|---|---|---|
| RapTB | 0.740 | 0.860 | 2.448 | 6.142 |
| RapTB w/o reward absorb | 0.716 | 0.805 | 2.031 | 5.296 |
| RapTB absorb max-only | 0.821 | 0.775 | 1.716 | 7.431 |
| RapTB absorb soft-only | 0.819 | 0.748 | 1.516 | 7.710 |
| RapTB + SubM (Full) | 0.844 | 0.898 | **2.726** | 7.435 |
| + length-only SubM | 0.773 | 0.885 | 1.914 | 6.569 |
| + diversity-only SubM | 0.741 | **0.942** | 2.602 | 5.459 |
| + reward-only SubM | **0.878** | 0.884 | 1.876 | 8.122 |
| RapTB w/o detach $p_{term}$ | 0.714 | 0.892 | 2.072 | 3.403 |

rooted prefix constraints and suffix-absorbed reward shaping. To address replay-induced distribution shift and limited external coverage, we introduced SubM, a submodular replay refresh strategy that balances reward with diversity and length support. Across tasks, RapTB+SubM improves long-horizon stability and coverage, yielding better reward–diversity trade-offs and substantially reduced prefix collapse. Scaling experiments up to 32B parameters across two architecture families and an additional biological sequence task confirm that these improvements are consistent across domains and model scales (Appendices B.3–C). We hope these results encourage future objectives that explicitly couple coverage-aware replay with selective, adaptively weighted subtrajectory learning for robust autoregressive GFlowNet training.

## Impact Statement

This paper advances learning methods for sampling diverse high-quality solutions from various reward distributions. In molecular generation, improved exploration and property optimization may accelerate candidate discovery, but generated molecules are only hypotheses and require downstream validation. We do not foresee direct negative societal impacts from the method itself beyond the general risks of misuse of generative models; appropriate safeguards and domain expert oversight remain necessary.

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

# A. Additional Results

## A.1. SMILES

Molecule generation is a central task in the application of AI to chemistry, encompassing two main challenges. The first is molecular representation: without an effective representation, the vast chemical space becomes difficult to navigate. In this work, we adopt SMILES as our primary representation. Nevertheless, a variety of alternative approaches have demonstrated strong potential. For instance, quantum chemistry-based and geometric descriptors offer physically grounded encodings of molecular structure (Shi et al., 2026), while chirality-aware representations have proven essential for distinguishing stereoisomers in downstream tasks (Peng et al., 2025). Reaction-targeted descriptors, such as substrate-aware features automatically extracted for performance prediction, provide another powerful lens through which to characterize molecular reactivity (Yu et al., 2025). More recently, round-trip molecule-text alignment frameworks have shown that bridging modalities can yield richer and more transferable representations (Chen et al., 2026). The second challenge concerns the choice of generative model. Here, we employ autoregressive modeling for sequential molecule construction. Diffusion-based methods have also emerged as a powerful paradigm in this space, with recent work demonstrating their effectiveness even in data-scarce regimes such as atropisomer generation through multi-task pretraining and classifier guidance (Chen et al., 2025).

### A.1.1. PER-LENGTH SMILES PERFORMANCE.

Tables 8–9 report valid-only metrics stratified by terminal length $L$, where the largest differences concentrate in the longest bins ($L \geq 8$) under rewards. Submodular Replay (SubM) substantially improves length-wise coverage for TB, largely eliminating its long-length degradation: for $L = 8$–$10$, TB's Score increases from $0.626/0.529/0.442$ to $0.861/0.880/0.875$, with diversity recovering simultaneously (Entropy $\approx 2$ and FPDiv $\approx 0.91$). Under this stronger coverage regime, the gap between RapTB and TB naturally narrows (a ceiling effect), yet RapTB+SubM remains competitive across lengths and tends to allocate more mass to the longest bin (higher Frac/Count at $L \geq 9$) while maintaining high quality, consistent with RapTB improving the learning signal for suffix credit assignment whereas SubM primarily improves external replay coverage across length/diversity/reward. In contrast, SubTB (with or without SubM) exhibits pronounced length skew, placing disproportionately large mass at $L = 9$–$10$ alongside a substantial validity drop, consistent with the abnormal termination dynamics discussed in the main text.

**All-length averaged summary.** Table 7 aggregates metrics across all generated lengths ($L_{\max} = 10$). SubM dramatically changes the effective training distribution of TB: the median length shifts from $2.15$ to $7.23$ and Frac[0–2] decreases from $0.601$ to $0.143$, while Frac[9–10] increases from $0.064$ to $0.323$, yielding a large gain in valid-only average Score ($0.717 \rightarrow 0.842$) and diversity ($2.503 \rightarrow 2.775$). With SubM, RapTB+SubM and TB+SubM achieve very similar averaged Score ($0.844$ vs. $0.842$), but RapTB+SubM allocates significantly more probability mass to the longest bin (Frac[9–10] $0.402$ vs. $0.323$), at a small but noticeable cost in overall validity (Acc $0.988$ vs. $0.996$). Finally, for SubTB variants, the reported valid-only averages should be interpreted cautiously due to low Acc ($\approx 0.30$): SubM further concentrates length mass at $L = 9$–$10$ (Frac $> 0.90$) without resolving the validity collapse.

| Method | Acc | Score | Entropy | FPDiv | Len$_\mu$ | Len$_{50}$ | Len$_{90}$ | Frac[0–2] | Frac[3–5] | Frac[6–8] | Frac[9–10] |
|---|---|---|---|---|---|---|---|---|---|---|---|
| TB | 0.998±0.001 | 0.717±0.001 | 2.503±0.026 | 0.807±0.003 | 3.06±0.02 | 2.15±0.03 | 6.42±0.13 | 0.601±0.004 | 0.252±0.007 | 0.084±0.005 | 0.064±0.004 |
| TB+SubM | 0.996±0.000 | 0.842±0.001 | 2.775±0.002 | 0.889±0.002 | 6.56±0.09 | 7.23±0.07 | 9.84±0.06 | 0.143±0.006 | 0.186±0.015 | 0.348±0.013 | 0.323±0.008 |
| RapTB | 0.996±0.001 | 0.740±0.004 | 2.448±0.017 | 0.860±0.001 | 6.14±0.03 | 6.58±0.03 | 9.01±0.04 | 0.134±0.007 | 0.204±0.009 | 0.466±0.017 | 0.197±0.015 |
| RapTB+SubM | 0.988±0.003 | 0.844±0.001 | 2.726±0.017 | 0.898±0.001 | 7.44±0.05 | 7.91±0.03 | 9.84±0.02 | 0.046±0.005 | 0.119±0.003 | 0.433±0.012 | 0.402±0.007 |
| SubTB | 0.328±0.016 | 0.755±0.004 | 2.127±0.003 | 0.836±0.003 | 8.35±0.06 | 9.22±0.05 | 9.97±0.03 | 0.006±0.001 | 0.047±0.005 | 0.076±0.005 | 0.871±0.001 |
| SubTB+SubM | 0.298±0.006 | 0.736±0.005 | 2.165±0.006 | 0.851±0.002 | 8.73±0.06 | 9.59±0.08 | 9.99±0.01 | 0.005±0.001 | 0.029±0.003 | 0.057±0.001 | 0.909±0.003 |

**Table 7.** All-length averaged SMILES performance and induced length distribution ($L_{\max} = 10$; mean±95% CI over 6 runs). Acc is computed over all samples; Score/Entropy/FPDiv are computed on valid samples only; Frac[·] is computed over all samples.

| $L$ | TB | | | | SubTB | | | | RapTB | | | | TB+SubM | | | | SubTB+SubM | | | | RapTB+SubM | | | |
|---|---|---|---|---|---|---|---|---|---|---|---|---|---|---|---|---|---|---|---|---|---|---|---|---|
| | Acc | Score | Frac | Count | Acc | Score | Frac | Count | Acc | Score | Frac | Count | Acc | Score | Frac | Count | Acc | Score | Frac | Count | Acc | Score | Frac | Count |
| 1 | 1±0 | 0.708±0.004 | 0.215±0.007 | 687±22.7 | 1±0 | 0.762±0.007 | 0.002±0.001 | 1.7±0.7 | 1±0 | 0.651±0.004 | 0.042±0.002 | 133±4.9 | 1±0 | 0.712±0.007 | 0.055±0.007 | 175±22.6 | 1±0 | 0.766±0.001 | 0.004±0.003 | 3.7±2.6 | 1±0 | 0.702±0.006 | 0.02±0.002 | 64.3±7.5 |
| 2 | 0.999±0.001 | 0.752±0.002 | 0.387±0.008 | 1235.7±28.1 | 1±0 | 0.748±0.003 | 0.017±0.003 | 17.7±3.5 | 0.999±0.002 | 0.725±0.004 | 0.092±0.006 | 294.7±17.4 | 1±0 | 0.79±0 | 0.058±0.003 | 282±7.9 | 1±0 | 0.75±0.014 | 0.012±0.004 | 11.7±4 | 1±0 | 0.783±0.005 | 0.025±0.003 | 83.3±10.8 |
| 3 | 0.998±0.002 | 0.727±0.002 | 0.138±0.003 | 439.3±10.5 | 1±0 | 0.774±0.01 | 0.009±0.004 | 9.3±4 | 1±0 | 0.754±0.001 | 0.056±0.002 | 177±6.8 | 1±0 | 0.787±0.003 | 0.058±0.008 | 184.3±26.7 | 1±0 | 0.759±0.006 | 0.024±0.003 | 23.3±2.4 | 0.996±0.008 | 0.759±0.006 | 0.031±0.006 | 95±19.3 |
| 4 | 0.998±0.003 | 0.722±0.001 | 0.062±0.002 | 199.3±6.8 | 1±0 | 0.787±0.003 | 0.035±0.003 | 36.3±2.4 | 0.995±0.006 | 0.748±0.003 | 0.058±0.004 | 185.7±13.6 | 0.997±0.001 | 0.793±0.003 | 0.046±0.005 | 145.3±16.8 | 1±0 | 0.789±0.006 | 0.024±0.003 | 23.3±2.4 | 1±0 | 0.813±0.012 | 0.031±0.002 | 99.3±5.3 |
| 5 | 0.996±0.008 | 0.739±0.003 | 0.052±0.002 | 165.3±5.8 | 1±0 | 0.813±0.001 | 0.099±0.011 | 103.7±9.9 | 0.998±0.004 | 0.78±0.009 | 0.09±0.005 | 288.3±14.6 | 0.997±0.002 | 0.851±0.004 | 0.083±0.004 | 263.7±13.6 | 0.995±0.006 | 0.836±0.003 | 0.059±0.004 | 185.3±12.1 | | | | |
| 6 | 0.995±0.006 | 0.736±0.019 | 0.041±0.005 | 131±17.4 | 1±0 | 0.814±0.001 | 0.057±0.015 | 60±13.8 | 0.997±0.001 | 0.79±0.004 | 0.155±0.007 | 494.7±23.1 | 0.998±0.002 | 0.844±0.005 | 0.094±0.004 | 301±13.1 | 1±0 | 0.819±0.001 | 0.051±0.007 | 48.7±6.2 | 0.995±0.005 | 0.864±0.003 | 0.107±0.008 | 338.3±24.9 |
| 7 | 0.995±0.009 | 0.732±0.017 | 0.021±0.002 | 68.7±7.4 | 1±0 | 0.829±0.003 | 0.086±0.009 | 90±9.8 | 0.997±0.002 | 0.785±0.004 | 0.17±0.003 | 540.3±8.2 | 0.997±0.002 | 0.86±0.004 | 0.122±0.006 | 387.7±18.9 | 1±0 | 0.831±0.005 | 0.064±0.002 | 60.7±3.5 | 0.993±0.003 | 0.854±0.002 | 0.177±0.0099 | 560±26.8 |
| 8 | 0.995±0.011 | 0.626±0.012 | 0.021±0.002 | 66.3±5.7 | 1±0 | 0.834±0.002 | 0.089±0.007 | 93.7±4.7 | 0.994±0.005 | 0.775±0.006 | 0.141±0.009 | 449±29.3 | 0.998±0.002 | 0.861±0.001 | 0.133±0.007 | 422.7±23.6 | 1±0 | 0.837±0.005 | 0.077±0.006 | 73±5.2 | 0.992±0.006 | 0.842±0.001 | 0.151±0.015 | 477.3±46.2 |
| 9 | 1±0 | 0.529±0.008 | 0.04±0.002 | 127.3±5.1 | 1±0 | 0.834±0.002 | 0.081±0.002 | 84.7±4 | 0.996±0.001 | 0.666±0.005 | 0.141±0.012 | 448.3±37.9 | 0.998±0.003 | 0.88±0.003 | 0.139±0.004 | 443.3±13.6 | 1±0 | 0.838±0.005 | 0.051±0.01 | 48.3±10.3 | 0.997±0.003 | 0.854±0.002 | 0.218±0.007 | 689.3±24.3 |
| 10 | 0.979±0.028 | 0.442±0.013 | 0.023±0.003 | 74.7±8.5 | 0.205±0.019 | 0.699±0.004 | 0.527±0.024 | 554±52 | 0.98±0.019 | 0.561±0.01 | 0.055±0.006 | 176±19.8 | 0.988±0.002 | 0.875±0.002 | 0.183±0.012 | 583.3±38.4 | 0.215±0.007 | 0.69±0.009 | 0.643±0.016 | 614.3±22.5 | 0.956±0.007 | 0.861±0.002 | 0.18±0.003 | 568.3±7.9 |

**Table 8.** Per-length valid-only core metrics of SMILES generation (mean±95% CI, $L_{\max} = 10$).

| L | TB | | SubTB | | RapTB | | TB+SubM | | SubTB+SubM | | RapTB+SubM | |
|---|---|---|---|---|---|---|---|---|---|---|---|---|
| | Entropy | FPDiv | Entropy | FPDiv | Entropy | FPDiv | Entropy | FPDiv | Entropy | FPDiv | Entropy | FPDiv |
| 1 | 2.39±0.07 | 0.789±0.012 | 0.23±0.45 | 0.25±0.283 | 2.28±0.11 | 0.824±0.005 | 1.35±0.09 | 0.628±0.014 | 0.17±0.33 | 0.2±0.226 | 1.32±0.08 | 0.651±0.027 |
| 2 | 2.54±0.01 | 0.812±0.005 | 1.05±0.33 | 0.445±0.069 | 2.33±0.03 | 0.827±0.003 | 2.12±0.07 | 0.84±0.004 | 0.99±0.09 | 0.435±0.056 | 1.82±0.13 | 0.759±0.022 |
| 3 | 2.44±0.02 | 0.825±0.001 | 1.1±0.17 | 0.722±0.022 | 1.99±0.03 | 0.818±0.003 | 2.21±0.07 | 0.877±0.002 | 0.87±0.43 | 0.604±0.091 | 1.76±0.04 | 0.838±0.002 |
| 4 | 2.46±0.05 | 0.854±0.002 | 1.35±0.14 | 0.703±0.03 | 2.02±0.06 | 0.867±0.005 | 2.39±0.01 | 0.909±0.005 | 1.31±0.36 | 0.739±0.05 | 2.19±0.07 | 0.888±0.007 |
| 5 | 2.41±0.02 | 0.88±0.005 | 1.19±0.11 | 0.714±0.01 | 2.21±0.04 | 0.859±0.003 | 2.48±0.04 | 0.906±0.004 | 1.15±0.05 | 0.721±0.004 | 2.21±0.12 | 0.877±0.015 |
| 6 | 2.42±0.08 | 0.886±0.002 | 1.7±0.06 | 0.836±0.008 | 2.15±0.04 | 0.88±0.005 | 2.44±0.05 | 0.92±0.001 | 1.69±0.04 | 0.845±0.001 | 2.37±0.03 | 0.886±0.007 |
| 7 | 2.35±0.02 | 0.891±0.003 | 1.5±0.05 | 0.836±0.007 | 2.18±0.02 | 0.884±0.001 | 2.49±0.01 | 0.926±0.001 | 1.68±0.02 | 0.864±0.008 | 2.47±0.03 | 0.919±0.003 |
| 8 | 1.61±0.23 | 0.703±0.049 | 1.62±0.03 | 0.834±0.012 | 2.17±0.02 | 0.882±0.001 | 2.54±0.01 | 0.918±0.001 | 1.55±0.06 | 0.829±0.01 | 2.59±0.03 | 0.921±0.001 |
| 9 | 1.35±0.05 | 0.693±0.016 | 1.71±0.07 | 0.853±0.002 | 1.98±0.04 | 0.848±0.006 | 2.36±0.02 | 0.909±0.001 | 1.67±0.05 | 0.857±0.008 | 2.28±0.05 | 0.916±0.001 |
| 10 | 1.06±0.11 | 0.58±0.023 | 2.16±0.04 | 0.881±0 | 1.75±0.03 | 0.815±0.004 | 2.08±0.02 | 0.906±0.001 | 2.18±0.01 | 0.884±0.003 | 2.16±0.02 | 0.91±0 |

**Table 9.** Per-length valid-only diversity metrics of SMILES generation(mean±95% CI, $L_{\max} = 10$).

| L | TB | | | | SubTB | | | | RapTB | | | | TB+SubM | | | | SubTB+SubM | | | | RapTB+SubM | | | |
|---|---|---|---|---|---|---|---|---|---|---|---|---|---|---|---|---|---|---|---|---|---|---|---|---|
| | UniqStr | UniqMol | UniqRateStr | UniqRateMol | UniqStr | UniqMol | UniqRateStr | UniqRateMol | UniqStr | UniqMol | UniqRateStr | UniqRateMol | UniqStr | UniqMol | UniqRateStr | UniqRateMol | UniqStr | UniqMol | UniqRateStr | UniqRateMol | UniqStr | UniqMol | UniqRateStr | UniqRateMol |
| 1 | 29.3±1.1 | 25.3±1.5 | 0±0 | 0±0 | 1.3±0.4 | 1.3±0.4 | 0.8±0.2 | 0.8±0.2 | 17.3±0.8 | 15.7±0.4 | 0.1±0 | 0.1±0 | 13.7±0.8 | 13.7±0.8 | 0.1±0 | 0.1±0 | 1.3±0.4 | 1.3±0.4 | 0.5±0.3 | 0.5±0.3 | 8.3±0.8 | 8.3±0.8 | 0.1±0 | 0.1±0 |
| 2 | 184.7±2.1 | 115.7±3 | 0.1±0 | 0.2±0 | 6.3±0.8 | 6±0.7 | 0.4±0 | 0.3±0 | 77±1.9 | 71±2.5 | 0.3±0 | 0.2±0 | 60.3±3 | 59.7±2.9 | 0.2±0 | 0.2±0 | 5±0.7 | 5±0.7 | 0.4±0 | 0.4±0 | 27.3±1.5 | 27.3±1.5 | 0.3±0 | 0.3±0 |
| 3 | 224±7.5 | 215.7±7.3 | 0.5±0 | 0.5±0 | 6.3±1.7 | 6.3±1.7 | 0.7±0.1 | 0.7±0.1 | 78.3±2.2 | 76.3±2.2 | 0.4±0 | 0.4±0 | 52.3±5.8 | 52.3±5.8 | 0.3±0 | 0.3±0 | 4.7±1.8 | 4.7±1.8 | 0.6±0.1 | 0.6±0.1 | 27.7±3 | 27.7±3 | 0.3±0 | 0.3±0 |
| 4 | 166±4 | 164±3.6 | 0.8±0 | 0.8±0 | 17±1.2 | 16.7±0.8 | 0.5±0 | 0.5±0 | 111±3.1 | 107.3±3 | 0.6±0 | 0.6±0 | 58±0.7 | 57.7±1.1 | 0.4±0 | 0.4±0 | 14.7±2.2 | 13.7±1.8 | 0.6±0.1 | 0.6±0.1 | 41±6.1 | 41±6.1 | 0.4±0.1 | 0.4±0.1 |
| 5 | 156±5.9 | 153.3±5.8 | 0.9±0 | 0.9±0 | 41.3±3.7 | 38.3±3.9 | 0.4±0 | 0.4±0 | 187±7.6 | 181.7±7.9 | 0.6±0 | 0.6±0 | 101.3±3.5 | 100.7±3.3 | 0.4±0 | 0.4±0 | 31.3±4.2 | 28.3±3.5 | 0.5±0 | 0.4±0 | 74±8.1 | 73.7±7.9 | 0.4±0 | 0.4±0 |
| 6 | 129.7±10.2 | 128.7±9.5 | 1±0 | 1±0 | 43.3±6.6 | 40±6.6 | 0.7±0 | 0.7±0 | 332.3±4.2 | 322±5 | 0.7±0 | 0.7±0 | 130.3±12.8 | 130±12.4 | 0.4±0 | 0.4±0 | 39±5.4 | 37±5.2 | 0.8±0 | 0.8±0 | 123.3±6 | 123.3±6 | 0.4±0 | 0.4±0 |
| 7 | 68±4.7 | 68±4.7 | 1±0 | 1±0 | 60±3.1 | 55±2.5 | 0.7±0 | 0.6±0 | 448.7±7.2 | 411.7±3.3 | 0.8±0 | 0.8±0 | 157±8.7 | 156.3±8.4 | 0.4±0 | 0.4±0 | 52.3±3 | 50.3±1.8 | 0.9±0 | 0.8±0 | 235±6.1 | 220.7±0.4 | 0.4±0 | 0.4±0 |
| 8 | 35±5 | 35±5 | 0.5±0.1 | 0.5±0.1 | 73.3±8.1 | 69.3±7 | 0.8±0.1 | 0.7±0.1 | 409.7±15.1 | 400±5.4 | 0.9±0 | 0.9±0 | 163±6.8 | 162.3±6.5 | 0.4±0 | 0.4±0 | 58.7±3.2 | 56.3±3.2 | 0.8±0 | 0.8±0 | 299±13.6 | 298.3±13.7 | 0.6±0 | 0.6±0 |
| 9 | 38.7±1.1 | 38.7±1.1 | 0.3±0 | 0.3±0 | 76±6.2 | 72±5.2 | 0.9±0.1 | 0.8±0 | 246.3±6 | 244.3±5.4 | 0.6±0 | 0.6±0 | 159.7±4.3 | 159.3±4.7 | 0.4±0 | 0.4±0 | 45.7±5.4 | 43.3±5.5 | 0.9±0 | 0.9±0 | 353.7±5 | 291.7±5.3 | 0.5±0 | 0.5±0 |
| 10 | 33±4.4 | 33±4.4 | 0.4±0 | 0.4±0 | 482.3±31.8 | 435.3±14.1 | 0.9±0 | 0.9±0 | 96.3±3.3 | 95.7±2.9 | 0.5±0 | 0.5±0 | 156.3±4.6 | 145.3±3.7 | 0.3±0 | 0.3±0 | 558±8.7 | 467.3±4.2 | 0.9±0 | 0.9±0 | 242±5.9 | 225.3±2.9 | 0.4±0 | 0.4±0 |

**Table 10.** Per-length valid-only uniqueness metrics of SMILES generation (mean±95% CI, $L_{\max} = 10$).

## A.1.2. PER-LENGTH PREFIX COLLAPSE ANALYSIS OF SMILES GENERATION.

Table 11 reports prefix distributions at depth $k$ (mean±95% CI), where Survival should be read jointly with concentration metrics (PefEnt/Eff/Top1/UniqueRate). Without SubM, TB exhibits rapid attrition (e.g., Survival drops to 0.105 at $k$=7 and 0.023 at $k$=10) and increased concentration among the remaining deep prefixes (Top1 reaches 0.406 at $k$=7), consistent with prefix collapse. RapTB substantially improves Survival at larger $k$ (e.g., 0.497 at $k$=7) while reducing deep-prefix concentration (Top1 0.132 at $k$=7), indicating more sustained branching beyond early decisions. Enabling SubM (Table 12) further alleviates collapse for both TB and RapTB by increasing deep-prefix diversity (higher PefEnt/Eff, lower Top1) and improving Survival at large $k$ (e.g., TB: 0.023→0.183 and RapTB: 0.055→0.180 at $k$=10).

| k | TB | | | | | SubTB | | | | | RapTB | | | | |
|---|---|---|---|---|---|---|---|---|---|---|---|---|---|---|---|
| | Survival | PefEnt | Eff | Top1 | UniqueRate | Survival | PefEnt | Eff | Top1 | UniqueRate | Survival | PefEnt | Eff | Top1 | UniqueRate |
| 1 | 1±0 | 2.531±0.022 | 12.57±0.28 | 0.274±0.01 | 0.013±0.001 | 1±0 | 1.539±0.036 | 4.66±0.17 | 0.419±0.01 | 0.017±0.001 | 1±0 | 1.841±0.025 | 6.3±0.15 | 0.4±0.004 | 0.01±0 |
| 2 | 0.785±0.005 | 4.748±0.019 | 115.34±2.25 | 0.077±0.001 | 0.135±0.002 | 0.998±0 | 3.544±0.063 | 34.69±2.15 | 0.096±0.004 | 0.096±0.002 | 0.958±0.001 | 3.838±0.023 | 46.46±1.05 | 0.174±0.005 | 0.076±0.002 |
| 3 | 0.398±0.002 | 5.418±0.027 | 225.52±6.1 | 0.121±0.005 | 0.45±0.005 | 0.982±0.002 | 4.563±0.044 | 95.34±4.19 | 0.088±0.006 | 0.24±0.003 | 0.866±0.005 | 4.717±0.031 | 111.95±3.52 | 0.184±0.005 | 0.215±0.005 |
| 4 | 0.261±0.002 | 5.17±0.038 | 176.02±6.81 | 0.167±0.009 | 0.566±0.012 | 0.973±0.004 | 5.054±0.035 | 156.78±5.48 | 0.088±0.006 | 0.355±0.003 | 0.81±0.006 | 5.45±0.047 | 233.1±11.25 | 0.135±0.007 | 0.357±0.01 |
| 5 | 0.198±0.002 | 4.796±0.041 | 121.2±4.96 | 0.219±0.014 | 0.599±0.009 | 0.938±0.005 | 5.575±0.048 | 264.06±12.39 | 0.03±0.003 | 0.471±0.004 | 0.752±0.004 | 5.747±0.051 | 313.88±16.34 | 0.138±0.007 | 0.474±0.01 |
| 6 | 0.146±0.003 | 4.051±0.075 | 57.65±4.37 | 0.293±0.021 | 0.533±0.007 | 0.839±0.011 | 6.012±0.046 | 408.87±18.66 | 0.026±0.001 | 0.638±0.006 | 0.662±0.006 | 6.027±0.044 | 414.97±17.93 | 0.102±0.004 | 0.575±0.012 |
| 7 | 0.105±0.002 | 2.928±0.02 | 18.7±0.37 | 0.406±0.019 | 0.363±0.008 | 0.782±0.012 | 6.175±0.07 | 482.25±33.04 | 0.022±0.002 | 0.738±0.01 | 0.506±0.004 | 6.011±0.05 | 408.6±20.46 | 0.132±0.004 | 0.637±0.01 |
| 8 | 0.084±0.001 | 3.018±0.002 | 20.48±0.84 | 0.217±0.008 | 0.246±0.012 | 0.696±0.011 | 6.213±0.081 | 501.27±39.04 | 0.22±0 | 0.803±0.016 | 0.337±0.003 | 5.492±0.072 | 243.57±17.66 | 0.126±0.011 | 0.6±0.013 |
| 9 | 0.063±0.002 | 3.378±0.059 | 29.37±1.76 | 0.159±0.008 | 0.27±0.009 | 0.607±0.015 | 6.124±0.076 | 458.44±33.86 | 0.022±0.001 | 0.83±0.015 | 0.196±0.009 | 4.774±0.031 | 118.44±3.61 | 0.16±0.01 | 0.474±0.012 |
| 10 | 0.023±0.002 | 3.119±0.131 | 22.88±3.02 | 0.185±0.032 | 0.442±0.045 | 0.527±0.015 | 6.065±0.076 | 431.97±31.82 | 0.022±0.003 | 0.871±0.01 | 0.055±0.004 | 4.276±0.022 | 71.97±1.57 | 0.049±0.007 | 0.549±0.023 |

**Table 11.** Prefix statistics by depth. Mean±95% CI.

| k | TB+SubM | | | | | SubTB+SubM | | | | | RapTB+SubM | | | | |
|---|---|---|---|---|---|---|---|---|---|---|---|---|---|---|---|
| | Survival | PefEnt | Eff | Top1 | UniqueRate | Survival | PefEnt | Eff | Top1 | UniqueRate | Survival | PefEnt | Eff | Top1 | UniqueRate |
| 1 | 1±0 | 2.715±0.017 | 15.11±0.25 | 0.165±0.006 | 0.011±0 | 1±0 | 1.559±0.037 | 4.76±0.18 | 0.394±0.01 | 0.017±0.001 | 1±0 | 2.563±0.004 | 12.98±0.05 | 0.193±0.005 | 0.009±0 |
| 2 | 0.945±0.004 | 4.778±0.004 | 118.92±0.46 | 0.052±0.003 | 0.094±0.001 | 0.996±0.002 | 3.669±0.041 | 39.26±1.58 | 0.094±0.005 | 0.108±0.004 | 0.98±0.001 | 4.43±0.015 | 83.96±1.27 | 0.065±0.001 | 0.076±0.001 |
| 3 | 0.857±0.004 | 5.224±0.01 | 185.6±1.81 | 0.033±0.002 | 0.164±0.005 | 0.984±0.002 | 4.723±0.024 | 112.55±2.69 | 0.089±0.004 | 0.278±0.008 | 0.953±0.003 | 5.019±0.022 | 151.34±3.33 | 0.042±0.006 | 0.141±0.001 |
| 4 | 0.799±0.005 | 5.434±0.014 | 229.16±3.26 | 0.034±0.002 | 0.211±0.006 | 0.975±0.004 | 5.275±0.021 | 195.48±4.12 | 0.09±0.004 | 0.437±0 | 0.923±0.007 | 5.342±0.036 | 209.02±7.37 | 0.037±0.002 | 0.196±0.004 |
| 5 | 0.753±0.008 | 5.667±0.021 | 289.19±6 | 0.024±0.002 | 0.254±0.004 | 0.951±0.004 | 5.765±0.014 | 318.94±4.35 | 0.029±0.001 | 0.563±0 | 0.892±0.006 | 5.641±0.047 | 282.07±13.11 | 0.036±0.004 | 0.254±0.006 |
| 6 | 0.671±0.01 | 5.75±0.019 | 314.29±5.93 | 0.019±0.001 | 0.29±0.006 | 0.885±0.007 | 6.136±0.02 | 462.51±9.06 | 0.025±0.002 | 0.704±0.014 | 0.833±0.006 | 6.002±0.019 | 404.41±7.56 | 0.025±0.001 | 0.326±0.002 |
| 7 | 0.576±0.01 | 5.617±0.009 | 275.21±2.49 | 0.022±0.001 | 0.299±0.003 | 0.834±0.006 | 6.29±0.021 | 539.52±11.44 | 0.015±0.002 | 0.793±0.013 | 0.726±0.009 | 6.189±0.023 | 487.47±11.36 | 0.022±0.003 | 0.404±0.001 |
| 8 | 0.455±0.007 | 5.363±0.015 | 213.33±3.24 | 0.028±0.001 | 0.293±0.002 | 0.771±0.005 | 6.292±0.016 | 540.19±8.66 | 0.015±0.002 | 0.832±0.012 | 0.549±0.005 | 6.116±0.022 | 453.29±9.84 | 0.02±0.002 | 0.456±0.003 |
| 9 | 0.322±0.005 | 4.997±0.014 | 148.03±2.02 | 0.039±0.001 | 0.284±0.006 | 0.694±0.008 | 6.242±0.014 | 513.7±6.98 | 0.016±0.002 | 0.865±0.011 | 0.398±0.005 | 5.79±0.017 | 327.02±5.68 | 0.024±0.003 | 0.445±0.005 |
| 10 | 0.183±0.008 | 4.447±0.025 | 85.38±2.14 | 0.053±0.007 | 0.268±0.012 | 0.643±0.01 | 6.262±0.013 | 524.33±6.63 | 0.01±0.001 | 0.909±0.01 | 0.18±0.002 | 4.936±0.021 | 139.26±2.9 | 0.046±0.006 | 0.426±0.007 |

**Table 12.** Prefix statistics by depth (Continue). Mean±95% CI.

## A.1.3. PER-LENGTH SMILES PERFORMANCE ON LONG HORIZON.

Table 13–15 show that increasing the horizon amplifies length-wise failure modes. TB rapidly loses support on long valid trajectories: its Frac/Count becomes negligible beyond $L \geq 12$ (e.g., Frac 0.003 at $L = 12$ and effectively zero thereafter),

and the corresponding Score degrades sharply at long lengths (e.g., 0.544/0.471/0.433/0.375 for $L = 9$–12), indicating severe long-horizon under-coverage. RapTB maintains substantially higher terminal quality on long trajectories (Score $\approx 0.75$–0.81 for $L = 10$–14 with near-perfect Acc), but still under-allocates mass to the extreme tail ($L = 15$) as reflected by a small Frac/Count. Combining RapTB with SubM shifts probability mass back to long lengths without sacrificing quality, yielding strong tail performance (at $L = 15$, Acc 0.84 and Score 0.85 with a much larger Frac/Count), and also improves long-length diversity/uniqueness compared to RapTB alone (Tables 14, 15). In contrast, SubTB places substantial mass on very long lengths (e.g., large Frac at $L = 15$) but exhibits low Acc/Score there, consistent with the termination/length instability discussed in the main text.

| $L$ | TB Acc | TB Score | TB Frac | TB Count | SubTB Acc | SubTB Score | SubTB Frac | SubTB Count | RapTB Acc | RapTB Score | RapTB Frac | RapTB Count | RapTB+SubM Acc | RapTB+SubM Score | RapTB+SubM Frac | RapTB+SubM Count |
|---|---|---|---|---|---|---|---|---|---|---|---|---|---|---|---|---|
| 1 | 1±0 | 0.707±0.003 | 0.242±0.012 | 774.7±37.9 | 1±0 | 0.765±0 | 0.002±0.001 | 1.7±0.7 | 1±0 | 0.7±0.005 | 0.033±0.004 | 105.7±11.6 | 1±0 | 0.694±0.012 | 0.018±0.001 | 56.5±2.9 |
| 2 | 1±0 | 0.748±0 | 0.34±0.01 | 1086.7±33.2 | 1±0 | 0.738±0.009 | 0.05±0.002 | 51.3±2.6 | 1±0 | 0.723±0.003 | 0.033±0.005 | 103±16.7 | 1±0 | 0.798±0.004 | 0.018±0.008 | 57.5±26.5 |
| 3 | 0.999±0.002 | 0.724±0.007 | 0.128±0.005 | 409±16.3 | 1±0 | 0.723±0.016 | 0.03±0.006 | 31±6 | 1±0 | 0.747±0.012 | 0.011±0.001 | 34.3±2.4 | 1±0 | 0.824±0.003 | 0.016±0.004 | 50.5±12.7 |
| 4 | 0.999±0.002 | 0.739±0 | 0.088±0.006 | 281.3±20.3 | 1±0 | 0.796±0.008 | 0.069±0.011 | 70.7±11.3 | 0.995±0.01 | 0.758±0.005 | 0.017±0.002 | 55.3±7.5 | 1±0 | 0.853±0.017 | 0.016±0.006 | 51±17.6 |
| 5 | 0.997±0.003 | 0.749±0.001 | 0.06±0.003 | 192.3±10.3 | 1±0 | 0.795±0.003 | 0.135±0.004 | 137±5.2 | 1±0 | 0.779±0.01 | 0.019±0.008 | 60±25.5 | 1±0 | 0.888±0.01 | 0.025±0.001 | 77±2 |
| 6 | 1±0 | 0.736±0.011 | 0.043±0.002 | 138.3±7.3 | 1±0 | 0.794±0.009 | 0.066±0.012 | 66.7±11.4 | 0.993±0.013 | 0.798±0.014 | 0.017±0.002 | 55.3±6.8 | 1±0 | 0.879±0.004 | 0.031±0.001 | 95.5±4.9 |
| 7 | 0.995±0.01 | 0.741±0.021 | 0.022±0.002 | 69.3±7.5 | 0.997±0.005 | 0.811±0.004 | 0.138±0.018 | 140.7±17.4 | 0.99±0.011 | 0.799±0.012 | 0.03±0.002 | 95.7±5.7 | 0.996±0.008 | 0.893±0.006 | 0.034±0.008 | 107±23.5 |
| 8 | 0.982±0.018 | 0.705±0.021 | 0.012±0.001 | 37±4.5 | 1±0 | 0.821±0.004 | 0.107±0.009 | 108.7±9.2 | 0.987±0.006 | 0.776±0.009 | 0.031±0.003 | 99±7.9 | 0.995±0.011 | 0.863±0.013 | 0.029±0.001 | 91±2 |
| 9 | 0.997±0.005 | 0.544±0.024 | 0.032±0.006 | 101±19.7 | 1±0 | 0.808±0.003 | 0.07±0.003 | 71±3 | 0.993±0.013 | 0.706±0.003 | 0.084±0.009 | 266.3±28 | 1±0 | 0.871±0.02 | 0.045±0.009 | 141±27.4 |
| 10 | 1±0 | 0.471±0.002 | 0.021±0.001 | 65.7±2.4 | 1±0 | 0.818±0.007 | 0.062±0.004 | 63.3±4 | 0.997±0.001 | 0.751±0.008 | 0.155±0.008 | 490.3±22.9 | 1±0 | 0.862±0.001 | 0.066±0.004 | 204±13.7 |
| 11 | 1±0 | 0.433±0.034 | 0.009±0.002 | 28±6.3 | 1±0 | 0.826±0.011 | 0.041±0.006 | 42±5.9 | 0.995±0.001 | 0.757±0.002 | 0.154±0.003 | 487.7±10.5 | 0.988±0.017 | 0.847±0.008 | 0.094±0.004 | 292±11.8 |
| 12 | 1±0 | 0.375±0.014 | 0.003±0.001 | 10±3.4 | 1±0 | 0.803±0.013 | 0.023±0.003 | 23.3±3.5 | 0.995±0.004 | 0.787±0.002 | 0.158±0.007 | 498.7±24.2 | 0.996±0.003 | 0.853±0.002 | 0.126±0.01 | 391.5±32.3 |
| 13 | 1±0 | 0.711±0 | 0±0 | – | 1±0 | 0.797±0.013 | 0.011±0.003 | 11.7±3.6 | 0.995±0.001 | 0.807±0.002 | 0.155±0.011 | 489±33.3 | 0.993±0.002 | 0.832±0.005 | 0.16±0.003 | 499±9.8 |
| 14 | 1±0 | 0.332±0.112 | 0.001±0 | 3.7±1.3 | 1±0 | 0.832±0.014 | 0.009±0.004 | 8.7±3.6 | 0.997±0.003 | 0.795±0.007 | 0.08±0.009 | 254±27.5 | 0.992±0.006 | 0.853±0.005 | 0.192±0.003 | 596±7.8 |
| 15 | – | – | – | – | 0.247±0.01 | 0.494±0.02 | 0.187±0.007 | 190.3±7.7 | 0.748±0.041 | 0.763±0.013 | 0.022±0.001 | 68.7±4 | 0.84±0.023 | 0.85±0.006 | 0.129±0.005 | 400±17.6 |

**Table 13.** Per-length valid-only core metrics of SMILES generation (mean±95% CI, $L_{\max} = 15$).

| $L$ | TB Entropy | TB FPDiv | SubTB Entropy | SubTB FPDiv | RapTB Entropy | RapTB FPDiv | RapTB+SubM Entropy | RapTB+SubM FPDiv |
|---|---|---|---|---|---|---|---|---|
| 1 | 2.45±0.04 | 0.82±0.011 | 0±0 | 0±0 | 1.8±0.06 | 0.625±0.009 | 0.92±0.03 | 0.455±0.024 |
| 2 | 2.57±0 | 0.813±0.004 | 0.83±0.21 | 0.344±0.071 | 2.07±0.12 | 0.802±0.004 | 1.04±0.2 | 0.661±0.025 |
| 3 | 2.49±0 | 0.834±0 | 1.49±0.05 | 0.793±0.005 | 1.6±0.14 | 0.706±0.02 | 1.21±0.24 | 0.682±0.027 |
| 4 | 2.18±0.03 | 0.791±0.009 | 1.21±0.08 | 0.673±0.012 | 1.27±0.23 | 0.748±0.04 | 1.17±0.06 | 0.653±0.026 |
| 5 | 2.42±0.04 | 0.868±0.003 | 1.25±0.02 | 0.748±0.007 | 1.72±0.05 | 0.825±0.008 | 1.32±0.19 | 0.793±0.017 |
| 6 | 2.5±0.06 | 0.879±0.004 | 1.62±0.02 | 0.841±0.005 | 1.53±0.19 | 0.79±0.028 | 1.73±0.04 | 0.879±0.003 |
| 7 | 2.32±0 | 0.879±0.003 | 0.93±0.05 | 0.705±0.015 | 1.55±0.11 | 0.826±0.006 | 1.69±0.1 | 0.857±0.013 |
| 8 | 2.23±0.03 | 0.881±0.005 | 1.19±0.09 | 0.796±0.007 | 1.82±0.02 | 0.835±0.003 | 2.05±0.1 | 0.873±0.007 |
| 9 | 1.51±0.17 | 0.718±0.019 | 1.43±0.03 | 0.846±0.001 | 1.75±0.04 | 0.808±0.005 | 1.91±0.06 | 0.872±0.009 |
| 10 | 1.18±0.04 | 0.648±0.006 | 1.2±0.13 | 0.782±0.019 | 1.72±0.02 | 0.833±0.003 | 2.17±0.03 | 0.891±0.007 |
| 11 | 0.82±0.23 | 0.524±0.048 | 1.33±0.07 | 0.835±0.006 | 1.74±0.01 | 0.835±0.001 | 2.08±0 | 0.883±0.006 |
| 12 | 0.46±0.03 | 0.422±0.029 | 1.38±0.14 | 0.846±0.003 | 1.57±0.02 | 0.819±0.001 | 2.05±0.01 | 0.9±0.001 |
| 13 | 0±0 | – | 1.24±0.3 | 0.83±0.017 | 1.38±0.02 | 0.797±0.003 | 2.02±0.06 | 0.894±0.003 |
| 14 | 0.22±0.35 | 0.39±0.209 | 1.09±0.12 | 0.813±0.015 | 1.46±0.05 | 0.82±0.002 | 2±0.03 | 0.897±0 |
| 15 | – | – | 2.29±0.05 | 0.884±0.001 | 1.57±0.07 | 0.825±0.007 | 1.85±0.02 | 0.889±0 |

**Table 14.** Per-length valid-only diversity metrics of SMILES generation (mean±95% CI, $L_{\max} = 15$).

| $L$ | TB UniqStr | TB UniqMol | TB UniqRateStr | TB UniqRateMol | SubTB UniqStr | SubTB UniqMol | SubTB UniqRateStr | SubTB UniqRateMol | RapTB UniqStr | RapTB UniqMol | RapTB UniqRateStr | RapTB UniqRateMol | RapTB+SubM UniqStr | RapTB+SubM UniqMol | RapTB+SubM UniqRateStr | RapTB+SubM UniqRateMol |
|---|---|---|---|---|---|---|---|---|---|---|---|---|---|---|---|---|
| 1 | 31±1.2 | 24.7±3 | 0±0 | 0±0 | 1±0 | 1±0 | 0.7±0.2 | 0.7±0.2 | 14±1.4 | 11.3±1.5 | 0.1±0 | 0.1±0 | 5.5±0.6 | 5.5±0.6 | 0.1±0 | 0.1±0 |
| 2 | 189±3.6 | 128.7±4.3 | 0.2±0 | 0.3±0 | 7.3±0.4 | 6.3±0.4 | 0.1±0 | 0.1±0 | 35.7±4.2 | 33±4.5 | 0.3±0 | 0.3±0 | 8.5±1.7 | 8.5±1.7 | 0.1±0 | 0.1±0 |
| 3 | 215.7±2.7 | 203.7±1.1 | 0.5±0 | 0.5±0 | 14.3±0.8 | 14.3±0.8 | 0.5±0.1 | 0.5±0.1 | 20.7±1.5 | 19.7±1.8 | 0.6±0 | 0.6±0 | 8.5±0.6 | 8.5±0.6 | 0.2±0 | 0.2±0 |
| 4 | 170±9.3 | 164.7±8.6 | 0.6±0 | 0.6±0 | 22±1.4 | 20.7±1.8 | 0.3±0 | 0.3±0 | 32±4.5 | 31.3±4.8 | 0.6±0.1 | 0.6±0.1 | 15.5±1.7 | 15.5±1.7 | 0.3±0 | 0.3±0 |
| 5 | 178.7±6.6 | 176±6.8 | 0.9±0 | 0.9±0 | 47±1.2 | 42.3±0.4 | 0.3±0 | 0.3±0 | 51.3±11 | 48.7±7.7 | 0.9±0 | 0.8±0.1 | 23±0 | 23±0 | 0.3±0 | 0.3±0 |
| 6 | 133±5.4 | 131.7±5.7 | 1±0 | 1±0 | 43.7±3 | 43±2.9 | 0.7±0 | 0.7±0.1 | 41.3±5 | 41.3±5 | 0.7±0.1 | 0.7±0.1 | 36.5±0.6 | 36.5±0.6 | 0.4±0 | 0.4±0 |
| 7 | 69.3±4.8 | 69.3±4.8 | 1±0 | 1±0 | 55.3±3 | 48±3.8 | 0.4±0 | 0.3±0 | 76.7±2.3 | 72.3±3.3 | 0.8±0 | 0.8±0 | 45±7.9 | 44.5±7.4 | 0.4±0 | 0.4±0 |
| 8 | 37±2.9 | 36.7±2.5 | 1±0 | 1±0 | 68±2.6 | 63.3±1.7 | 0.6±0 | 0.6±0 | 94.7±3.5 | 93±2.6 | 1±0 | 0.9±0 | 39.5±4 | 39.5±4 | 0.4±0 | 0.4±0 |
| 9 | 40±3.3 | 36.3±1.5 | 0.4±0 | 0.4±0 | 55.3±1.5 | 54.7±1.7 | 0.8±0 | 0.8±0 | 167.3±4.7 | 157.7±2.2 | 0.6±0 | 0.6±0 | 63±4.5 | 63±4.5 | 0.4±0 | 0.4±0 |
| 10 | 26.3±1.5 | 26.3±1.5 | 0.4±0 | 0.4±0 | 41.7±1.5 | 41.3±1.7 | 0.7±0 | 0.7±0 | 358.7±7.2 | 324.7±6.5 | 0.7±0 | 0.7±0 | 101±0 | 101±0 | 0.5±0 | 0.5±0 |
| 11 | 14.7±1.1 | 14.7±1.1 | 0.5±0.1 | 0.5±0.1 | 35.7±3.6 | 35.7±3.6 | 0.8±0 | 0.8±0 | 415.7±9.9 | 397.3±8.8 | 0.9±0 | 0.8±0 | 139.5±6.2 | 139.5±6.2 | 0.5±0 | 0.5±0 |
| 12 | 5.7±0.8 | 5.7±0.8 | 0.6±0.1 | 0.6±0.1 | 22.3±2.3 | 22.3±2.3 | 1±0.1 | 1±0.1 | 391±10.3 | 360±10 | 0.8±0 | 0.7±0 | 226.5±4 | 226.5±4 | 0.6±0 | 0.6±0 |
| 13 | 1±0 | 1±0 | 1±0 | 1±0 | 11.3±2.2 | 11.3±2.2 | 1±0 | 1±0 | 357.7±12.8 | 328.7±11.5 | 0.7±0 | 0.7±0 | 334.5±18.7 | 334±19.2 | 0.7±0 | 0.7±0 |
| 14 | 2.3±0.4 | 2.3±0.4 | 0.7±0.2 | 0.7±0.2 | 8.7±2.3 | 8.7±2.3 | 1±0 | 1±0 | 219.7±12.6 | 211.3±12.5 | 0.9±0 | 0.8±0 | 303±9.1 | 261.5±11.9 | 0.5±0 | 0.5±0 |
| 15 | – | – | – | – | 186±5.7 | 186±5.7 | 1±0 | 1±0 | 67±2.1 | 66.7±1.8 | 1±0 | 1±0 | 217.5±11.9 | 217.5±11.9 | 0.5±0 | 0.5±0 |

**Table 15.** Per-length valid-only uniqueness metrics of SMILES generation (mean±95% CI, $L_{\max} = 15$).

A.1.4. PER-DEPTH PREFIX COLLAPSE ON LONG HORIZON ($L_{\max} = 15$).

Tables 16–17 report prefix statistics at depth $k$, where concentration metrics (PefEnt/Eff/Top1/UniqueRate) must be interpreted jointly with Survival. On the long horizon, TB exhibits a textbook collapse pattern: Survival drops from $0.758$ at $k=2$ to $0.034$ at $k=10$ and is essentially zero for $k \geq 12$, while Top1 peaks at $0.424/0.457$ for $k=7/8$, indicating that only a tiny fraction of trajectories reach deep prefixes and those prefixes are highly shared. RapTB substantially improves deep-prefix survival (e.g., $0.723$ at $k=10$) and keeps Top1 low at depth ($\approx 0.06$–$0.07$ for $k=7$–$10$), consistent with sustained branching and reduced prefix concentration; RapTB+SubM further lowers deep-prefix Top1 while maintaining high Survival, suggesting that improved replay coverage helps prevent the buffer from over-focusing on a few dominant prefixes under long-horizon generation. At $L_{\max} = 15$, we only report RapTB+SubM as the combined setting due to the additional training cost of re-running all baseline+SubM variants at this horizon. Finally, although SubTB shows strong prefix-level dispersion (low Top1 with high PefEnt/Eff and high Survival), this alone does not imply better terminal quality or validity; thus these prefix statistics should be interpreted together with the per-length terminal metrics in Tables 13–15, where SubTB degrades on long-length performance.

| $k$ | TB | | | | | SubTB | | | | |
|---|---|---|---|---|---|---|---|---|---|---|
| | Survival | PefEnt | Eff | Top1 | UniqueRate | Survival | PefEnt | Eff | Top1 | UniqueRate |
| 1 | 1±0 | 2.637±0.021 | 13.97±0.3 | 0.236±0.009 | 0.013±0 | 1±0 | 1.235±0.016 | 3.44±0.05 | 0.508±0.007 | 0.012±0.001 |
| 2 | 0.758±0.007 | 4.775±0.027 | 118.61±3.24 | 0.082±0.005 | 0.152±0.008 | 0.998±0 | 3.085±0.009 | 21.87±0.2 | 0.164±0.009 | 0.07±0.003 |
| 3 | 0.418±0.005 | 5.495±0.048 | 243.8±11.8 | 0.108±0.008 | 0.447±0.01 | 0.948±0.002 | 3.987±0.031 | 53.93±1.7 | 0.098±0 | 0.179±0.004 |
| 4 | 0.29±0.006 | 5.452±0.052 | 233.74±12.43 | 0.145±0.008 | 0.591±0.016 | 0.917±0.002 | 4.466±0.035 | 87.04±3 | 0.082±0.004 | 0.262±0.003 |
| 5 | 0.202±0.004 | 5.178±0.092 | 178.4±16.91 | 0.208±0.014 | 0.682±0.02 | 0.848±0.007 | 5.041±0.093 | 155.54±14.12 | 0.058±0.008 | 0.375±0.016 |
| 6 | 0.142±0.006 | 4.373±0.141 | 80.31±11.33 | 0.294±0.02 | 0.607±0.025 | 0.714±0.008 | 5.393±0.093 | 221.07±20.27 | 0.066±0.011 | 0.522±0.016 |
| 7 | 0.099±0.005 | 3.278±0.175 | 27.03±4.53 | 0.424±0.027 | 0.457±0.031 | 0.648±0.008 | 5.46±0.069 | 235.93±16.3 | 0.071±0.012 | 0.593±0.011 |
| 8 | 0.077±0.004 | 2.639±0.112 | 14.11±1.54 | 0.457±0.002 | 0.316±0.031 | 0.51±0.005 | 5.702±0.033 | 299.69±10.07 | 0.03±0.006 | 0.727±0.008 |
| 9 | 0.065±0.003 | 2.835±0.08 | 17.1±1.37 | 0.309±0.03 | 0.28±0.024 | 0.403±0.004 | 5.654±0.036 | 285.55±10.24 | 0.038±0.007 | 0.815±0.012 |
| 10 | 0.034±0.001 | 2.977±0.088 | 19.73±1.72 | 0.154±0.018 | 0.333±0.031 | 0.333±0.002 | 5.547±0.039 | 256.65±9.97 | 0.045±0.007 | 0.864±0.013 |
| 11 | 0.013±0.002 | 2.415±0.052 | 11.21±0.59 | 0.326±0.015 | 0.441±0.031 | 0.271±0.004 | 5.51±0.03 | 247.38±7.36 | 0.019±0.001 | 0.933±0.008 |
| 12 | 0.004±0.001 | 1.737±0.216 | 5.86±1.35 | 0.365±0.045 | 0.517±0.046 | 0.23±0.004 | 5.398±0.029 | 221.18±6.45 | 0.021±0.003 | 0.967±0.005 |
| 13 | 0.001±0 | 0.347±0.43 | 1.61±0.76 | 0.833±0.207 | 0.428±0.205 | 0.207±0.006 | 5.304±0.039 | 201.39±7.93 | 0.024±0.004 | 0.976±0.004 |
| 14 | 0.001±0 | 0.745±0.224 | 2.18±0.52 | 0.6±0.172 | 0.689±0.215 | 0.195±0.005 | 5.252±0.037 | 191.04±7.2 | 0.024±0.003 | 0.978±0.005 |
| 15 | 0±0 | 0±0 | 0±0 | 0±0 | – | 0.187±0.004 | 5.205±0.035 | 182.38±6.38 | 0.025±0.003 | 0.977±0.005 |

**Table 16.** Prefix statistics by depth on SMILES generation (mean±95% CI, $L_{\max} = 15$): TB vs. SubTB.

| $k$ | RapTB | | | | | RapTB+SubM | | | | |
|---|---|---|---|---|---|---|---|---|---|---|
| | Survival | PefEnt | Eff | Top1 | UniqueRate | Survival | PefEnt | Eff | Top1 | UniqueRate |
| 1 | 1±0 | 1.693±0.014 | 5.44±0.08 | 0.375±0.004 | 0.008±0 | 1±0 | 1.938±0.001 | 6.94±0.01 | 0.342±0 | 0.007±0 |
| 2 | 0.967±0.002 | 3.438±0.033 | 31.14±1.03 | 0.204±0.006 | 0.051±0.002 | 0.982±0.001 | 3.659±0.011 | 38.83±0.43 | 0.125±0.009 | 0.039±0 |
| 3 | 0.934±0.001 | 4.542±0.04 | 93.95±3.74 | 0.106±0.006 | 0.138±0.006 | 0.963±0.005 | 4.227±0.033 | 68.53±2.26 | 0.126±0.01 | 0.09±0.003 |
| 4 | 0.923±0.001 | 5.249±0.026 | 190.42±4.86 | 0.071±0.002 | 0.238±0.004 | 0.947±0.003 | 4.829±0.009 | 125.11±1.11 | 0.094±0.003 | 0.148±0.002 |
| 5 | 0.906±0.001 | 5.822±0.019 | 337.73±6.41 | 0.07±0.005 | 0.345±0.003 | 0.931±0.001 | 5.184±0.001 | 178.44±0.24 | 0.095±0.003 | 0.204±0.002 |
| 6 | 0.887±0.005 | 6.239±0.013 | 512.44±6.71 | 0.071±0.005 | 0.445±0.004 | 0.906±0 | 5.701±0.007 | 299.31±2.04 | 0.044±0.002 | 0.263±0.002 |
| 7 | 0.869±0.005 | 6.559±0.021 | 705.68±14.92 | 0.073±0.005 | 0.542±0.002 | 0.875±0.001 | 6.07±0.015 | 432.83±6.29 | 0.02±0.001 | 0.322±0.002 |
| 8 | 0.839±0.004 | 6.78±0.016 | 879.89±14.09 | 0.058±0.004 | 0.614±0.001 | 0.841±0.004 | 6.151±0.022 | 469.45±10.24 | 0.018±0.002 | 0.357±0.003 |
| 9 | 0.808±0.003 | 6.89±0.013 | 982.52±12.91 | 0.053±0.004 | 0.658±0.002 | 0.812±0.003 | 6.203±0.001 | 494.29±0.41 | 0.019±0.002 | 0.382±0.004 |
| 10 | 0.723±0.008 | 7.041±0.005 | 1142.49±5.93 | 0.024±0.001 | 0.718±0.006 | 0.766±0.008 | 6.313±0.017 | 551.78±9.32 | 0.02±0.002 | 0.432±0.001 |
| 11 | 0.568±0.011 | 6.93±0.016 | 1022.69±16.25 | 0.024±0 | 0.769±0.007 | 0.701±0.006 | 6.386±0.007 | 593.32±4.26 | 0.022±0.002 | 0.481±0.002 |
| 12 | 0.414±0.01 | 6.638±0.006 | 763.56±4.76 | 0.03±0.004 | 0.775±0.007 | 0.607±0.008 | 6.371±0.018 | 584.42±10.32 | 0.016±0.003 | 0.52±0.004 |
| 13 | 0.257±0.012 | 6.201±0.01 | 493.35±5.11 | 0.046±0.007 | 0.793±0.009 | 0.481±0.003 | 6.205±0.02 | 495.45±9.87 | 0.02±0.004 | 0.548±0 |
| 14 | 0.102±0.005 | 5.587±0.029 | 267.14±7.85 | 0.024±0.004 | 0.889±0.021 | 0.32±0.001 | 5.677±0.083 | 292.79±24.31 | 0.031±0.005 | 0.511±0.019 |
| 15 | 0.022±0.001 | 4.195±0.03 | 66.4±2.02 | 0.029±0.001 | 0.976±0.005 | 0.129±0.003 | 4.899±0.098 | 134.67±13.23 | 0.077±0.015 | 0.543±0.016 |

**Table 17.** Prefix statistics by depth on SMILES generation (mean±95% CI, $L_{\max} = 15$): RapTB vs. RapTB+SubM.

## A.2. Expr24

We provide results with 95% CI and per-length termination probability analysis of different objectives under various replay strategies.

## A.3. CommonGen.

To make the behavioral differences visible, we show a few representative validation probes where the contrast is the clearest. Each row corresponds to the *same* probe instance across methods (IDs omitted for compactness; see released CSV logs for exact indices). We report the effective length (Len) before padding.

**Table 18. Expr24 results under four replay schemes.** All the experiments are run under 3 different random seeds with 95% CI. Per-run sample size is 6400.

| Replay | Objective | Unique✓ | NormCov | Acc | $KL(\pi \to p^*)$ | $KL(p^* \to \pi)$ | $JS_{tok}$ |
|---|---|---|---|---|---|---|---|
| PRT | TB | 103.7±3.2 | 0.016±0.001 | 0.999±0.000 | 1.105±0.002 | 7.803±0.060 | 0.292±0.001 |
| | SubTB | 292.0±2.9 | 0.046±0.000 | 0.311±0.002 | 0.424±0.010 | 0.672±0.077 | 0.107±0.003 |
| | RapTB | 129.3±0.4 | 0.020±0.000 | 0.992±0.001 | 0.908±0.003 | 5.538±0.005 | 0.230±0.001 |
| RP | TB | 5.3±0.4 | 0.001±0.000 | 1.000±0.000 | 1.297±0.001 | 11.403±0.282 | 0.339±0.000 |
| | SubTB | 324.7±2.7 | 0.051±0.000 | 0.229±0.005 | 0.455±0.005 | 0.865±0.083 | 0.109±0.002 |
| | RapTB | 246.7±7.1 | 0.039±0.001 | 0.991±0.000 | 0.561±0.001 | 4.480±0.002 | 0.147±0.000 |
| SubM | TB | 642.0±5.6 | 0.100±0.001 | 0.996±0.001 | 0.182±0.001 | 0.441±0.005 | 0.049±0.000 |
| | SubTB | 331.3±22.7 | 0.052±0.004 | 0.061±0.005 | 0.149±0.008 | 0.286±0.070 | 0.040±0.002 |
| | RapTB | 1337.3±7.5 | 0.209±0.001 | 0.994±0.001 | 0.169±0.001 | 0.623±0.004 | 0.048±0.000 |
| Oracle | TB | 5198.0±5.2 | 0.812±0.001 | 0.919±0.001 | 0.062±0.001 | 0.066±0.001 | 0.016±0.000 |
| | SubTB | 35.7±2.9 | 0.006±0.000 | 0.006±0.000 | 0.266±0.009 | 1.491±0.413 | 0.071±0.003 |
| | RapTB | 5220.7±4.3 | 0.816±0.001 | 0.945±0.001 | 0.052±0.001 | 0.056±0.001 | 0.013±0.000 |

| Replay | Objective | $\ell = 3$ | $\ell = 5$ | $\ell = 7$ | $\ell = 9$ |
|---|---|---|---|---|---|
| RP | TB | – | – | -0.000 | -0.001 |
| | SubTB | -5.220 | -1.292 | -1.064 | -79.638 |
| | RapTB | – | -2.451 | -2.319 | -0.065 |
| | RootSubTBLogZ | -0.709 | -0.606 | -0.411 | -0.068 |
| Oracle | TB | – | -4.391 | -1.779 | -0.441 |
| | SubTB | -1.017 | -2.803 | -4.530 | -86.415 |
| | RapTB | – | -8.312 | -3.341 | -0.644 |
| | RootSubTBLogZ | -0.442 | -0.417 | -0.354 | -1.432 |

**Table 19.** Per-length $\log p_{term}$ on Expr24.

# B. Additional Baselines

## B.1. RL Baselines: PPO and GRPO

To contextualize RapTB within the broader LLM fine-tuning landscape, we compare against PPO and GRPO (Shao et al., 2024). Both methods are trained with the same backbone (Llama-3.2-1B, LoRA rank 16) and reward function. As shown in the main-text Tables 1 and 3, reward-maximizing RL methods achieve reasonable task scores but suffer severe diversity collapse: PPO collapses to a single mode on both tasks, and GRPO achieves Entropy $\leq 0.98$ on SMILES. This is consistent with the fundamental difference between reward maximization and reward-proportional sampling (Hu et al., 2024): RL concentrates mass on the single best mode, whereas GFlowNets target the full reward-proportional distribution.

## B.2. AvgPrefixTB: Uniform Prefix Trajectory Balance

**Definition.** In a terminable prefix tree, every prefix $s_{0:k}$ can be regarded as a complete trajectory by appending a stop token $\top$. AvgPrefixTB averages the standard TB residual over all such prefix terminations along a sampled trajectory:

$$\mathcal{L}_{AvgPrefixTB}(\xi) \triangleq \frac{1}{\tau} \sum_{k=1}^{\tau} \left(\Delta_k^{TB}(\xi)\right)^2, \tag{13}$$

where $\Delta_k^{TB}(\xi)$ is the TB residual at prefix $s_{0:k}$ (Eq. 21). This provides $O(N)$ constraints like RapTB, but differs in two key respects: (i) each residual retains the learnable $\log Z_\theta$ instead of canceling it via rooting, and (ii) it uses the raw stop-reward $\log R(s_{0:k}^\top)$ at each prefix rather than absorbed suffix targets.

**Motivation as a baseline.** AvgPrefixTB tests whether simply densifying TB supervision across all prefixes—without the structural design choices of RapTB (rooted residuals, absorbed targets, termination gradient stopping)—is sufficient to address prefix collapse and length bias.

**Table 20. Samples from CommonGen task.** SubTB often exhibits max-length / off-distribution artifacts, while RapTB+SubM yields shorter and cleaner sentences.

| TB | SubTB | RapTB | RapTB+SubM |
|---|---|---|---|
| (Len=19)
The cat likes to sit on the couch with a pet that lives in a cat, because it. | (Len=20)
My pet cat likes to sleep on a couch, I know, what a pet perv! This is. | (Len=15)
The pet cat loves the couch, and the cat's mom was an mascot. | (Len=10)
A pet cat likes to sleep on a couch. |
| (Len=15)
A kid loves to dance in a room that a kid likes to dance in. | (Len=20)
My kid loves to dance in her room with all of her clothes on except her persnare cal. | (Len=7)
I dance in the room for kid. | (Len=8)
kid loves to dance in her room!! |
| (Len=20)
There is a hockey team that plays at a rink and plays on a team on a hockey Claus. | (Len=20)
The team practiced hockey at the rink, and do freeze to ourselves towards the rinks touring doors. | (Len=15)
A team played at the hockey rink with the best players on their own. | (Len=11)
The team practiced hockey at the rink for four hours. |
| (Len=20)
There was a flight of stairs, and a jump, a flight of stairs, and a jump, | (Len=20)
I jump up a flight of stair, landing in a little flight-recorded levy down to current it. | (Len=14)
I jump up a flight of stair to get to school in the morning. | (Len=11)
I walk up a stair: jump up a flight. |

**Results.** Table 21 compares AvgPrefixTB against TB and RapTB on both SMILES and Expr24.

**Table 21. AvgPrefixTB comparison on SMILES and Expr24.** All methods use the same backbone (Llama-3.2-1B, LoRA rank 16) and RP replay. SMILES metrics are computed on valid samples; Len denotes mean token length ($L_{\max} = 10$).

| Method | SMILES | | | | Expr24 | |
| | Acc↑ | Score↑ | Entropy↑ | Len | Acc↑ | Unique$_\checkmark$↑ |
|---|---|---|---|---|---|---|
| TB | 0.998 | 0.717 | 2.503 | 3.06 | 1.000 | 5.3 |
| AvgPrefixTB | 1.000 | 0.661 | 0.665 | 2.89 | 0.998 | 142.0 |
| RapTB | 0.996 | 0.740 | 2.448 | 6.14 | 0.991 | 246.7 |
| RapTB+SubM | 0.988 | 0.844 | 2.726 | 7.44 | 0.994 | 1337.3 |

**Analysis.** AvgPrefixTB exhibits a pronounced short-sequence bias on SMILES: average length is 2.89 with 54% of generated mass concentrated on lengths 1–2. This is because averaging TB residuals uniformly across all prefixes creates a *shortcut*: early prefixes already have near-zero residuals (they are close to the root and have few accumulated transitions), so the model can minimize the average loss cheaply by terminating early and concentrating probability on short, high-reward trajectories. Score (0.661) and Entropy (0.665) are both substantially below TB (0.717 / 2.503) and RapTB (0.740 / 2.448).

On Expr24, AvgPrefixTB improves unique correct solutions over TB (142.0 vs. 5.3), indicating that prefix-level supervision does help with mode diversity even in its simplest form. However, it remains well below RapTB (246.7; NormCov 0.016 vs. 0.039), confirming that the specific design choices of RapTB—rooted residuals that cancel $\log Z_\theta$, absorbed suffix targets for informative prefix credit, and termination gradient stopping—are individually and jointly important. The ablation in Table 6 (main text) provides further evidence for each component.

## B.3. AMP Biological Sequence Generation

We evaluate on the antimicrobial peptide (AMP) generation task (Jain et al., 2022), a standard GFlowNet benchmark with amino-acid vocabulary and non-differentiable reward. We additionally compare against DynaPPO, COMs, and a standard GFlowNet baseline from Jain et al. (2022).

RapTB+SubM achieves the best performance–diversity–novelty trade-off within only 3K training steps. SubTB exhibits the same termination drift observed in SMILES and Expr24, collapsing to maximum length (49.3) and inflating diversity/novelty through unnaturally long sequences rather than genuine structural variation. This confirms that the termination drift failure mode generalizes to biological sequence generation with a fundamentally different vocabulary and reward structure.

**Table 22. AMP biological sequence generation.** †: SubTB collapses to max length; its diversity/novelty is inflated by raw edit distance over unnaturally long sequences.

| Method | Perf↑ | Div↑ | Novelty↑ | Len | Steps |
|---|---|---|---|---|---|
| DynaPPO | 0.938 | 12.12 | 9.31 | $\sim$20 | 10K |
| COMs | 0.761 | 19.38 | 26.47 | $\sim$20 | 10K |
| GFlowNet | 0.868 | 11.32 | 15.72 | $\sim$20 | 10K |
| TB | 0.927 | 7.39 | 10.65 | 17.4 | 10K |
| SubTB$^\dagger$ | 0.897 | 21.37 | 28.68 | 49.3 | 10K |
| RapTB | 0.919 | 8.83 | 14.44 | 22.4 | 5K |
| RapTB+SubM | **0.916** | **16.92** | **15.77** | 25.6 | 3K |

# C. Scaling Study

To verify that the identified failure modes are structural rather than capacity-limited, we scale from Llama-3.2-1B to 3B, 8B (Llama-3.2), and 32B (Qwen3) on SMILES. All runs use LoRA (rank 16) and the same reward/decoding configuration as the 1B experiments.

**Table 23. Scaling study on SMILES generation across model sizes and architectures.**

| Scale | Method | Acc↑ | Score↑ | FPDiv↑ | Ent↑ | Len |
|---|---|---|---|---|---|---|
| 3B (Llama) | TB | 0.999 | 0.716 | 0.838 | 1.92 | 2.69 |
| | SubTB | 0.311 | 0.222 | 0.854 | 2.56 | 9.52 |
| | RapTB | 1.000 | 0.795 | 0.839 | 1.81 | 7.99 |
| | RapTB+SubM | 0.998 | 0.869 | 0.936 | 2.41 | 8.05 |
| 8B (Llama) | TB | 1.000 | 0.715 | 0.775 | 1.84 | 2.98 |
| | SubTB | 0.391 | 0.307 | 0.869 | 2.72 | 9.22 |
| | RapTB | 0.999 | 0.825 | 0.852 | 1.89 | 8.09 |
| | RapTB+SubM | 0.998 | 0.873 | 0.937 | 2.51 | 7.65 |
| 32B (Qwen3) | TB | 1.000 | 0.762 | 0.860 | 2.06 | 3.15 |
| | SubTB | 0.795 | 0.626 | 0.864 | 2.19 | 7.31 |
| | RapTB | 0.998 | 0.794 | 0.880 | 2.16 | 6.19 |
| | RapTB+SubM | 0.998 | 0.867 | 0.896 | 2.23 | 7.56 |

Three observations emerge:

(a) **SubTB termination drift persists across all scales.** Accuracy remains low at every size (0.311/0.391/0.795 at 3B/8B/32B), and even 32B cannot fully resolve the issue. This confirms the failure is structural (arising from the overlapping-window objective) rather than capacity-limited.

(b) **RapTB scales reliably.** Accuracy stays above 0.998 at all scales, with diversity improving with model size (FPDiv: $0.839 \rightarrow 0.852 \rightarrow 0.880$).

(c) **RapTB+SubM achieves the best quality–diversity trade-off at every scale.** At 32B: highest QED (0.867), strong diversity (FPDiv 0.896), and near-perfect validity (Acc 0.998). These results across two architecture families confirm that the benefits of RapTB+SubM are architecture-independent.

# D. Hyperparameter Sensitivity

RapTB introduces several hyperparameters beyond TB: auxiliary weight $\eta$, soft-backup temperature $\beta$, distance penalty $\rho$, mix weight $\alpha$, distance discount $\gamma$, minimum prefix depth $k_{\min}$ (with schedule), and horizon cap $K$. To assess robustness, we conduct cross-task sweeps over $(\beta, \rho)$ (18 configs total) plus separate $\eta$ and $k_{\min}$ ablations. Across all settings, no catastrophic failure, length collapse, or termination drift is observed.

### D.1. Expr24

$(\beta, \rho)$ **grid (9 experiments).** $\beta$ and $\rho$ govern the bias–variance trade-off of the absorbed target: $\beta$ controls the smoothness of the soft backup, and $\rho$ controls the distance-decay rate.

**Table 24.** $(\beta, \rho)$ sensitivity on Expr24 (RP replay).

| $\beta$ | $\rho$ | Acc↑ | Diversity↑ |
|---|---|---|---|
| 1 | 0.0 | 0.999 | 1.010 |
| 1 | 0.1 | 0.999 | 0.993 |
| 1 | 0.5 | 1.000 | 1.015 |
| 3 | 0.0 | 0.990 | 0.950 |
| 3 | 0.1 | 1.000 | 0.769 |
| 3 | 0.5 | 0.997 | 0.978 |
| 5 | 0.0 | 0.983 | 1.022 |
| 5 | 0.1 | 0.998 | 0.968 |
| 5 | 0.5 | 0.999 | 0.912 |

All 9 configs maintain Acc $\geq 0.983$. The parameters primarily affect the accuracy–diversity balance rather than causing qualitative failures.

$\eta$ **sweep ($\beta=3, \rho=0.5$).** Higher $\eta$ strengthens the auxiliary prefix credit signal, improving diversity at a mild cost to accuracy.

**Table 25.** Auxiliary weight $\eta$ sensitivity on Expr24.

| $\eta$ | Acc↑ | Diversity↑ |
|---|---|---|
| 0.10 | 0.999 | 0.983 |
| 0.25 | 0.997 | 0.978 |
| 0.50 | 0.987 | 1.149 |

$k_{\min}$ **ablation ($\beta=3, \rho=0.5$).** $k_{\min}$ controls the minimum prefix depth receiving auxiliary supervision. Smaller $k_{\min}$ emphasizes shorter prefixes, creating a shortcut toward high-reward short sequences that improves accuracy but reduces diversity. The linear schedule ($7{\rightarrow}3$) used in the paper provides the best balance.

**Table 26.** $k_{\min}$ schedule sensitivity on Expr24.

| $k_{\min}$ variant | Acc↑ | Diversity↑ |
|---|---|---|
| Fixed $k_{\min}{=}3$ | 0.998 | 0.852 |
| Schedule $7{\rightarrow}3$ | 0.997 | 0.978 |
| Fixed $k_{\min}{=}7$ | 0.969 | 1.025 |

### D.2. SMILES

$(\beta, \rho)$ **grid (9 experiments).** Eight of nine configs achieve Acc $\geq 0.991$; only $(\beta=10, \rho=0)$ shows mild degradation (0.968), corresponding to high temperature with zero distance penalty. No config exhibits length collapse: all average lengths fall in $[5.5, 7.5]$, far from TB's collapsed 3.06. Score ($\geq 0.72$) and FPDiv ($\geq 0.83$) remain robust across all settings.

## E. Metrics: Formal Definitions and Protocol

**Sampling and aggregation protocol.** For each run, we draw $N$ i.i.d. terminal samples $\{x_i\}_{i=1}^{N}$ from the learned sampler (For SMILES, $N = 3200$, For Expr24, $N = 6400$). Let $\mathbb{I}_{\text{valid}}(x) \in \{0, 1\}$ indicate whether $x$ satisfies task constraints. Let $\mathcal{D}$ denote the multiset of all samples and let

$$\mathcal{D}_{\text{valid}} \triangleq \{x_i \in \mathcal{D} : \mathbb{I}_{\text{valid}}(x_i) = 1\}, \qquad n_{\text{valid}} \triangleq |\mathcal{D}_{\text{valid}}|.$$

**Table 27.** $(\beta, \rho)$ sensitivity on SMILES (RP replay).

| $\beta$ | $\rho$ | Acc↑ | Entropy↑ |
|---|---|---|---|
| 1 | 0.0 | 0.992 | 2.173 |
| 1 | 0.1 | 0.994 | 2.161 |
| 1 | 0.5 | 0.992 | 2.162 |
| 5 | 0.0 | 0.995 | 1.997 |
| 5 | 0.1 | 0.991 | 2.076 |
| 5 | 0.5 | 0.997 | 2.079 |
| 10 | 0.0 | 0.968 | 2.279 |
| 10 | 0.1 | 0.999 | 1.986 |
| 10 | 0.5 | 0.997 | 2.036 |

Unless explicitly stated otherwise, all metrics *except* Acc are computed on $\mathcal{D}_{\text{valid}}$. We report a metric as 0 if its denominator is 0 (e.g., $n_{\text{valid}} = 0$).

Across random seeds, we report the mean and a two-sided 95% confidence interval. With $S$ seeds and per-seed values $m_1, \ldots, m_S$, we report

$$\bar{m} \pm t_{S-1, 0.975} \frac{\text{sd}(m_1, \ldots, m_S)}{\sqrt{S}}.$$

### E.1. Terminal-level metrics

**Accuracy / validity rate (Acc).**   Acc measures the fraction of valid samples among all $N$ draws:

$$\text{Acc} \triangleq \frac{1}{N} \sum_{i=1}^{N} \mathbb{I}_{\text{valid}}(x_i). \tag{14}$$

For SMILES, $\mathbb{I}_{\text{valid}}(x) = 1$ requires chemical validity and scaffold consistency. For Expr24, $\mathbb{I}_{\text{valid}}(x) = 1$ requires syntactic validity and $\text{eval}(x) = 24$.

**Task score on valid samples (Score).**   Let $s(x)$ be the task score.

$$\text{Score} \triangleq \frac{1}{n_{\text{valid}}} \sum_{x \in \mathcal{D}_{\text{valid}}} s(x). \tag{15}$$

For Expr24 with binary reward, $s(x) = \mathbb{I}_{\text{valid}}(x)$, so Score numerically equals Acc.

**Pre-EOS length and length statistics.**   Each terminal $x$ is a variable-length token sequence with pre-EOS length $\ell(x)$ (number of tokens before EOS/$\top$). We report

$$\text{Len} \triangleq \frac{1}{n_{\text{valid}}} \sum_{x \in \mathcal{D}_{\text{valid}}} \ell(x), \qquad \text{Len}_{50}, \text{Len}_{90} \text{ as percentiles of } \{\ell(x) : x \in \mathcal{D}_{\text{valid}}\}.$$

**Length-bin fractions and counts.**   Given an integer bin $[a, b]$ (inclusive), define

$$\text{Frac}[a\text{--}b] \triangleq \frac{1}{n_{\text{valid}}} \sum_{x \in \mathcal{D}_{\text{valid}}} \mathbb{I}[a \leq \ell(x) \leq b], \qquad \text{Count}[a\text{--}b] \triangleq \sum_{x \in \mathcal{D}_{\text{valid}}} \mathbb{I}[a \leq \ell(x) \leq b].$$

For an open-ended bin $[a, +\infty)$, replace the indicator with $\mathbb{I}[\ell(x) \geq a]$.

**Termination calibration** ($\log p_{\text{term}}(\tau)$).   For each sampled trajectory, let $\tau_i$ be the sampled stop step (the position where EOS/$\top$ is taken). Define the per-sample termination log-probability

$$\log p_{\text{term}}(\tau_i) \triangleq \log q_\theta(\top \mid s_{0:\tau_i}),$$

evaluated from the model's raw termination head (no masking/renormalization). We report the mean over all samples:

$$\log p_{\text{term}}(\tau) \;\triangleq\; \frac{1}{N} \sum_{i=1}^{N} \log p_{\text{term}}(\tau_i).$$

More negative values indicate overly suppressed termination.

**Uniqueness metrics for SMILES.** Let $\text{canon}(x)$ denote canonicalization used in evaluation (e.g., canonical SMILES / canonical molecule identity). We define

$$\texttt{UniqStr} \triangleq |\{x : x \in \mathcal{D}_{\text{valid}}\}|, \qquad \texttt{UniqRateStr} \triangleq \texttt{UniqStr}/n_{\text{valid}}.$$

For molecule-level uniqueness, define $\texttt{UniqMol} \triangleq |\{\text{canon}(x) : x \in \mathcal{D}_{\text{valid}}\}|$ and $\texttt{UniqRateMol} \triangleq \texttt{UniqMol}/n_{\text{valid}}$.

### E.2. Token entropy

**Ragged token entropy (`Entropy`).** Let $\mathcal{D}_{\text{valid}} = \{x_i\}_{i=1}^{n_{\text{valid}}}$ and let $\ell_i \triangleq \ell(x_i)$. For each position $t \geq 1$, consider the survivor index set $\mathcal{I}_t \triangleq \{i : \ell_i \geq t\}$ with $n_t \triangleq |\mathcal{I}_t|$. If $n_t \leq 1$, skip this position. Otherwise define the empirical marginal

$$\hat{p}_t(v) \triangleq \frac{1}{n_t} \sum_{i \in \mathcal{I}_t} \mathbb{I}[x_{i,t} = v]$$

and its entropy (natural log)

$$H_t \triangleq -\sum_v \hat{p}_t(v) \log(\hat{p}_t(v) + \epsilon), \qquad \epsilon = 10^{-10}.$$

Let $\mathcal{T} \triangleq \{t : n_t > 1\}$ and report

$$\texttt{Entropy} \triangleq \frac{1}{|\mathcal{T}|} \sum_{t \in \mathcal{T}} H_t.$$

**Length-bucketed token entropy (`Entropy`($\ell$)).** Group valid samples by their pre-EOS length $\ell$ and compute `Entropy` on each fixed-length bucket after truncation to $\ell$. We set $\texttt{Entropy}(\ell) = 0$ if the bucket has $\leq 1$ sample or $\ell \leq 0$.

**Fingerprint diversity (`FPDiv`) for SMILES.** Let $f(x)$ be a fingerprint and $\text{sim}(x, x')$ a similarity (Tanimoto in our SMILES experiments). We report

$$\texttt{FPDiv} \triangleq 1 - \frac{2}{n_{\text{valid}}(n_{\text{valid}} - 1)} \sum_{\substack{x,x' \in \mathcal{D}_{\text{valid}} \\ x < x'}} \text{sim}(x, x'). \tag{16}$$

**Macro-averaged fingerprint diversity (`MacroFP`).** Given length bins $\mathcal{B}$ (e.g., 0–5, 6–10, 11+), let $\mathcal{D}_{\text{valid}}^{(b)}$ be the valid subset in bin $b$. Define

$$\texttt{MacroFP} \triangleq \frac{1}{|\mathcal{B}|} \sum_{b \in \mathcal{B}} \texttt{FPDiv}(\mathcal{D}_{\text{valid}}^{(b)}),$$

where $\texttt{FPDiv}(\cdot) = 0$ if a bin contains fewer than 2 samples.

### E.3. Prefix-collapse metrics

**Prefixes and survivors.** For a terminal $x = (x_1, \ldots, x_{\ell(x)})$, define its length-$k$ prefix $s_{0:k}(x) \triangleq (x_1, \ldots, x_k)$ for $k \leq \ell(x)$. At depth $k$, the valid survivor multiset is

$$\mathcal{D}_{\text{valid},k} \triangleq \{x \in \mathcal{D}_{\text{valid}} : \ell(x) \geq k\}, \qquad n_k \triangleq |\mathcal{D}_{\text{valid},k}|.$$

**Prefix survival (`Surv`($k$)).**

$$\texttt{Surv}(k) \triangleq \frac{n_k}{n_{\text{valid}}}.$$

**Prefix entropy (`PefEnt`$(k)$) and effective prefix count (`Eff`$(k)$).** Let $\hat{p}_k(p)$ be the empirical frequency of prefix $p$ among survivors:

$$\hat{p}_k(p) \triangleq \frac{1}{n_k} \sum_{x \in \mathcal{D}_{\text{valid},k}} \mathbb{I}[s_{0:k}(x) = p].$$

Define

$$\text{PefEnt}(k) \triangleq - \sum_p \hat{p}_k(p) \log \hat{p}_k(p), \qquad \text{Eff}(k) \triangleq \exp(\text{PefEnt}(k)).$$

**Top-1 prefix mass (`Top1`$(k)$).**

$$\text{Top1}(k) \triangleq \max_p \hat{p}_k(p).$$

**Unique prefix rate (`UniqueRate`$(k)$).**

$$\text{UniqueRate}(k) \triangleq \frac{|\{s_{0:k}(x) : x \in \mathcal{D}_{\text{valid},k}\}|}{n_k}.$$

### E.4. Distribution metrics for Expr24 (oracle reference)

For Expr24, valid means correct. Let $\mathcal{U}_{\text{valid}} \triangleq \{\text{tuple}(x) : x \in \mathcal{D}_{\text{valid}}\}$ be the set of unique valid sequences. Let the enumerated oracle set be $\mathcal{Y}^\star$. We report

$$\text{Unique}_{\checkmark} \triangleq |\mathcal{U}_{\text{valid}}|, \qquad \text{CovCount} \triangleq |\mathcal{U}_{\text{valid}} \cap \mathcal{Y}^\star|, \qquad \text{Cov} \triangleq \text{CovCount}/|\mathcal{Y}^\star|,$$

and the sampling-cap-normalized variant

$$\text{NormCov} \triangleq \text{CovCount}/\min(N, |\mathcal{Y}^\star|).$$

**Position-wise token marginals.** Given a multiset of sequences $\mathcal{S}$ (oracle or sampled), define survivors at 0-indexed position $t$: $\mathcal{S}_t \triangleq \{x \in \mathcal{S} : |x| > t\}$. Let $C_t(v)$ be the count of token $v$ at position $t$ among $\mathcal{S}_t$ and $Z_t \triangleq \sum_v C_t(v)$. If $Z_t > 0$, define the empirical marginal $q_t(v) \triangleq C_t(v)/Z_t$.

**Stabilized KL/JS.** Let $\epsilon = 10^{-9}$ and let $\mathcal{A}_t$ be the union support of oracle and sampled marginals at position $t$. Define

$$\text{KL}_\epsilon(p\|q) \triangleq \sum_{v \in \mathcal{A}_t} (p(v) + \epsilon) \log \frac{p(v) + \epsilon}{q(v) + \epsilon},$$

and $\text{JS}_\epsilon(p,q) \triangleq \frac{1}{2}\text{KL}_\epsilon(p\|m) + \frac{1}{2}\text{KL}_\epsilon(q\|m)$ where $m = \frac{1}{2}(p+q)$.

**Scalar divergences.** Let $p_t^\star$ be the oracle marginal at position $t$ computed from $\mathcal{Y}^\star$, and $\pi_t$ the sampled marginal at $t$ computed from $\mathcal{D}_{\text{valid}}$ (duplicates kept). Let $T_{\max}$ be the maximum length across oracle and sampled sequences. We report

$$\text{KL}(\pi \to p^\star) \triangleq \frac{1}{T_{\max}} \sum_{t=0}^{T_{\max}-1} \text{KL}_\epsilon(\pi_t \| p_t^\star), \quad \text{KL}(p^\star \to \pi) \triangleq \frac{1}{T_{\max}} \sum_{t=0}^{T_{\max}-1} \text{KL}_\epsilon(p_t^\star \| \pi_t),$$

$$\text{JS}_{\text{tok}} \triangleq \frac{1}{T_{\max}} \sum_{t=0}^{T_{\max}-1} \text{JS}_\epsilon(\pi_t, p_t^\star).$$

## F. Derivations and Objective Details

### F.1. Reference-prior reward shaping

We stabilize exploration by mixing (i) a frozen reference-LM prior as a log-regularizer and (ii) an external task score. Following Method 3.1, we represent the LLM-GFlowNet state by a generated prefix. Let $s_{0:k}$ denote the length-$k$ prefix state (with $s_{0:0} \equiv s_0$ as the root), and let termination occur by emitting $\top$ at $s_{0:\tau}$. Let $q_{\text{ref}}(\cdot \mid s_{0:k})$ be a frozen reference LM over $\mathcal{V} \cup \{\top\}$.

**Reference log-score at a stop cut.** We define the reference log-probability of stopping at $s_{0:k}$ as

$$\log P_{\text{ref}}(s_{0:k}^\top) \triangleq \sum_{t=0}^{k-1} \log q_{\text{ref}}(s_{t+1} \mid s_{0:t}) + \log q_{\text{ref}}(\top \mid s_{0:k}) \tag{17}$$

**Mixed stop-reward and task-only component.** Given an external task score $S(s_{0:k})$ (e.g., validity/QED/Expr-hit), we define the mixed log stop-reward

$$\log R(s_{0:k}^\top) \triangleq \kappa \log P_{\text{ref}}(s_{0:k}^\top) + \lambda S(s_{0:k}^\top), \tag{18}$$

where $\kappa$ is a fixed scaling (typically $\kappa=1$) and $\lambda$ sets the task term scale (empirically, $\lambda = 50$). Equivalently, define the *task-only* log component

$$u(s_{0:k}^\top) \triangleq \log R(s_{0:k}^\top) - \kappa \log P_{\text{ref}}(s_{0:k}^\top), \tag{19}$$

which is the part we "absorb" in RapTB (the reference-derived baseline remains intact).

**Ablation: removing the reference prior.** Table 28 shows that dropping the reference-prior term can severely destabilize training (here shown for TB on SMILES), causing sharp validity collapse and degenerate length behavior.

**Table 28. SMILES generation ablations (TB reference).** Unless specified, all metrics are computed on valid samples. `Len` denotes the mean token length of valid samples ($L_{\text{max}} = 10$).

| Method | Acc ↑ | Score ↑ | Entropy ↑ | FPDiv ↑ | Len |
|---|---|---|---|---|---|
| TB | **0.998** | **0.717** | **2.503** | **0.807** | 3.065 |
| TB w/o ref | 0.381 | 0.601 | 0.418 | 0.425 | 10.000 |

## F.2. Terminable prefix-tree specialization of TB

Following Method 3.1, $q_\theta(\cdot \mid s_{0:k})$ parametrizes the forward policy over $\mathcal{V} \cup \{\top\}$. For readability, write $q_\theta(s_{t+1} \mid \phi_t)$ for token actions and $q_\theta(\top \mid s_{0:k})$ for termination at prefix $s_{0:k}$. On the prefix tree, each non-root prefix has a unique parent, hence backward factors are deterministic and vanish in log-space.

**Terminal TB residual.** For a realized termination index $\tau$, the TB log-residual is

$$\Delta_{\text{TB}}(\xi) = \log Z_\theta + \sum_{t=0}^{\tau-1} \log q_\theta(s_{t+1} \mid s_{0:t}) + \log q_\theta(\top \mid s_{0:\tau}) - \log R(s_{0:\tau}^\top) \tag{20}$$

We minimize $\mathcal{L}_{\text{TB}} = \mathbb{E}_\xi[\Delta_{\text{TB}}(\xi)^2]$.

**Residual at an intermediate prefix.** In our implementation, the model outputs per-prefix stop logits $\log p_{\text{term}}[k] \equiv \log q_\theta(\top \mid s_{0:k})$ and per-prefix stop-reward logs $\log r[k] \equiv \log R(s_{0:k}^\top)$ for *all* $k \in \{0, \dots, L-1\}$, including $k = 0$ (stopping immediately after the prompt). Token-transition log-probabilities are stored as $\log p_F[t] \equiv \log q_\theta(s_{t+1} \mid s_{0:t})$ for steps $t \in \{0, \dots, L-2\}$.

Define the TB-style residual at prefix $s_{0:k}$ by:

$$\Delta_k^{\text{TB}}(\xi) \triangleq \log Z_\theta + \sum_{t=0}^{k-1} \log p_F[t] + \log p_{\text{term}}[k] - \log r[k], \qquad k \in \{0, \dots, L-1\}, \tag{21}$$

where $\Delta_0^{\text{TB}}(\xi) = \log Z_\theta + \log p_{\text{term}}[0] - \log r[0]$.

RapTB uses the *rooted* version (cancels $\log Z_\theta$):

$$\bar{\Delta}_k(\xi) \triangleq \Delta_k^{\text{TB}}(\xi) - \Delta_0^{\text{TB}}(\xi), \qquad k \geq 1. \tag{22}$$

### F.3. Absorbed suffix backups in RapTB

RapTB absorbs only the task-only component. The reference term stays unchanged. For a sampled trajectory $\xi = s_{0:\tau}$, define

$$u_k \triangleq \lambda S(s_{0:k}^\top).$$

**Finite horizon.** Let $K \in \{1, \ldots, L_{\max}\}$ be the auxiliary horizon cap. We define the auxiliary backup window end as $h \triangleq \min(\tau, K)$. We compute suffix targets using indices $j \in [k, h]$.

**Max and soft backups.** Define

$$u_k^{\max} \triangleq \max_{j \in [k,h]} u_j, \tag{23}$$

$$u_k^{soft} \triangleq \frac{1}{\beta} \log \sum_{j=k}^h \exp\Big(\beta u_j - \beta \rho \, (j-k)\Big), \qquad \beta > 0, \ \rho \geq 0, \tag{24}$$

$$u_k^{tgt} \triangleq \alpha \, u_k^{\max} + (1 - \alpha) \, u_k^{soft}, \qquad \alpha \in [0, 1]. \tag{25}$$

We compute $u_k^{soft}$ with LogSumExp trick to ensure numerical stability and prevent overflow.

### F.4. Practical details: prefix eligibility, scheduling, and stop gradients

**Eligibility.** We compute $\mathcal{L}_{\text{aux}}$ only for prefixes $k \in \{1, \ldots, h\}$. We skip very short prefixes. We set a minimum depth $k_{\min} \geq 1$.

**Absorb terms** In the main paper we use a mixed stop-reward

$$\log R(s_{0:k}^\top) = \kappa \log P_{\text{ref}}(s_{0:k}^\top) + u_k, \qquad u_k \triangleq \lambda S(s_{0:k}^\top).$$

Absorption modifies only the task-only term. It keeps the reference term fixed. For a given prefix $k$, define a surrogate stop-reward

$$\log \widetilde{R}(s_{0:k}^\top) \triangleq \kappa \log P_{\text{ref}}(s_{0:k}^\top) + u_k^{tgt}.$$

Let $\bar{\Delta}_k(\xi)$ be the rooted residual from Eq. (4). If we recompute the rooted residual at prefix $k$ using $\widetilde{R}$ (and keep the root term unchanged), we get

$$\bar{\Delta}_k^{Rap}(\xi) = \bar{\Delta}_k(\xi) + \big( \log R(s_{0:k}^\top) - \log \widetilde{R}(s_{0:k}^\top) \big)$$
$$= \bar{\Delta}_k(\xi) + \big( u_k - u_k^{tgt} \big).$$

**Absorption Correction with Distance Discounting.** For an eligible prefix at step $k$, we inject a correction term that pulls the trajectory flow towards the future target. This correction is discounted by the distance to the horizon $h$:

$$\bar{\Delta}_k^{Rap}(\xi) = \bar{\Delta}_k(\xi) + \gamma^{(h-k)} \cdot \big( u_k - u_k^{tgt} \big). \tag{26}$$

This ensures that credit assignment decays exponentially as the distance between the decision point and the realized outcome increases.

**Auxiliary loss.** For $k \leq h$, let $u_k^{tgt}$ be computed by Eq. (23). We use

$$\mathcal{L}_{\text{aux}}(\xi) \triangleq \frac{\sum_{k=1}^h w_k \left( \bar{\Delta}_k^{Rap}(\xi) \right)^2}{\sum_{k=1}^h w_k}. \tag{27}$$

**Stop-gradient on the termination head in the auxiliary branch.** We stop gradients through termination log-probabilities only in the auxiliary branch. Inside $\mathcal{L}_{\text{aux}}$, we replace

$$\log q_\theta(\top \mid s_{0:k}) \mapsto \text{stopgrad}(\log q_\theta(\top \mid s_{0:k})). \tag{28}$$

The terminal TB loss uses full gradients.

---

**Algorithm 1** RapTB auxiliary targets (single trajectory)

---

1: Roll out a terminable trajectory $\xi = (s_{0:0} \to \cdots \to s_{0:\tau})$.
2: Set $h$ as the end of the backup window.
3: Compute $u_k = \lambda S(s_{0:k}^\top)$ for $k \leq h$.
4: Compute $u_k^{\text{tgt}}$ via Eq. (23).
5: Compute $\bar{\Delta}_k(\xi)$ for eligible $k$.
6: In this branch, use $\text{stopgrad}(\log q_\theta(\top \mid s_{0:k}))$.
7: Apply the absorbed difference $u_k - u_k^{\text{tgt}}$ when absorption is enabled.
8: Compute $\mathcal{L}_{\text{aux}}(\xi)$ via Eq. (27).
9: Train with $\mathcal{L}_{\text{TB}} + \eta \mathcal{L}_{\text{aux}}$.

---

**Final objective.** We optimize

$$\mathcal{L}_{\text{RapTB}} \triangleq \mathbb{E}_{\xi \sim q_\theta^\top} \left[ \Delta^{\text{TB}}(\xi)^2 + \eta \, \mathcal{L}_{\text{aux}}(\xi) \right]. \tag{29}$$

### F.5. Variance reduction view of RapTB

**TB tail as a stochastic regression target.** Fix a prefix cut at index $m$ along a sampled terminable trajectory $\xi = (s_{0:0} \to s_{0:1} \to \cdots \to s_{0:\tau})$ on the prefix tree. Starting from the terminal TB residual in Eq. (1), we decompose it as

$$\Delta_{\text{TB}}(\xi) = X(s_{0:m}(\xi)) - Y_m(\xi), \tag{30}$$

where the *prefix term*

$$X(s_{0:m}(\xi)) \triangleq \log Z_\theta + \sum_{t=0}^{m-1} \log q_\theta(s_{t+1} \mid s_{0:t}), \tag{31}$$

is deterministic given the prefix $s_{0:m}$, and the *tail term*

$$Y_m(\xi) \triangleq \log R(s_{0:\tau}^\top) - \log q_\theta(\top \mid s_{0:\tau}) + \sum_{t=m}^{\tau-1} \log q_\theta(s_{t+1} \mid s_{0:t}), \tag{32}$$

depends only on the sampled suffix $(s_{0:m} \to \cdots \to s_{0:\tau})$.

Conditioning on $s_{0:m}$, TB minimizes a conditional least-squares error:

$$\mathbb{E}\left[ \left( X(s_{0:m}) - Y_m(\xi) \right)^2 \Big| s_{0:m} \right] = \left( X(s_{0:m}) - \mu(s_{0:m}) \right)^2 + \text{Var}(Y_m(\xi) \mid s_{0:m}), \tag{33}$$

where $\mu(s_{0:m}) \triangleq \mathbb{E}[Y_m(\xi) \mid s_{0:m}]$ is the minimum-MSE target for that prefix.

**Connecting to RapTB.** Eq. (33) highlights a core difficulty under terminal rewards: early prefixes are trained through a single stochastic tail target $Y_m(\xi)$ whose conditional variance can be large, so credit assignment to early decisions is noisy A natural response is to add subtrajectory consistency, but enforcing arbitrary-start windows introduces state-dependent boundary terms; in terminable prefix trees, the commonly used SubTB in LLM-GflowNets ties these boundaries to $-\log q_\theta(\top \mid s)$, so heterogeneous starts combined with discontinuous rewards can be absorbed by the shared termination head, inducing termination/length drift. RapTB takes a conservative middle ground: it keeps terminal TB unchanged as the global anchor, and adds only *root-start* residuals $\bar{\Delta}_k = \Delta_k^{\text{TB}} - \Delta_0^{\text{TB}}$ (Appendix F.2), providing $O(\tau)$ prefix-local supervision without introducing a separate flow head or arbitrary-start boundary heterogeneity. To further reduce variance in auxiliary targets, RapTB densifies only the *external* component of the stop-reward by absorbing high-reward evidence along the sampled suffix (Appendix F.3), while leaving any reference-derived baseline intact; this yields a lower-variance proxy for the reward term inside rooted constraints without changing the terminal TB semantics. Finally, we stop gradients through termination logits in the auxiliary branch so these additional prefix constraints cannot be satisfied by globally shifting $q_\theta(\top \mid s)$, preventing auxiliary-driven length bias.

## F.6. LLM-SubTB analysis

**SubTB on terminable prefix trees and the implicit local baseline.** We use the Subtrajectory Balance (SubTB) objective for autoregressive prefix trees (Madan et al., 2023; Hu et al., 2024). For any subtrajectory window indexed by $0 \leq i < j \leq \tau$ along a sampled trajectory $\xi = (s_0 \to \cdots \to s_\tau)$,

$$\Delta_{i \to j}^{\mathrm{SubTB}}(\xi) = \sum_{k=i}^{j-1} \log P_F^\theta(s_{k+1} \mid s_{0:k}) + \left( \log P_F^\theta(\top \mid s_{0:j}) - \log P_F^\theta(\top \mid s_{0:i}) \right) + \left( \log R(s_{0:i}^\top) - \log R(s_{0:j}^\top) \right), \quad (34)$$

and the per-trajectory objective aggregates all window residuals. This form can be viewed as eliminating the state-dependent flow/normalizer $F_\theta(\cdot)$ by using the terminable identity $R(s^\top) \approx F_\theta(s) P_F^\theta(\top \mid s)$ at optimum (Hu et al., 2024); on prefix trees $P_B \equiv 1$, so $\log P_F^\theta(\top \mid s)$ becomes the only learned term that can represent the missing *local baseline* across different start states.

**Termination drift.** In stop-reward LLM settings, rewards $R(s_{0:k}^\top)$ are often sparse and weakly prefix-dependent. When $\log R(s_{0:i}^\top) \approx \log R(s_{0:j}^\top)$ for many $(i, j)$, the reward-difference term in Eq. (34) carries little signal, and the window residual is dominated by the token-transition sum and the termination-logit difference $\log P_F^\theta(\top \mid s_{0:j}) - \log P_F^\theta(\top \mid s_{0:i})$. Crucially, each $\log P_F^\theta(\top \mid s_{0:t})$ participates in *many* windows (all windows spanning $t$), so adjusting the termination head provides a high-leverage way to reduce a large number of squared window residuals simultaneously. As a result, optimization can partially satisfy arbitrary-start consistency by shifting $\log P_F^\theta(\top \mid s)$, which miscalibrates stopping probabilities and manifests as length bias / termination drift.

**How RapTB blocks this failure channel.** RapTB prevents auxiliary consistency from being satisfied via termination-head drift in two complementary ways. First, it restricts auxiliary consistency to *rooted* (start-at-$i = 0$) prefixes, which avoids imposing many arbitrary-start windows that implicitly demand accurate local baselines for each intermediate start state. Second, in the auxiliary branch it stops gradients through $\log P_F^\theta(\top \mid s)$, so auxiliary constraints cannot be optimized by moving the termination head; length calibration is instead anchored by the terminal TB objective, while the auxiliary signal focuses on improving prefix credit assignment without inducing systematic termination shifts.

# G. Implementation Details and Reproducibility

**Config provenance.** The hyperparameters in Tables 29–32 are taken from the task configs used to produce our main results.

## G.1. Model, decoding, and context-free grammar (CFG)

**Backbone and parameterization.** We fine-tune an autoregressive LLM with a terminable action at every prefix state. The forward kernel is parameterized as in Sec. 3 by (i) a token head $p_F(\cdot \mid s)$ and (ii) a termination head $p_{\mathrm{term}}(s)$. We use parameter-efficient fine-tuning (LoRA) unless otherwise specified.

**LoRA fine-tuning details.** We use parameter-efficient fine-tuning with LoRA on the LLM backbone. Concretely, we apply LoRA to the attention and MLP projection modules {q_proj, k_proj, v_proj, o_proj, gate_proj, down_proj, up_proj} with rank $r=16$, $\alpha=16$, dropout 0.1, and no bias parameters. The backbone is meta-llama/Llama-3.2-1B. We optimize with AdamW (lr $10^{-4}$). Any auxiliary schedulers used in training (e.g., temperature/replay schedules) are reported explicitly in Table 32.

**Molecular fingerprints for similarity/diversity.** For SMILES similarity used by SubM, we compute RDKit Morgan fingerprints with radius 2 and 2048 bits, and use Tanimoto similarity for the facility-location diversity term. These settings are fixed across all SMILES experiments.

**Context-free grammar (CFG) constrained decoding.** We enforce syntactic constraints during generation using a grammar processor with an incremental EBNF parser. At each step, the parser consumes the current prefix and returns the set of next tokens that keep the prefix valid under the grammar. The processor helps decide tokens only, without masking or modifying their logits. This is a decoding-time feasibility filter shared by all objectives (TB/SubTB/RapTB), so it does not

change the training objective while substantially reducing invalid roll-outs. Figures 4–5 list the grammars used for SMILES and Expr24, respectively.

```
root ::= smiles

smiles ::= atom ( chain | branch )*

chain ::= (dot atom | bond? ( atom | ring_closure ) )+

branch ::= "(" ( ( dot | bond )? smiles )+ ")"

atom ::= organic_symbol | aromatic_symbol | atom_spec | wildcard

bond ::= "" | "=" | "#" | "$" | ":" | "@" | "@@"

dot ::= "."

wildcard ::=  "*"

atom_spec ::= "[" ( "se" | "as" | aromatic_symbol | element_symbol | wildcard ) chiral_class? h_count? ( charge | class? ) "]"

organic_symbol ::= "B" | "C" | "N" | "O" | "P" | "S" | "F" | "I" | "Br" | "Cl" | "At" | "Ts"

aromatic_symbol ::= "b" | "c" | "n" | "o" | "p" | "s"

element_symbol  ::= "A" ( "c" | "g" | "l" | "m" | "r" | "s" | "t" | "u" ) |
                    "B" ( "a" | "e" | "h" | "i" | "k" | "r" )? |
                    "C" ( "a" | "d" | "e" | "f" | "l" | "m" | "n" | "o" | "r" | "s" | "u" )? |
                    "D" ( "b" | "s" | "y" ) |
                    "E" ( "r" | "s" | "u" ) |
                    "F" ( "e" | "l" | "m" | "r" )? |
                    "G" ( "a" | "d" | "e" ) |
                    "H" ( "e" | "f" | "g" | "o" | "s" )? |
                    "I" ( "n" | "r" )? |
                    "K" "r"? |
                    "L" ( "a" | "i" | "r" | "u" | "v" ) |
                    "M" ( "c" | "g" | "n" | "o" | "t" ) |
                    "N" ( "a" | "b" | "d" | "e" | "h" | "i" | "o" | "p" )? |
                    "O" ( "g" | "s" )? |
                    "P" ( "a" | "b" | "d" | "m" | "o" | "r" | "t" | "u" )? |
                    "R" ( "a" | "b" | "e" | "f" | "g" | "h" | "n" | "u" ) |
                    "S" ( "b" | "c" | "e" | "g" | "i" | "m" | "n" | "r" )? |
                    "T" ( "a" | "b" | "c" | "e" | "h" | "i" | "l" | "m" | "s" ) |
                    "U" | "V" | "W" | "Xe" | "Y" "b"? |
                    "Z" ( "n" | "r" )

ring_closure ::= "%" [1-9] [0-9] | [0-9]

chiral_class ::= ( "@" ( "@" | "TH" [1-2] | "AL" [1-2] | "SP" [1-3] | "TB" ( "1" [0-9]? | "2" "0"? | [3-9] ) | "OH" ( "1"

[0-9]? | "2" [0-9]? | "3" "0"? | [4-9] ) )? )?

charge   ::= "-" ( "-" | "0" | "1" [0-5]? | [2-9] )? | "+" ( "+" | "0" | "1" [0-5]? | [2-9] )?

h_count   ::= "H" [0-9]?

class   ::= ":" [0-9]+
```

**Figure 4. EBNF grammar used for constrained SMILES decoding.**

```
root  ::= expr4
expr4 ::= num | num op num | num op num op num | num op num op num op num
| num op num op num op num op num | num op num op num op num op num op num
op    ::= "+" | "-" | "*" | "/"
num   ::= [0-9]
```

**Figure 5. EBNF grammar used for constrained Expr24 decoding.**

## G.2. SMILES constraints and validity evaluation

**Constrained decoding and legal token list.**  We apply the grammar processor at *every* decoding step. In addition, we use a fixed allowlist of tokenizer tokens deemed legal for SMILES generation. The combination of EBNF parsing and token allowed lists substantially reduces invalid generations, without changing the learning objective.

**Table 29. Core training and decoding hyperparameters.** TB and SubTB share all hyperparameters with RapTB except the loss definition (and RapTB-specific coefficients).

|  | SMILES | Expr24 |
|---|---|---|
| Trainer steps (max) | 5000 | 5000 |
| Precision | Bf16 | Bf16 |
| Grad clip | 0.5 | 0.5 |
| Grad accumulation | 4 | 4 |
| Optimizer | AdamW ($lr = 10^{-4}$) | AdamW ($lr = 10^{-4}$) |
| **Sampling (train)** | | |
| # trajectories per update ($n_{\text{samples}}$) | 32 | 32 |
| $p_F$ temperature mix | $T_{\text{hi}} : 1.5 \to 1.0;\ T_{\text{lo}} : 0.8 \to 1.0$ | $T_{\text{hi}} : 1.5 \to 1.0;\ T_{\text{lo}} : 0.8 \to 1.0$ |
| low-temp probability | 0.666 | 0.666 |
| replay mixture ratio | 0.7 | 0.7 |
| **CFG decoding (grammar)** | | |
| Min/max length | 1 / 10 (15) | 3 / 9 |

**Table 30. RapTB-specific hyperparameters.** $k_{\min}$ is linearly scheduled by training step.

|  | SMILES | Expr24 |
|---|---|---|
| Aux weight $\eta$ | 0.25 | 0.25 |
| Distance discount $\gamma$ | 0.99 | 0.99 |
| Detach $p_{\text{term}}$ in aux | True | True |
| Absorb gate threshold $\varepsilon_{\text{ab}}$ | $10^{-6}$ | $10^{-6}$ |
| Target mode | mix | mix |
| Mix weight $\alpha$ (max vs soft) | 0.5 | 0.8 |
| Soft backup $\beta$ | 5.0 | 3.0 |
| Distance penalty $\rho$ | 0.1 | 0.5 |
| $k_{\min}$ schedule | $5 \to 2$ (5000 steps) | $7 \to 3$ (5000 steps) |
| Aux horizon cap $K$ | $L_{\max}$ | $L_{\max}$ |

**Validity and scoring.** We use RDKit-based validation to identify valid SMILES and to compute the task reward (QED in our setup). Invalid generations receive an invalidity shaping schedule as configured in the reward module.

### G.3. CommonGen Diagnostic Subset Details

As discussed in Section 4.6, we utilize a fixed diagnostic subset to monitor optimization stability. The subset consists of 10 distinct concept-sets selected from the CommonGen validation split (Lin et al., 2020).

Table 31 lists the specific concept-sets used in our diagnostic experiments. Despite the uniform input size, these concepts cover diverse semantic domains (e.g., physical action, sports, indoor scenes), requiring the model to generate logically coherent sentences with natural termination points.

**Table 31. Diagnostic subset of CommonGen.** The table lists the 10 concept-sets used to monitor termination drift and prefix entropy. All sets require relational reasoning to link three distinct concepts into a coherent scenario.

| ID | Concept Set | Domain Context |
|---|---|---|
| 1 | {field, look, stand} | Outdoor / Observation |
| 2 | {kid, room, dance} | Indoor / Activity |
| 3 | {cat, pet, couch} | Domestic / Animal |
| 4 | {climb, building, side} | Urban / Action |
| 5 | {climb, wall, talk} | Social / Action |
| 6 | {drive, snow, car} | Travel / Environment |
| 7 | {talk, wear, phone} | Communication |
| 8 | {hockey, rink, team} | Sports |
| 9 | {ocean, surfer, surf} | Nature / Sports |
| 10 | {stair, jump, flight} | Motion |

**Reward Definition for CommonGen Task** In the CommonGen task, we define a composite reward function designed to guide the model toward three distinct goals: structural validity, concept coverage, and linguistic quality. First, to ensure grammatical correctness, we enforce strict structural constraints where a generated sequence is deemed valid only if it meets specific criteria—such as containing at least one verb, proper capitalization, and appropriate terminal punctuation. Second, to maximize concept coverage, we compute rewards based on concept coverage, while adding a rigorous lemma-based Hard Coverage Bonus if all target concepts are successfully included. Finally, to promote fluency and semantic alignment with the ground truth, we incorporate a quality-based shaping term that calculates the BLEU scores between the generated prefix and the reference sentence, serving as a step-wise proxy for generation quality relative to human-written text.

**Evaluation Protocol on Diagnostic Set.** For each concept set, we generate $N = 320$ samples using the learned policy. We record the raw logits of the termination head ($\log p_{\text{term}}$) at every step to visualize the drift reported in the main text. The small scale of this subset allows for this dense, step-wise instrumentation which would be storage-prohibitive on the full benchmark.

### G.4. Replay buffers

**Reward-prioritized replay buffer (RP).** We maintain a replay buffer keyed by the decoded prompt string. For each prompt, the buffer stores up to $B$ trajectories using a min-heap over the reward proxy, thus keeping the current top-$B$ items. Exact duplicates (identical decoded strings) are discarded. To prevent near-duplicate high-reward items from dominating the buffer, we additionally apply a near-duplicate filter using edit distance between the tokenized answers (excluding the termination token): a new item is rejected if it is too similar to an existing buffer item, unless it has a higher reward (or is explicitly forced to be added). When enabled, we use a small buffer-augmentation trick for forced/validated items to ensure they are preferred. In our configs, the per-prompt buffer capacity is $B = 200$, and the near-duplicate tolerance is $0.25$.

**Reward-prioritized replay training (PRT).** The buffer is sufficiently populated, and replay sampling follows a two-tier scheme inspired by Shen et al. (2023): An $\alpha$ fraction of each replay minibatch is sampled uniformly from the top-$\beta$ reward tier (i.e., the highest-reward $\lceil \beta|B| \rceil$ items), and the rest is sampled uniformly from the remaining items; sampling falls back

to uniform replay when prioritization is disabled or infeasible.

### G.4.1. SUBMODULAR REPLAY DETAILS

**Submodular replay (SubM): objective and selection.** SubM replaces the "keep top-$B$ by reward" rule with a *submodular subset selection* step that refreshes the buffer to maximize a weighted objective combining quality, diversity, and length coverage. for each candidate item $x$, we define a *static score* $s(x) = w_{\text{rew}} r(x) + w_{\text{val}}\mathbf{1}[\text{valid}(x)]$ and maximize a facility-location style diversity term together with a concave length-bin coverage term. Concretely, during greedy selection we use the per-candidate marginal gain

$$\Delta(x \mid S) = s(x) + w_{\text{div}} \sum_{u \in \mathcal{G}} \max\left\{0, \ \text{sim}(u,x) - \text{msim}(u,S)\right\} + w_{\text{len}}\alpha_{b(x)}\Big(\log(1 + c_{b(x)}(S)+1) - \log(1 + c_{b(x)}(S))\Big),$$
(35)

where $\text{sim}(\cdot,\cdot) \in [0,1]$ is a task-dependent similarity, $b(x)$ is the length-bin index of $x$, $c_b(S)$ is the current count in bin $b$, and $\alpha_b \geq 0$ controls how strongly we bias coverage toward specific length regimes. In our configs we use uniform weights, so $\alpha_b = 1$ for all bins. Importantly, the implementation uses token length rather than raw string length to define bins, avoiding the common mismatch between tokenizer length and character length.

**Similarity backends.** For SMILES, we canonicalize valid molecules with RDKit and compute Morgan fingerprints (radius 2, 2048 bits); similarity is Tanimoto computed by RDKit bulk routines. For Expr24 (string-domain tasks), we use $k$-gram shingles (default $k=2$) and Jaccard similarity between shingle sets. These similarities are only used by the diversity term in Eq. (35).

**Efficiency: cached coverage updates and greedy variants.** To compute the facility-location marginal efficiently, we maintain $\text{msim}(u,S)$ for each ground element $u \in \mathcal{G}$. When evaluating a candidate $x$, we compute $\{\text{sim}(u,x)\}_{u \in \mathcal{G}}$ via a bulk similarity call and accumulate $\sum_u \max\{0, \ \text{sim}(u,x) - \text{msim}(u,S)\}$, then update $\text{msim}(u,S) \leftarrow \max\{\text{msim}(u,S), \text{sim}(u,x)\}$ after selecting $x$. We use standard greedy algorithm in our submodular replay method.

**Validity gating for diversity.** SubM optionally restricts the diversity optimization to a valid-heavy candidate pool: it keeps all valid items and only adds enough invalid items (ranked by static score) to ensure the valid proportion is at least the specified ratio. This prevents invalid strings from consuming the diversity budget while still allowing the buffer to remain populated when valid samples are scarce. In SMILES we set the validity-gating ratio to $0.0$, while in Expr24 we set the ratio to $1.0$, so the facility-location term is computed only among valid items.

**Table 32. Factor schedulers used in our training runs.** Each factor uses linear interpolation from `start` to `end` over a fixed `horizon` (in steps).

| Task | Factor | start | end | horizon |
|------|--------|-------|-----|---------|
| SMILES | replay_buffer | 0.50 | 0.25 | 5000 |
| SMILES | k_min | 5 | 2 | 5000 |
| Expr24 | replay_buffer | 0.50 | 0.25 | 5000 |
| Expr24 | oracle_buffer | 0.75 | 0.25 | 5000 |
| Expr24 | k_min | 7 | 3 | 5000 |

**Table 33. Submodular replay hyperparameters (SMILES).** We select a buffer of size $B$ using greedy maximization of a facility-location + length objective.

| Hyperparameter | Value |
| --- | --- |
| Buffer capacity $B$ | 200 |
| Similarity backend | RDKit bulk Tanimoto (Morgan r=2, 2048-bit) |
| Weights $(w_{\mathrm{rew}}, w_{\mathrm{val}}, w_{\mathrm{div}}, w_{\mathrm{len}})$ | $(1.0, 1.0, 1.0, 1.0)$ |
| Length bin size | 1 token |
| Length alpha mode / power | uniform / 1.0 |
| Validity gating ratio | 0.0 |
| Selection strategy | standard greedy |

**Table 34. Submodular replay hyperparameters (Expr24).** We select a buffer of size $B$ using greedy maximization of a facility-location + length objective.

| Hyperparameter | Value |
| --- | --- |
| Buffer capacity $B$ | 200 |
| Refresh period $K_{\mathrm{buf}}$ | 1 |
| Similarity backend | $k$-gram shingles + Jaccard |
| Weights $(w_{\mathrm{rew}}, w_{\mathrm{val}}, w_{\mathrm{div}}, w_{\mathrm{len}})$ | $(1.0, 1.0, 1.0, 0.0)$ |
| Length bin size | - |
| Length alpha mode / power | - |
| Validity gating ratio | 1.0 |
| Selection strategy | standard greedy |

