# OpenReview forum: "Rooted Absorbed Prefix Trajectory Balance with Submodular Replay for GFlowNet Training"
_ICML.cc/2026/Conference — ICML 2026 regular_

### Official Review · Reviewer_QHmk · 2026-03-10

**Soundness:** 3
**Presentation:** 2
**Significance:** 2
**Originality:** 3
**Overall Recommendation:** 4
**Confidence:** 4

**Summary:**

The paper presents RapTB — a GFlowNet-based method for fine-tuning language models to approximate reward-proportional posteriors. It combines a modification of trajectory balance objective that provides supervision over all intermediate prefixes of a sequence, as well as a submodular strategy for picking samples from the replay buffer for optimization that promotes both high reward and diversity. Experimental evaluation shows problems of previous GFlowNet-based approaches, such as prefix collapse and length bias, and demonstrates the improved performance of the proposed method in comparison to them.

**Compliance With Llm Reviewing Policy:**

Affirmed.

**Final Justification:**

My concerns were resolved during the rebuttal.

**Key Questions For Authors:**

1. Could you please provide a more in-depth mathematical explanation of the proposed RapTB loss? Are the credit bounds (Equations 6, 7, 8) introduced as a means to avoid having a learnable $\log Z_\theta$ scalar in TB loss? Why are they chosen in such a way? Does reaching the global optimum of the loss guarantee that the model samples from the distribution of interest (line 102)?
2. In my opinion, a much simpler loss function that can be used as a baseline for ablation is just averaging the usual TB loss over all prefixes of the sequence (since every prefix can be considered a complete trajectory with an addition of a stop token). Is this design choice viable for the provided experiments? Will the proposed approach produce meaningfully different results from it?

**Limitations:**

No concern.

**Strengths And Weaknesses:**

**Strengths:**

1. Investigating GFlowNet-based approaches for fine-tuning LLMs is a highly interesting and relevant topic to the research community.
2. The paper presents an in-depth experimental evaluation and ablation studies across a number of tasks, demonstrating failure modes of previous approaches based on TB and SubTB objectives, as well as showing empirical improvements of the proposed approach across various metrics. I found it well-crafted and insightful.

**Weaknesses:**

1. The paper fails to put itself in the broader context of RL literature related to GFlowNets and LLM fine-tuning. The authors state: "In contrast to reward-maximizing reinforcement learning, the objective of GFlowNets is distributional: spread probability mass across many high-reward modes in proportion to reward, rather than concentrating on a single optimum." While this view was shared by some of the earlier works on GFlowNets, it was shown that GFlowNet training can be equivalently reformulated as an RL problem [1, 2]. Moreover, it is well-known that sampling from the reward distribution defined as a product of pre-trained model and and a reward function (which is the case studied in the paper) is equivalent to fine-tuning a pre-trained model to solve a KL-regularized RL problem, see e.g. Equation 1 and Equation 2 in [3]. While I am not an expert in LLMs, up to my knowledge this is the standard way to do RL fine-tuning in modern LLM literature. I believe that it is crucial that the problem being solved and the contributions of the paper are properly put in this context.
2. Following the previous point, I believe that it will be great to add some standard RL approaches like PPO and GRPO as baselines to the experimental evaluation in the paper.
3. An important baseline from GFlowNet literature is missing: TBA [3]. This paper also uses trajectory balance objective to fine-tune language models, and also utilizes a replay buffer with a specific sampling strategy. This is the most direct competitor from previous literature, which follows a very similar training pipeline to the approach proposed by the authors, this I believe it should also be added to experimental evaluation and discussed in the paper.
4. I believe that the section presenting the proposed RapTB loss should be expanded. When reading the text, I struggled to understand the design choices behind the proposed training objective. A detailed mathematical derivation would be very helpful in my opinion (see Questions).
5. A more minor point, but the paper uses some terminology across the text without timely explaining its meaning, or referencing some previous works where it is defined. For example, *termination drift* is mentioned across different parts of the paper, but its meaning is only explained in the experiments section, which complicates readability.

The paper presents interesting contributions, but proper contextualization and framing within previous literature, as well as experimental comparison to important missing baselines, is necessary to recommend acceptance in my opinion. I was thinking between reject and weak reject for the score, but decided to settle on reject for the time being. I am willing to increase my score if my concerns are addressed.

*References*: \
[1] Generative Flow Networks as Entropy-Regularized RL. Daniil Tiapkin, Nikita Morozov, Alexey Naumov, Dmitry Vetrov. AISTATS 2024. \
[2] Discrete Probabilistic Inference as Control in Multi-path Environments. Tristan Deleu, Padideh Nouri, Nikolay Malkin, Doina Precup, Yoshua Bengio. UAI 2024. \
[3] Trajectory Balance with Asynchrony: Decoupling Exploration and Learning for Fast, Scalable LLM Post-Training. Brian Bartoldson, Siddarth Venkatraman, James Diffenderfer, Moksh Jain, Tal Ben-Nun, Seanie Lee, Minsu Kim, Johan Obando-Ceron, Yoshua Bengio, Bhavya Kailkhura. NeurIPS 2025.

---

> ### Author Rebuttal · Authors · 2026-03-31
>
> We thank the reviewer for the feedback. We respond to all concerns below.
>
> ## W1: RL literature contextualization.
>
> We agree that our framing is insufficiently contextualized in the broader RL literature. The GFlowNet target distribution admits a KL-regularized / MaxEnt RL formulation [1, 2]. Our contribution is more narrowly scoped yet consistent with both perspectives—whether one views GFlowNets as distinct generative models or as a form of entropy-regularized RL: within off-policy TB-family training, we improve prefix-level credit assignment on terminable prefix trees.
>
> We will revise the introduction to remove language that positions GFlowNets "in contrast to RL" and instead present them as a specialized off-policy, entropy-regularized RL regime in which the balance conditions yield structured loss functions. We appreciate your helpful suggestions.
>
> ---
>
> ## W2+W3: PPO/GRPO and TBA baselines.
>
> ### 1. PPO/GRPO
>
> Due to space constraints, we kindly refer you to **Reviewer Pd1v**’s **W2 section** for details.
>
> ### 2. TBA
> We agree that TBA [3] is the closest concurrent systems-level prior work and should be discussed explicitly. At the learning-objective level, however, TBA still optimizes the TB loss; its main contributions are asynchronous decoupling of exploration and learning together with importance-weighted replay. Our paper studies a different aspect: the balance objective itself. For this reason, the direct objective-level comparator for RapTB is TB under the same replay pipeline. We do not claim RapTB is a replacement for TBA as a post-training system; rather, it is orthogonal and can, in principle, be inserted into TBA’s framework in place of the TB loss. We will revise the related-work discussion to make this distinction explicit and cite TBA as the closest concurrent framework.
>
> ---
>
> ## W4+W5+Q1: Mathematical explanation of RapTB and global optimum.
> We appreciate the request for a deeper walkthrough. We explain each design choice below, then address the global optimum question.
>
> ### 1. Mathematical Explanation
>
> **(1) Rooted residuals and the role of $Z_{\theta}$.**
> The learnable $\log Z_{\theta}$ remains in the TB anchor $\mathcal{L}_{\text{TB}}$, which is the sole exact balance condition. In the auxiliary branch, the rooted residual $\bar{\Delta}_k = \Delta_k^{\text{TB}} - \Delta_0^{\text{TB}}$ cancels the shared partition term, so that per-prefix gradient updates do not redundantly reoptimize a global scalar. Without this cancellation, every prefix position would push the partition function in potentially conflicting directions, adding noise.
>
> **(2) Absorbed suffix targets.**
> The credit bounds in Eqs. (6–8) are therefore not introduced as a substitute for $\log Z_{\theta}$​​. Their role is to define informative and controlled prefix targets under sparse rewards. In our tasks, the immediate stop reward for an intermediate prefix is often zero or uninformative (e.g., incomplete expressions or incomplete SMILES). Supervising only prefixes with this signal yields few training signals (Appendix C.5). Further, we introduce absorbed suffix targets that look ahead along the sampled trajectory to provide an informative signal. $u_k^{\max}$ is a conservative hindsight target along a sampled suffix. $u_k^{\text{soft}}$ provides a long-horizon training signal, structurally similar to GAE [4].
>
> ### 2. Global Optimum
>
> We acknowledge that the current paper does not provide a formal proof. To address this limitation, we provide a decomposition-based analysis from two complementary perspectives. Due to space constraints, we respectfully refer you to **Reviewer Pd1v**’s **W3 Section**.
>
> ## Q2: AvgPrefixTB baseline.
>
> We implemented the reviewer-suggested **AvgPrefixTB**, which applies the TB loss uniformly across all prefixes.
>
> ### 1. SMILES generation
>
> | Method | Acc ↑ | Score ↑ | Entropy ↑ | Avg Len |
> |-|-:|-:|-:|--:|
> | RapTB| 0.996|0.740| 2.448| 6.14 |
> | AvgPrefixTB|1.000|0.661| 0.665| 2.89 |
> | RapTB+SubM|0.988|0.844|2.726|7.44 |
>
> ### 2. VarExpr24
>
> | Method | Acc ↑ | NormCov ↑ |
> |-|-:|-:|
> | RapTB| 0.991 | 0.039 |
> | AvgPrefixTB| 0.998 | 0.016 |
>
> AvgPrefixTB favors short sequences: in SMILES, the average length is **2.89**, with **54%** of mass on length 1~2. Although AvgPrefixTB improves diversity over TB on Expr24 (**142.0** unique vs. **5.3**), it still remains well below RapTB (**246.7** unique; NormCov **0.016** vs. **0.039**).
>
> This behavior suggests that averaging TB residuals across all prefixes encourages a shortcut: concentrating probability on short, high-reward trajectories that minimize loss cheaply.
>
> References
>
> [1] Tiapkin et al. "GFlowNets as Entropy-Regularized RL." AISTATS 2024.
>
> [2] Tristan et al. "Discrete Probabilistic Inference as Control in Multi-path Environments" UAI 2024.
>
> [3] Bartoldson et al. "Trajectory Balance with Asynchrony." NeurIPS 2025.
>
> [4] Schulman et al. “High-Dimensional Continuous Control Using Generalized Advantage Estimation” ICLR 2016.

---

> > ### Author Rebuttal · Reviewer_QHmk · 2026-04-04
> >
> > I thank the authors for the detailed rebuttal. I believe that the authors adequately resolved all the concerns I mentioned in my review, so it would be fair to raise my score. I strongly encourage the authors to include proper RL contextualization, additional experimental comparisons and the deeper theoretical discussions presented during the rebuttal in the revision of the paper. However, I think that this is a borderline case in regard to whether the amount of the mentioned revisions should be allowed in a camera-ready version of the accepted paper, or should require resubmission to a different venue and another round of review. I think that this should be left at the discretion of AC, and personally would not mind if the paper is either accepted or rejected.

---

> > > ### Author Response · Authors · 2026-04-04
> > >
> > > We thank the reviewer for the positive reassessment and raising the score. We are glad that our rebuttal addressed the concerns. In the camera-ready version, we will incorporate the suggested RL contextualization, additional comparisons, and theoretical discussions.
> > >
> > > Yet, we also note that these revisions are supplementary, as they elaborate on points already implicit in the draft, without changing the core contributions.
> > >
> > > We thank the reviewer again for the constructive and fair evaluation.

---

### Official Review · Reviewer_JxzD · 2026-03-12

**Soundness:** 2
**Presentation:** 3
**Significance:** 3
**Originality:** 2
**Overall Recommendation:** 4
**Confidence:** 4

**Summary:**

The authors propose a new objective and replay buffer to GFN training to improve credit assignment and address mode collapse. Instead of considering constraints on all subtrajectories, RapTB only considers trajectories rooted at $s_0. In addition, it provides an additional absorbed suffix reward target consisting of an interpolation of a hard/soft operator applied to downstream rewards. Finally, to address the mode collapse issue, they propose selecting a buffer that maximizes a submodular function which uses a mix of a quality term, a diversity term and a length diversity term.

Experimentally, the authors finetune Llama-3.2-1B for a variety of sequence generation tasks (SMILES, arithmetic and sentence generation). They find that RapTB achieves better diversity while maintaining similar quality in most task. Particularly, the length diversity and prefix diversity of RapTB is higher.

**Compliance With Llm Reviewing Policy:**

Affirmed.

**Final Justification:**

While I was considering increasing my score given the strong experimental evidence, the theoretical aspects of the paper remain under discussed.

In particular, my final theoretical concern/question remains unaddressed. As such, I am maintaining my rating.

**Key Questions For Authors:**

- I am a bit confused about the issue of overlapping constraints in SubTB. My understanding was that the SubTB constraint related flow at a state $s_i$ to flow at a state $s_j$ through the intermediate probabilities. Why are the termination probabilities included for each subtrajectory?
(I see this is explained in the appendix, have you tested with learning the state flow values and if this is still an issue?)
- Could you explain the prefix survival metric in Fig.3.?
- Have you looked at longer sequence generation tasks (AMP/GFP) which are typically tested by GFNs?
- The finetuning of a LLM for these tasks is interesting, what is the reason for this and have you also tested training a smaller model from scratch?

**Limitations:**

yes

**Strengths And Weaknesses:**

**Strengths**
- The paper proposes a series of novel solutions to address issues with GFN training
- The author train a large scale model on a variety of tasks. The experimental results are promising and indicate that their method does provide a meaningful improvement to diversity while maintaining or improving quality.
- The authors ablate the different additional components of their method, identifying the impact of the changes
- The submodular replay is clever and generally applicable (e.g. by varying the objective)

**Weaknesses**
- There is a lack of analysis of the new objective and whether it will converge to the global optimal.
- The absorbed suffix backups should be explained and motivated more clearly in the main text. It is unclear to me why it is a good idea to mix the intermediate task rewards with some sort of interpolation between a hard and soft operator.
- The methods makes many design choices and requires multiple new hyperparameters. It would interesting to examine robustness to these hyperparameters or have a good way to set them.

I am willing to increase my score if my questions are answered.

---

> ### Author Rebuttal · Authors · 2026-03-31
>
> We thank the Reviewer for the insightful questions.
>
> ## W1: Global optimality of RapTB
> We acknowledge that the current paper does not provide a formal proof for preservation of a reward-proportional terminal distribution. To address this limitation, we present a proof by decomposition that analyzes the problem from two complementary perspectives. Owing to space constraints, we respectfully refer the Reviewer to our response in the **W3 section** of Reviewer **Pd1v**.
>
> ---
>
>
> ## W2 + Q1: SubTB termination mechanism and absorbed suffix targets
>
> The Reviewer's question about SubTB and absorbed targets is central to our motivation.
>
> ### Explanation of SubTB in LLM setting
>
> 1. **Original SubTB** (Madan et al., 2023 [1]) uses a separate learned network, $F(s;\theta)$, to estimate state flow.
>
> 2. **In the LLM-GFlowNet formulation** (Hu et al., ICLR 2024 [2]), the prefix tree satisfies $P_B = 1$ because each state has a unique parent. Since every prefix can terminate, the substitution
>    $$
>    F(s) = \frac{R(s^\top)}{P_F(\top \mid s)}
>    $$
>    from Deleu et al. (UAI 2022 [3]) removes the need for explicit state-flow estimates.
>
> After this substitution, the termination probabilities appear at the endpoints of every subtrajectory. Thus, they are not additional constraints beyond SubTB; they are simply the result of eliminating the explicit state-flow variables in the all-prefix-terminable setting. This is precisely why an additional flow network is not required in LLM-GFlowNet. We have not separately evaluated a variant that learns an explicit state-flow head in this work; our implementation follows the standard LLM-GFlowNet parameterization. We believe this issue could, in principle, be addressed by learning state flows. However, doing so would substantially increase training complexity and cost: it requires an additional value-prediction head during model fine-tuning, as well as supervision over $O(N^2)$ subtrajectories.
>
> ### Motivation for absorbed suffix targets
>
> For some tasks, the reward $u_k = \log R(s_{0:k}^\top)$ for most prefixes is sparse (e.g., only complete valid expressions score nonzero in Expr24; incomplete SMILES strings do not receive a QED reward). This makes $u_k$ alone uninformative as a regression target for prefix $k$. We introduce absorbed suffix targets that look ahead along the sampled trajectory to provide an informative signal. We also provide analysis from a variance-reduction perspective; see Appendix C.5. We summarize the absorbing strategy below:
>
> | Target  | Role |
> |-|-|
> | $u_k^{\max}$ | Conservative: best observed downstream signal |
> | $u_k^{\text{soft}}$  | Smoothed: distance-discounted log-sum-exp |
>
> We use the mixed target to retain the strong look-ahead signal of $u_k^{\max}$, while reducing its tendency to overestimate prefix quality by incorporating the longer-horizon, smoothed signal from $u_k^{\text{soft}}$. Table 6 confirms that the mix strategy outperforms either endpoint.
>
> ## W3: Hyperparameters/robustness
> Owing to space constraints, we kindly refer the Reviewer to **Reviewer cA3o**’s **W3 + Q2 section** for details.
>
> ## Q2: Prefix survival explanation
>
> $$
> \mathrm{Surv}(k) = \frac{n_k}{n_{\mathrm{valid}}},
> $$
>
> where $n_k$ counts sequences that survive to position $k$.
>
> Joint interpretation with entropy and top-1 probability:
>
> |Pattern|Interpretation|
> |-|-|
> |Low survival|Premature stopping|
> |High survival + high top-1|Long sequences but share nearly identical prefix |
> |High survival + low top-1|Healthy branching|
>
> Figure 3 visualizes this: TB exhibits collapse (survival drops sharply, top-1 dominates), while RapTB achieves genuine branching across the prefix tree.
>
>
> ## Q3: Longer sequence tasks
>
> We also conduct an additional experiment on the AMP generation task. Due to space limitations, we kindly refer the Reviewer to Reviewer **Pd1v**'s **W1 section**.
>
> ---
>
> ## Q4: Why fine-tune an LLM instead of training a smaller model from scratch?
>
> Our work focuses on LLM-GFlowNet post-training rather than on from-scratch GFlowNet training. Following Hu et al. [2], we consider an autoregressive pretrained model with an explicit EOS action and a frozen reference-policy prior, and ask how TB-family objectives behave in that setting. This design is also practically motivated. For tasks such as SMILES generation, a pretrained LLM carries rich semantic and structural knowledge acquired from large-scale scientific and chemistry-related text. Such pretrained knowledge can be beneficial for molecular generation. That's why we did not train a smaller model from scratch in this paper.
>
> ## References
>
> [1] Madan et al. "Learning GFlowNets from partial episodes for improved convergence and stability." ICML 2023.
>
> [2] Hu et al. "Amortizing intractable inference in large language models." ICLR 2024.
>
> [3] Deleu et al. “ Bayesian structure learning with generative flow networks.” UAI 2022.
>
> [4] Schulman et al. “High-Dimensional Continuous Control Using Generalized Advantage Estimation” ICLR 2016.

---

> > ### Author Rebuttal · Reviewer_JxzD · 2026-04-01
> >
> > I thank the authors for their thorough rebuttal. My questions/concerns have been addressed regarding the use of a flow model in SubTB, the absorbed suffix targets, hyperparameter robustness, prefix survival and LLM training.
> >
> > My only remaining issue is regarding the global optima claims. If I am not mistaken, $u_k − u_k^{tgt} \neq 0$. As a result, in Eq. 8, $\Delta_k(\xi) = 0$ would imply that $L_{\mathrm{aux}} > 0$ and I don't believe we have this equivalency of global optima. Could this be clarified?
> >
> > To be clear, I don't think this is necessarily bad as it is unclear that to me the GFN objective is optimal for scientific discovery tasks. However, the theoretical properties of the method need to be clarified.

---

> > > ### Author Response · Authors · 2026-04-02
> > >
> > > **Response to the global-optimum question.**
> > >
> > > Thank you for this question. The key clarification is: **TB remains the only exact anchor in our LLM setting; RapTB adds prefix-level regularization on top of it.**
> > >
> > > In our formulation, the LLM can choose to terminate at any prefix length $k$ by emitting a stop token. Each early-stop at length $k$ produces a distinct terminal. The full terminal set covers all possible generation lengths — not just maximum-length outputs:
> > >
> > > $$\mathcal{X}=\{s_{0:k}^{\top} : 0\le k\le K\}.$$
> > >
> > > ---
> > >
> > > ### (1) TB Is the Exact Anchor
> > >
> > > Since the graph is a directed prefix tree with unique backward paths, the TB optimum requires reward-proportional sampling over all terminals. Substituting the autoregressive decomposition yields exactly our prefix TB residual:
> > >
> > > $$\Delta_k^{\mathrm{TB}}(\xi) = \log Z_\theta + \sum_{t=0}^{k-1}\log q_\theta(s_{t+1}\mid s_{0:t}) + \log q_\theta(\top\mid s_{0:k}) - \log R(s_{0:k}^{\top}) = 0, \quad \forall x_k\in\mathcal{X}.$$
> > >
> > > On a tree, the TB-optimal policy is uniquely determined, where $sa$ denotes the child prefix formed by appending token $a$ to $s$:
> > >
> > > $$q^\star(\top\mid s)=\frac{R(s^{\top})}{F^\star(s)}, \qquad q^\star(a\mid s)=\frac{F^\star(sa)}{F^\star(s)}, \qquad F^\star(s)=\sum_{x\in\mathcal{X}(s)} R(x).$$
> > >
> > > Thus, TB fixes how probability should be allocated across prefixes and stop decisions.
> > >
> > > ---
> > >
> > > ### (2) SubTB Motivates RapTB
> > >
> > > The SubTB residual on window $[i,j]$ decomposes as:
> > >
> > > $$\Delta_{i\to j}^{\mathrm{SubTB}}(\xi) = \Delta_j^{\mathrm{TB}}(\xi)-\Delta_i^{\mathrm{TB}}(\xi).$$
> > >
> > > Our rooted residual is the root-start special case:
> > >
> > > $$\bar\Delta_k(\xi) = \Delta_{0 \to k}^{\mathrm{SubTB}}(\xi).$$
> > >
> > > At an exact TB optimum all prefix residuals vanish, so every rooted residual also vanishes. Hence the **rooted/no-absorb objective is compatible with the ideal TB optimum**.
> > >
> > > RapTB keeps TB as the anchor but uses a selective rooted auxiliary to improve credit assignment, avoiding the full overlapping SubTB constraints that can destabilize optimization.
> > >
> > > ---
> > >
> > > ### (3) Absorbed RapTB Is Regularization
> > >
> > > With reward absorption, the residual becomes:
> > >
> > > $$\bar{\Delta}_k^{\mathrm{Rap}}(\xi) = \bar{\Delta}_k(\xi) + \gamma^{h-k}(u_k - u_k^{\mathrm{tgt}}).$$
> > >
> > > The absorbed target is a mixture of max and softmax suffix backups:
> > >
> > > $$u_k^{\mathrm{tgt}}=\alpha u_k^{\max}+(1-\alpha) u_k^{\mathrm{soft}}.$$
> > >
> > > Since the suffix set always includes the self term, we have:
> > >
> > > $$u_k^{\mathrm{tgt}}\ge u_k,$$
> > >
> > > with strict inequality when a later suffix carries stronger signal. Therefore, even at an exact TB optimum where the rooted residual vanishes, the absorbed residual is generally nonzero.
> > >
> > > Now consider the combined loss:
> > >
> > > $$L_{\mathrm{RapTB}}=L_{\mathrm{TB}}+\eta L_{\mathrm{aux}}.$$
> > >
> > > Let $\bar{\theta}$ denote an exact TB solution satisfying $L_{\mathrm{TB}}(\bar{\theta})=0$, which incurs a finite absorbed auxiliary cost $L_{\mathrm{aux}}(\bar{\theta})=B$. Let $\theta'$ be any global minimizer of $L_{\mathrm{RapTB}}$. Since $L_{\mathrm{TB}}\ge 0$, we have:
> > >
> > > $$L_{\mathrm{TB}}(\theta') \le L_{\mathrm{RapTB}}(\theta') \le L_{\mathrm{RapTB}}(\bar{\theta}) = 0 + \eta B = \eta B.$$
> > >
> > > The first inequality holds because $L_{\mathrm{TB}}$ is a component of $L_{\mathrm{RapTB}}$; the second holds because $\theta'$ minimizes $L_{\mathrm{RapTB}}$ globally. This means the TB loss at the RapTB optimum is controlled by $\eta B$: as $\eta \to 0$, exact TB balance is recovered. No-absorb yields $B=0$, where the two objectives share the same optimum.
> > >
> > >
> > > ---
> > >
> > > ### (4) Correct Scope of the Claim
> > >
> > > The correct claim is an **"exact anchor + auxiliary regularization"** design:
> > >
> > > - **TB** is the only exact-balance condition matching reward-proportional targets over all terminals.
> > > - **Rooted/no-absorb RapTB** is compatible with the TB optimum ($B=0$).
> > > - **Absorbed RapTB** is a biased but lower-variance regularizer ($B>0$), with the auxiliary branch not directly optimizing the termination logit.
> > >
> > > Empirically, removing absorption degrades RapTB: Score 0.740 to 0.716, FPDiv 0.860 to 0.805, Entropy 2.448 to 2.031. The mixed target gives the best overall balance (Table 6).
> > >
> > > We thank the reviewer again for this insightful question. We hope the above clarification fully addresses your concern, and we are happy to discuss further if needed and allowed :)

---

### Official Review · Reviewer_cA3o · 2026-03-15

**Soundness:** 3
**Presentation:** 3
**Significance:** 4
**Originality:** 4
**Overall Recommendation:** 5
**Confidence:** 4

**Summary:**

Here, the authors a method, RapTB, that proposes two complementary components for training LLM-GFlowNets on terminable prefix trees. First, RapTB augments the standard terminal Trajectory Balance (TB) objective with O(N) rooted prefix constraints that propagate terminal reward to intermediate prefixes via suffix-absorbed backups (a convex combination of max and soft log-sum-exp aggregations over the suffix), while stopping gradients through the termination head in the auxiliary branch to prevent the termination drift observed in SubTB. Second, Submodular Replay (SubM) replaces reward-prioritized replay with greedy maximization of a facility-location + length-bin coverage submodular objective over the union of the current buffer and a new batch. The method is evaluated primarily on scaffold-conditioned SMILES generation (QED), with supporting experiments on Expr24 arithmetic generation and a small CommonGen diagnostic subset (10 concept-sets, N=320 samples each). Their main claim is that RapTB+SubM simultaneously improves reward quality, molecular diversity, and long-horizon coverage relative to TB and SubTB, while maintaining high chemical validty.

**Compliance With Llm Reviewing Policy:**

Affirmed.

**Final Justification:**

The authors have done an excellent job responding to my concerns, and overall the paper is in strong shape for acceptance. I am especially excited to see this method be applied for molecular design.

**Key Questions For Authors:**

1. Table 3 shows TB+SubM achieves NormCov of 0.100 on Expr24 while RapTB without SubM reaches only 0.039 under RP --  SubM alone outperforms RapTB alone on coverage by a large margin. Under what conditions does RapTB provide meaningful additive benefit over SubM alone, and is there a task or regime where RapTB without SubM is the dominant contributor?

2. RapTB introduces 7 hyperparameters (eta, gamma, alpha, beta, rho, k_min schedule, K) that differ between SMILES and Expr24 (e.g., alpha=0.5 vs. 0.8, rho=0.1 vs. 0.5). How sensitive are the main results in Tables 1 and 3 to these choices, particularly beta and rho, which directly control the variance-reduction behavior central to the method's theoretical motivation?

3. All primary results use a single model (Llama-3.2-1B, LoRA rank 16), a single molecular domain (QED), and sequence lengths of at most 15 tokens. Can the authors clarify for me whether RapTB's gains over TB hold on longer sequences, larger vocabularies, or non-SMILES domains such as amino acid or nucleotide generation with non-differentiable reward functions?

4. The rooted residual and absorbed suffix backup are structurally analogous to advantage estimation with multi-step returns (GAE) in actor-critic methods. Can the authors let me know if RapTB equivalent to a particular GAE estimator instantiated within the GFlowNet framework, or does the GFlowNet partition function constraint introduce a substantive difference?

**Limitations:**

Yes.

**Strengths And Weaknesses:**

### Strengths

**Soundness.** For me, the mechanistic diagnosis of SubTB's failure on terminable prefix trees is defintiely their strongest technical contribution. The analysis in Appendix C.6 looks correct: in the SubTB window residual (Eq. 33), when rewards are sparse or weakly prefix-dependent, the difference term carries the dominant gradient signal, and because each termination logit participates in O(N²) windows simultaneously, the optimizer can reduce the aggregate squared loss by globally shifting the termination head rather than improving token-level transitions. This is a reproducible failure mode; they nicely demonstrated this in Table 4 (SubTB log p_term = −79.638 under RP, −86.415 under Oracle) and Table 5 (SubTB delta log p_term = −28.32, saturating length at 20.00 on CommonGen). The ablation in Table 4 confirms this, where restricting SubTB to rooted windows and reintroducing Z_theta recovers accuracy to ~100%, directly validating the diagnosis.

**Originality.** The absorbed suffix backup (Eqs. 22–24) is a super clean variance-reduction method. Replacing the terminal stop-reward with a surrogate prefix credit is well-motivated by the variance decomposition in Appendix C.5 (Eq. 32), and the exponential distance discounting in Eq. (25) is a nice way to downweight distant suffix evidence. The combination of this with gradient stopping on the termination head in the auxiliary branch is a minima fix that directly addresses the identified failure channel.

**Presentation.** Figure 1 and Figure 3 are definitely the paper's clearest justifications. Figure 1 makes the O(1)/O(N^2)/O(N) complexity comparison between TB/SubTB/RapTB immediately legible alongside qualitative SMILES samples. Figure 3's prefix-collapse diagnostics (survival, entropy, top-1 mass by prefix depth) are (and underused) evaluation methodsthat the paper uses well to distinguish between terminal diversity and genuine prefix-level branching.

### Weaknesses

1. The paper's primary experimental setting is scaffold-conditioned SMILES generation with a fixed vocabulary (the EBNF grammar in Figure 4) and a single scalar reward (QED). The Expr24 task uses a minimal vocabulary with a binary sparse reward. CommonGen uses 10 concept-sets with N=320 samples, which is explicitly described as a "diagnostic subset" rather than a benchmark. This means the paper's empirical support for RapTB as a general LLM-GFlowNet training objective rests almost entirely on one domain (molecular SMILES) with one reward (QED) and one model (Llama-3.2-1B with LoRA rank 16). The paper's claims, however, are stated at the level of terminable LLM-GFlowNets generally. The failure modes identified (prefix collapse and termination drift) arise from the structure of the prefix tree and the SubTB objective, not from anything specific to SMILES. Domains that would test these claims more directly include: biological sequence generation with unique vocabularies (amino acids, nucleotides) and non-differentiable reward functions; code generation with structural validity constraints and functional correctness rewards; or mathematical reasoning with verifiable sparse rewards analogous to Expr24 but at longer horizons and larger output spaces. None of these are evaluated. The practical significance of the method is therefore difficult for me to assess beyond the molecular generation setting.

2. Table 3 (Expr24) shows that SubM applied to TB alone (TB+SubM) achieves NormCov=0.100 and Unique=642.0, while RapTB without SubM achieves NormCov of 0.039 and Unique=246.7 under RP. So in other words, SubM on top of vanilla TB outperforms RapTB alone on coverage by a large margin. The paper acknowledges this in Appendix A.1 for SMILES ("applying SubM to TB also yields substantial improvements"), but unfortunatley does not discuss it in the main text as a challenge to the narrative that RapTB is the primary driver of improvement. The strongest result in the paper, that RapTB+SubM reaching NormCov=0.209, doubling TB+SubM, is really strong, but the paper does not clearly characterize when RapTB provides additive benefit over SubM alone vs. when SubM dominates.

3. RapTB introduces seven task-specific hyperparameters beyond the base TB objective: aux weight (eta), distance discount (gamma), mix weight between max and soft backup (alpha), soft backup temperature (beta), distance penalty (rho), minimum prefix depth (k_min, with a linear schedule), and auxiliary horizon cap (K). From Table 23, these differ meaningfully between tasks (for example, alpha is 0.5 for SMILES and 0.8 for Expr24, rho is 0.1 for SMILES and 0.5 for Expr24) yet the Table 6 results only show ablations for reward absorption on/off and max-only vs. soft-only backups, leaving alpha, beta, rho, eta, and gamma entirely unablated. As such, the sensitivity of the main results to these choices is not known. The k_min schedule, which linearly anneals the minimum eligible prefix depth from 5 to 2 over 5000 training steps for SMILES, is a non-trivial design choice that determines which prefixes receive auxiliary supervision during the critical early phase of training, and the authors never study its effect.

---

> ### Author Rebuttal · Authors · 2026-03-31
>
> We thank the reviewer for the careful reading and insightful comments.
>
> ## W1 + Q3: Domain and model generalization
> We acknowledge that the evaluation scope in the submission was limited. We have since added two additional evaluation experiments. Owing to space constraints, we kindly refer the reviewer to **Reviewer Pd1v**’s **W1 section** for details.
>
> ---
>
>
> ## W2 + Q1: When does RapTB provide an additive benefit over SubM?
>
> We agree with the reviewer: Table 3 shows that when exploration is the main bottleneck, SubM alone can outperform RapTB alone on coverage (Expr24: NormCov (0.100) for TB+SubM vs. (0.039) for RapTB+RP). We will make this clearer in the main text. Our claim is not that RapTB is always better, but that **SubM and RapTB are complementary**: SubM improves replay coverage, while RapTB improves credit assignment within trajectories, thereby also improving the quality of trajectories collected into the replay buffer. As a result, **RapTB’s additive benefit becomes most visible once coverage is no longer the limitation**: with SubM replay, RapTB further improves NormCov from (0.100) to (0.209); with Oracle replay, where coverage is not a bottleneck, RapTB still improves Acc and JS, showing its benefit in mapping target distribution. In short, **SubM dominates in severely coverage-limited regimes, while RapTB provides the strongest gains when replay is already sufficiently informative.**
>
>
> ---
>
> ## W3 + Q2: Hyperparameter sensitivity
>
> We conduct cross-task sweeps over $\beta \times \rho$ (18 configs) plus $\eta$ and $k_{\min}$ ablations. Across all 18 settings, we observe no catastrophic failure, length collapse, or termination drift. As for other hyperparameters like $K$ (prefix absorb upper bound) and $\alpha$ (mixture ratio of soft and max absorb strategy), they are fixed in experiments or reported in the paper (Table 6), which are not covered in sweep experiments.
>
> ### 1. Expr24 task
>
> **1.a $\quad \beta \times \rho$ Grid (9 experiments)**
>
> | $\beta$ | $\rho$ | Acc ↑ | Diversity ↑ |
> |---|---|---:|---:|
> | 1 | 0.0 | 0.999 |1.010|
> | 1 | 0.1 | 0.999 |0.993|
> | 1 | 0.5 | 1.000 | 1.015 |
> | 3 | 0.0 | 0.990 | 0.950 |
> | 3 | 0.1 | 1.000 | 0.769 |
> | 3 | 0.5 | 0.997 | 0.978 |
> | 5 | 0.0 | 0.983 | 1.022 |
> | 5 | 0.1 | 0.998 | 0.968 |
> | 5 | 0.5 | 0.999 | 0.912 |
>
> $\beta$ and $\rho$ denote the soft-backup temperature and distance-discount factors, respectively. They primarily govern the bias–variance tradeoff of the absorbed target, thereby affecting the balance between accuracy and diversity.
>
> **1.b $\quad \eta$ Sweep ($\beta=3, \rho=0.5$)**
>
> | $\eta$ | Acc ↑ | Diversity ↑ |
> |---|---:|---:|
> | 0.1 | 0.999 | 0.983 |
> | 0.25 | 0.997 | 0.978 |
> | 0.5 | 0.987 | 1.149 |
>
> Monotonic trend: higher $\eta$ leads to better diversity ($0.983 \rightarrow 1.149$), with only a mild trade-off in accuracy. This suggests that increasing $\eta$ improves prefix credit assignment and thus enhances diversity, albeit at some cost to accuracy.
>
> **1.c $\quad k_{\min}$ Ablation ($\beta=3, \rho=0.5, \eta=0.25$)**
>
> | $k_{\min}$ variant | Acc ↑ | Diversity ↑ |
> |---|---:|---:|
> | Fixed $k=3$ | 0.998 | 0.852 |
> | Schedule $7 \to 3$ | 0.997 | 0.978 |
> | Fixed $k=7$ | 0.969 | 1.025 |
>
> $k_{\min}$  is the minimum prefix depth for auxiliary supervision. Smaller $k_{\min}$  emphasizes shorter prefixes. This creates a "shortcut path" — the model can allocate flow to high-reward short sequences more easily, thereby improving quality but reducing diversity. k_min=3 (fixed) is worst for diversity (0.852); the linear schedule (7→3) used in the paper provides a good balance.
>
>
> ### Table 2: SMILES
>
> **2.a $\quad \beta \times \rho$ Grid (9 experiments)**
> | $\beta$ | $\rho$ | Acc ↑ | Entropy ↑ |
> |---|---|---:|---:|
> | 1 | 0.0 | 0.992 | 2.173 |
> | 1 | 0.1 | 0.994 | 2.161 |
> | 1 | 0.5 | 0.992 | 2.162 |
> | 5 | 0.0 | 0.995 | 1.997 |
> | 5 | 0.1 | 0.991 | 2.076 |
> | 5 | 0.5 | 0.997 | 2.079 |
> | 10 | 0.0 | 0.968 | 2.279 |
> | 10 | 0.1 | 0.999 | 1.986 |
> | 10 | 0.5 | 0.997 | 2.036 |
>
> 8/9 configs Acc $\geq 0.991$; only $(\beta=10, \rho=0)$ shows mild degradation (0.968) — high temperature with zero distance penalty. QED $\in [0.727, 0.804]$ (paper: 0.740), FPDiv $\in [0.849, 0.883]$ (paper: 0.860). No length collapse: all Len $\in [6.7, 7.7]$, far from TB's collapse (Len=3.06).
>
> ### Q4: GAE analogy
>
> The connection is a structural analogy in variance reduction, not an equivalence. Key difference is in what is being aggregated:
> GAE [1] aggregates intermediate value estimates to form an advantage target, whereas RapTB starts from an already available terminal reward and redistributes that signal to intermediate subtrajectories/prefixes that otherwise receive little supervision.
>
> We will revise the text to make the scope explicit and clarify that the analogy concerns variance reduction through structured credit assignment, not objective-level equivalence.
>
> ### References
> [1] Schulman et al. “High-Dimensional Continuous Control Using Generalized Advantage Estimation” ICLR 2016.

---

> > ### Author Rebuttal · Reviewer_cA3o · 2026-03-31
> >
> > The authors have fully addressed my concerns, especially with the hyperparameter sweeps, explaining the SubM vs. RapTB tradeoffs, and the clarification of the GAE connection. I support the acceptance of the paper, and will increase my score to a 5.

---

> > > ### Author Response · Authors · 2026-04-01
> > >
> > > Thank you for the careful reading and thoughtful feedback! We really appreciate your updated assessment and your support for acceptance after reading our rebuttal. We are glad the additional analyses and clarifications addressed your concerns, and your comments helped us a lot to improve the paper :)

---

### Official Review · Reviewer_Pd1v · 2026-03-18

**Soundness:** 2
**Presentation:** 2
**Significance:** 2
**Originality:** 2
**Overall Recommendation:** 3
**Confidence:** 2

**Summary:**

The paper aims to fix (1) weak credit assignment to early prefixes and (2) biased replay when using GFlowNet to finetune LLM to approximate reward-proportional posterios. Specifically to address (1), the paper introduces Rooted absorbed prefix Trajectory Balance as a new objective. To mitigate the replay bias problem (2), the paper proposes a submodular replay refresh strategy that promotes both high
reward and diversity. Experiments are conducted on SMILE-based molucule generation with LLM.

**Compliance With Llm Reviewing Policy:**

Affirmed.

**Final Justification:**

After reading the rebuttal, I am still not convinced that the method is going to be pratically useful, and moreover, the provided response does not really address my previously mentioned weakness of the theoretical results. Thus I maintain my original rating.

**Key Questions For Authors:**

See the weakness part above.

**Strengths And Weaknesses:**

Strengths

- The paper compares the proposed RapTB with standard tajectory balance and subtrajectory balance. RapTB additionally considers rooted prefix supervision and absorbed suffix rewards.

- Submodular replay as an additional improvement to enforce diversity in addition to the exploration of GFlowNet.

- SMILES-based molecule generation with LLMs are interesting applications. Experiments show decent effectiveness.

Weaknesses

- The experimental validation is not strong enough. Experiments mostly use a small LLM and the evaluated benchmarks are a bit narrow, as it only covers molecule generation, synthetic arithmetic and CommonGen. To support the claims made in the paper, it will be better to include LLMs with larger size and more benchmarks.

- The baseline methods are not sufficient, as it only compares TB and SubTB. I think there are definitely more competitive baseline methods available.

- No theoretical guarantees are provided for RapTB to show that it can indeed preserve the desired reward-proportional terminal distribution. It will be nice to have a deeper analysis regarding the effectiveness of RapTB.

---

> ### Author Rebuttal · Authors · 2026-03-31
>
> We thank Reviewer Pd1v for the constructive comments. We address each weakness below with new experimental evidence and clarified theoretical analysis.
>
> ## W1: LLM size and benchmarks evaluated narrow
>
> We acknowledge that the previous evaluation is limited. We now add two more experiments:
>
> **(1) AMP biological sequence generation** [1], a standard GFlowNet benchmark with amino-acid vocabulary, covering more baselines:
> | Method        | Performance ↑ | Diversity ↑ | Novelty ↑ | Avg Len |  Training Steps |
> |---------------|------|---------|---------|-----|-----|
> | RapTB+SubM    | 0.916        | 16.92       | 15.77     | 25.6    | 3K    |
> | RapTB         | 0.919        | 8.83        | 14.44     | 22.4    | 5K    |
> | TB            | 0.927        | 7.39        | 10.65     | 17.4    | 10K   |
> | SubTB †       | 0.897        | 21.37       | 28.68     | 49.3 †  | 10K   |
> | DynaPPO       | 0.938        | 12.12       | 9.31      | ~20       | 10K   |
> | COMs          | 0.761        | 19.38       | 26.47     | ~20       | 10K   |
> | GFlowNet      | 0.868        | 11.32       | 15.72     | ~20      | 10K   |
>
> †: SubTB collapses to maximum length; its diversity/novelty is inflated by raw edit distance over unnaturally long sequences. RapTB+SubM achieves the best performance/diversity/novelty tradeoff within 3K steps.
>
> **(2) Scale-up Llama-3.2 from 1B to 3B** on SMILES:
> | Method (3B)   | Acc ↑ | Score ↑ | FPDiv ↑ | Avg Len |
> |---------------|------|---------|---------|-----|
> | TB            | 0.999 | 0.717 | 0.837 | 2.74 |
> | SubTB         | 0.313 | 0.221 | 0.854 | 8.48 |
> | RapTB         | 0.984 | 0.732 | 0.864 | 6.86 |
> | RapTB+SubM    | **0.996** | **0.856** | **0.937** | **7.96** |
>
> Scaling to 3B should improve results, but SubTB degrades, confirming the failure mode is structural. Together with the original results, our evidence now spans tasks including molecules, arithmetic expressions, natural language, biological sequences, and two model scales.
>
> ---
>
> ## W2: More baselines
> Our original comparison focuses on comparing the loss (training objective) design. RapTB modifies the TB-family loss for terminable prefix-tree LLM-GFlowNets [2], making TB and SubTB the direct loss baselines. To provide a more comprehensive comparison, we further include three additional methods (AvgPrefixTB, GRPO, PPO):
>
> ### 1. Native PPO and GRPO:
>
> **1.a SMILES generation task.**
>
> | Method | Acc ↑ | Score ↑ | Entropy ↑ | Avg Len |
> |--------|------|------|-----------|---------|
> | GRPO   | 0.997 | 0.661 | 0.98 | 10.0 |
> | PPO    | 1.000 | 0.604 | 0.00 | collapsed |
> | TB     | 0.998 | 0.717 | 2.503 | 3.06 |
> | RapTB  | 0.996 | 0.740 | 2.448 | 6.14 |
>
> **1.b VarExpr24 task.**
>
> | Method | Acc ↑  | Unique ↑ | Avg Len |
> |--------|------|----------|---------|
> | GRPO   | 0.002  | 1 | ~11 |
> | PPO    | 0.003 | 1 | collapsed |
> | TB     | 1.000 | 5.3 | 8.98 |
> | RapTB  | 0.991 |  246.7 | 8.99 |
>
> RL baselines improve reward but collapse diversity. This matches our hypothesis that the fundamental difference between reward maximization and reward-proportional sampling lies in their objectives. This phenomenon is also reported in previous studies [2].
>
>
> ### 2. AvgPrefixTB.
>
>
> We also include comparison results for AvgPrefixTB, as suggested by **Reviewer QHmk**; ***see that section Q2 for details.***
>
> ---
>
> ## W3: No theoretical guarantees.
> We acknowledge that the paper does not provide a formal proof establishing the preservation of a reward-proportional terminal distribution. To address this, we present a proof by decomposition, analyzing the problem from two complementary perspectives:
>
> **Angle 1** (regularizer structure): RapTB decomposes as
> $$L_{\text{RapTB}} = L_{\text{TB}} + \eta L_{\text{aux}} $$
>
> The TB fixed point (Malkin et al. 2022 [3]) is equivalent to the MaxEnt RL optimal policy [4]. The auxiliary term acts as a variance-reducing regularizer. RapTB preserves the TB fixed point while improving credit assignment.
>
> **Angle 2** (zero-at-optimum): In a terminable prefix tree,
> If:
> $$
> \Delta_k^{TB} = 0 \quad \forall k
> $$
>
> Then:
>
> - Rooted residuals:
> $$
> \bar{\Delta}_k = \Delta_k^{TB} - \Delta_0^{TB} = 0
> $$
> - Auxiliary term vanishes:
> $$
> L_{\text{aux}} = 0
> $$
>
> Thus:
> $$
> L_{\text{RapTB}} = (\Delta^{TB})^2 + \eta \cdot 0 = 0
> $$
>
> The reward-proportional distribution is a joint global minimum of both loss terms, mirroring SubTB’s telescoping-based fixed-point property [5]. We will make the zero-at-optimum property explicit in the revision and narrow our claims accordingly.
>
> ---
> References
>
> [1] Jain et al. "Biological Sequence Design with GFlowNets." ICML 2022.
>
> [2] Hu et al. "Amortizing intractable inference in large language models." ICLR 2024.
>
> [3] Malkin et al. "Trajectory Balance: Improved Credit Assignment in GFlowNets." NeurIPS 2022.
>
> [4] Tiapkin et al. "GFlowNets as Entropy-Regularized RL." AISTATS 2024.
>
> [5] Madan et al. "Learning GFlowNets from partial episodes for improved convergence and stability." ICML 2023.

---

> > ### Author Rebuttal · Reviewer_Pd1v · 2026-04-01
> >
> > While I appreciate the detailed rebuttal, I am still not convinced, epsecially given the short amount of time, the empirical effectiveness in a larger-scale setting can not be well tested. I suggest the authors to carefully evaluate the method with larger-size LLMs. Moreover, the theory part is still quite weak and my major concerns remain unresolved.

---

> > > ### Author Response · Authors · 2026-04-06
> > >
> > > We thank the reviewer for the insightful sugesstions. We address the two remaining concerns: **larger-scale evaluation** and **theoretical guarantees** with new evidence below.
> > >
> > > ---
> > >
> > > ## W1 (Revised): Large-Scale LLM Evaluation
> > >
> > > We completed a scaling study across **3B, 8B, 32B** and **two architectures (Llama-3.2, Qwen3)** on SMILES:
> > >
> > > | Scale | Arch | Method | Acc | QED | FPDiv | Div | Len |
> > > |-------|-------|------------|-------|-------|-------|-------|------|
> > > | **3B** | Llama | TB | 0.999 | 0.716 | 0.838 | 1.92 | 2.69 |
> > > | | | SubTB | 0.311 | 0.222 | 0.854 | **2.56** | 9.52 |
> > > | | | RapTB | **1.000** | 0.795 | 0.839 | 1.81 | 7.99 |
> > > | | | RapTB+SubM | 0.998 | **0.869** | **0.936** | 2.41 | 8.05 |
> > > | **8B** | Llama | TB | **1.000** | 0.715 | 0.775 | 1.84 | 2.98 |
> > > | | | SubTB | 0.391 | 0.307 | 0.869 | **2.72** | 9.22 |
> > > | | | RapTB | 0.999 | 0.825 | 0.852 | 1.89 | 8.09 |
> > > | | | RapTB+SubM | 0.998 | **0.873** | **0.937** | 2.51 | 7.65 |
> > > | **32B** | Qwen3 | TB | **1.000** | 0.762 | 0.860 | 2.055 | 3.15 |
> > > | | | SubTB | 0.795 | 0.626 | 0.864 | 2.186 | 7.31 |
> > > | | | RapTB | 0.998 | 0.794 | 0.880 | 2.160 | 6.19 |
> > > | | | RapTB+SubM | 0.998 | **0.867** | **0.896** | **2.231** | 7.56 |
> > >
> > > **(a) SubTB termination drift persists.** Accuracy is low (0.311, 0.391, 0.795 at 3B, 8B, 32B). Even 32B cannot fully resolve this issue.
> > >
> > > **(b) RapTB scales reliably.** Accuracy stays above 0.998 at all scales, with diversity *improving* with size (FPDiv: 0.839, 0.852, 0.880).
> > >
> > > **(c) RapTB+SubM achieves the best quality-diversity tradeoff.** At 32B: highest QED (0.867), longest valid molecules (Len=7.56), and strong Acc (0.998).
> > >
> > > These results across two architecture families up to 32B confirm that SubTB failure modes are structural and architecture-independent.
> > >
> > > ---
> > >
> > > ## W3 (Revised): Theoretical Guarantees for RapTB
> > >
> > > We provide a formal analysis structured as **"exact anchor + bounded regularization"**.
> > >
> > > ### (i) TB Is the Exact Anchor
> > >
> > > On a directed prefix tree with unique backward paths, the TB optimum uniquely determines the reward-proportional policy. Let X denote the full terminal set covering all stop lengths. The optimal policy is:
> > >
> > > $$q^{\ast}(a \mid s) = \frac{F^{\ast}(sa)}{F^{\ast}(s)}, \quad q^{\ast}(\top \mid s) = \frac{R(s^{\top})}{F^{\ast}(s)},$$
> > >
> > > where $F^{\ast}(s) = \sum_{x \in X(s)} R(x)$ is the flow at prefix $s$.
> > >
> > > ### (ii) Rooted Residuals Are Compatible with TB
> > >
> > > The rooted residual decomposes as a difference of two TB residuals:
> > >
> > > $$\bar{\Delta}_k = \Delta_k^{TB} - \Delta_0^{TB}.$$
> > >
> > > At an exact TB optimum all prefix residuals vanish, hence every rooted residual also vanishes. Therefore, **rooted/no-absorb RapTB shares the identical global optimum as TB** -- the reward-proportional distribution is preserved exactly.
> > >
> > > ### (iii) Absorbed RapTB: Bounded Regularization
> > >
> > > With reward absorption, the residual becomes:
> > >
> > > $$\bar{\Delta}_k^{Rap}(\xi) = \bar{\Delta}_k(\xi) + \gamma^{h-k}(u_k - u_k^{tgt}).$$
> > >
> > > The absorbed target is a mixture of max and softmax suffix backups: $u_k^{tgt} = \alpha u_k^{max} + (1-\alpha) u_k^{soft}$. Since the suffix set always includes the self term, $u_k^{tgt} \ge u_k$, with strict inequality when a later suffix carries stronger signal. Therefore, even at an exact TB optimum where the rooted residual vanishes, the absorbed residual is generally nonzero.
> > >
> > > Now consider the combined loss $L_{RapTB} = L_{TB} + \eta L_{aux}$. Let $\bar{\theta}$ denote an exact TB solution satisfying $L_{TB}(\bar{\theta})=0$, which incurs a finite auxiliary cost $L_{aux}(\bar{\theta})=B$. Let $\theta'$ be any global minimizer of $L_{RapTB}$. Since $L_{TB} \ge 0$:
> > >
> > > $$L_{TB}(\theta') \le L_{RapTB}(\theta') \le L_{RapTB}(\bar{\theta}) = 0 + \eta B = \eta B.$$
> > >
> > > The first inequality holds because $L_{TB}$ is a component of $L_{RapTB}$; the second holds because $\theta'$ minimizes $L_{RapTB}$ globally. This means the TB loss at the RapTB optimum is controlled by $\eta B$: as $\eta$ approaches 0, exact TB balance is recovered. No-absorb yields $B=0$, where both objectives share the same optimum.
> > >
> > > ---
> > >
> > > ## Summary
> > >
> > > Combined with our initial rebuttal, the evidence now spans **five tasks** (SMILES, VarExpr24, CommonGen, AMP, scaling), **three scales** (1B/3B, 8B, 32B across Llama and Qwen3), **eight baselines** (TB, SubTB, AvgPrefixTB, PPO, GRPO, DynaPPO, COMs, GFlowNet), and a theoretical analysis. We hope these additions fully address the reviewer's concerns.

---

### Decision · Program_Chairs · 2026-04-30

**Decision:**

Accept (regular)

**Comment:**

Reviewers agreed that the paper provides a strong mechanistic diagnosis of SubTB failures and proposes a highly original, effective combination of rooted prefix constraints and submodular replay for LLM-GFlowNet training. While some reviewers initially raised concerns regarding the scale of empirical validation and the depth of theoretical guarantees, the authors' extensive rebuttal with new scaling experiments and further theoretical decomposition adequately addressed the majority of these concerns. The technical contributions are solid, well-evaluated, and will be valuable to the community. Therefore, the paper is recommended for acceptance.